# Metadata-based Multi-Task Bandits
# with Bayesian Hierarchical Models

**Runzhe Wan**    **Lin Ge**    **Rui Song**
Department of Statistics
North Carolina State University
{rwan, lge, rsong}@ncsu.edu

## Abstract

How to explore efficiently is a central problem in multi-armed bandits. In this paper, we introduce the metadata-based multi-task bandit problem, where the agent needs to solve a large number of related multi-armed bandit tasks and can leverage some task-specific features (i.e., metadata) to share knowledge across tasks. As a general framework, we propose to capture task relations through the lens of Bayesian hierarchical models, upon which a Thompson sampling algorithm is designed to efficiently learn task relations, share information, and minimize the cumulative regrets. Two concrete examples for Gaussian bandits and Bernoulli bandits are carefully analyzed. The Bayes regret for Gaussian bandits clearly demonstrates the benefits of information sharing with our algorithm. The proposed method is further supported by extensive experiments.

## 1 Introduction

The multi-armed bandit (MAB) is a popular framework for sequential decision making problems, where the agent will sequentially choose among a few arms and receive a random reward for the arm [37]. Since the mean rewards for the arms are not known *a priori* and must be learned through partial feedbacks, the agent is faced with the well-known exploration-exploitation trade-off. MAB is receiving increasing attention and has been widely applied to areas such as clinical trials [19], finance [54], recommendation systems [70], among others.

How to explore efficiently is a central problem in MAB. In many modern applications, we usually have a large number of separate but related MAB tasks. For example, in e-commerce, the company needs to find the optimal display mode for each of many products, and in medical applications, each patient needs to individually undergo a series of treatment periods to find the personalized optimal treatment. These tasks typically share similar problem structures, but may have different reward distributions. Intuitively, appropriate information sharing can largely speed up our learning process and reduce the regret, while a naive pooling may cause a linear regret due to the bias.

This paper is concerned with the following question: *given a large number of MAB tasks, how do we efficiently share information among them?* The central task is to capture task relations in a principled way [68]. While a few approaches have been proposed (see Section 2), most of them only utilize the action-reward pairs observed for each task. To our knowledge, none of the existing works can leverage another valuable information source, i.e., the *metadata* of each task, which contains some *static* task-specific features. Such metadata is commonly available in real applications [66, 68, 57], such as the demographic features of each customer or patient, or basic information of each web page or product. Although the metadata can hardly be directly utilized in a single MAB task, it usually contains intrinsic information about each task, and hence can guide us to learn task relations and efficiently share knowledge in the multi-task setup. Specifically, suppose task $i$ has a feature vector $\mathbf{x}_i$ (i.e., metadata) and its expected reward vector for all arms is $\boldsymbol{r}_i$. We consider a general formulation that $\boldsymbol{r}_i$ is sampled from the conditional distribution $\mathbb{P}(\boldsymbol{r}_i|\mathbf{x}_i)$. When $\mathbb{P}(\boldsymbol{r}_i|\mathbf{x}_i)$ is

35th Conference on Neural Information Processing Systems (NeurIPS 2021).

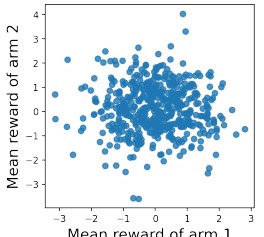 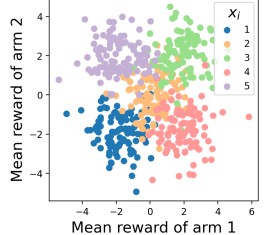 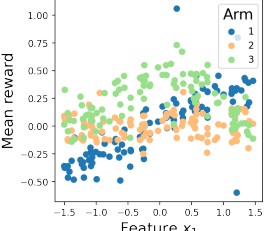

**(a)** The meta MAB setting [34, 28, 8], where $\mathbf{x}_i$ is a constant.

**(b)** There exists several task clusters, where $\mathbf{x}_i$ is categorical.

**(c)** $\mathbf{x}_1$ is a continuous variable.

**Figure 1:** Illustrations of the task distribution $\mathbb{P}(\boldsymbol{r}_i|\mathbf{x}_i)$. In the first two subplots, each dot represents a task and the two axes denote the expected rewards for two arms. In the last subplot, we have three arms, each dot represents one arm of a task, the x-axis denotes one continuous feature $x_1$, and the y-axis denotes the expected reward. In the last two settings, the metadata is informative and can guide the exploration to reduce regrets, while there is still inter-task heterogeneity that can not be captured.

informative, the metadata, being appropriately utilized, can guide our exploration and hence reduce the regret. Several specific cases of $\mathbb{P}(\boldsymbol{r}_i|\mathbf{x}_i)$ are illustrated in Figure 1.

As one of our motivating examples, consider that a company needs to learn about the personal preference of many users over several options (e.g., message templates), via sequential interactions with each user. Suppose some user features (e.g., gender, age, geography) are informative in predicting the preference. Then, even before any interactions with a specific user, we can predict her preference with high probability, based on her features and our experience with other users. Such knowledge can be concisely encoded as $\mathbb{P}(\boldsymbol{r}_i|\mathbf{x}_i)$. However, $\boldsymbol{r}_i$ typically can not be fully determined by $\mathbf{x}_i$. Given the possible heterogeneity of $\boldsymbol{r}_i$ conditional on $\mathbf{x}_i$, the true reward distribution (and optimal arm) of this specific user still needs to be learned and confirmed via interactions with herself. Therefore, we must carefully utilize these features in a multi-task algorithm. Indeed, we conjecture this is one way how humans adapt to tasks effectively, by constructing a rough first impression according to features, and then adjusting the impression gradually in the following interactions.

To judiciously share information and minimize cumulative regrets, we design a multi-task bandit algorithm. Recall that, when $\mathbb{P}(\boldsymbol{r}_i|\mathbf{x}_i)$ is known *a priori*, a Thompson sampling (TS) algorithm [53] with $\mathbb{P}(\boldsymbol{r}_i|\mathbf{x}_i)$ as the prior of $\boldsymbol{r}_i$ for each task $i$ is known to be (nearly) optimal in Bayes regret, for a wide range of problems [37]. However, such knowledge typically is absent, and many concerns have been raised regarding that TS is sensitive to the prior [42, 25, 7, 23]. In this work, we aim to aggregate data from all tasks to construct an informative prior for each task. Specifically, we propose to characterize the task relations via a Bayesian hierarchical model, upon which the Multi-Task TS (MTTS) algorithm is designed to continuously learn $\mathbb{P}(\boldsymbol{r}_i|\mathbf{x}_i)$ by aggregating data. With such a design, MTTS yields a similar low regret to the oracle TS algorithm which knows $\mathbb{P}(\boldsymbol{r}_i|\mathbf{x}_i)$ *a priori*. In particular, the metadata $\mathbf{x}_i$ allows us to efficiently capture task relations and share information. The benefits can be seen clearly from our experiments and analysis.

**Contribution.** Our contributions can be summarized as follows. First of all, motivated by the usefulness of the commonly available metadata, we introduce and formalize the metadata-based multi-task MAB problem. The related notion of regret (multi-task regret) is defined, which serves as an appropriate metric for this problem. Second, we propose to address this problem with a unified hierarchical Bayesian framework, which characterizes the task relations in a principled way. We design a TS-type algorithm, named MTTS, to continuously learn the task distribution while minimizing the cumulative regrets. The framework is general to accommodate various reward distributions and sequences of interactions, and interpretable as one can study how the information from other tasks affects one specific task and how its metadata determines its prior. Two concrete examples for Gaussian bandits and Bernoulli bandits are carefully analyzed, and efficient implementations are discussed. Our formulation and framework open a door to connect the multi-task bandit problem with the rich literature on Bayesian hierarchical models. Third, we theoretically demonstrate the benefits of metadata-based information sharing, by deriving the regret bounds for MTTS and several baselines under a Gaussian bandit setting. Specifically, the average multi-task regret of MTTS decreases as the number of tasks grows, while the other baselines suffer a linear regret. Lastly, systematic simulation experiments are conducted to investigate the performance of MTTS under different conditions, which provides meaningful insights.

## 2 Related Work

First of all, our setup subsumes the meta MAB problem [28, 10, 34, 35] as a special case, where $\mathbf{x}_i$ degenerates to a constant. Meta MAB assumes the tasks arrive *sequentially* and are drawn from one *metadata-agnostic* distribution. Among these works, meta TS [34] is the most related one, which assumes the mean reward vectors $\{\boldsymbol{r}_i\}$ are i.i.d. sampled from some distribution $f(\boldsymbol{r}; \boldsymbol{\theta})$, and pool data to learn $\boldsymbol{\theta}$. The other works either pool data from accomplished tasks to optimize algorithm hyper-parameters [28] or directly optimize the policy via policy gradient [10, 35]. Compared with meta MAB, our setting is more general as we allow incorporating metadata to learn and capture task relations, and also allow the interactions with different tasks to arrive in arbitrary orders. The former is particularly important when there exists significant heterogeneity among tasks (e.g., setting (c) in Figure 1), where a naive pooling may lose most information; the latter allows broader applications. The results developed in this paper are equally applicable to meta MAB.

In addition to meta MAB, there are several other topics concerned with multiple bandit tasks. We review the most related papers in this section, and provide a comprehensive comparison with additional related works and concepts in Appendix B. First of all, there is a large number of works on multi-agent bandits [30, 12, 62, 17] studying that multiple agents interact with the same task concurrently. In contrast, we consider there is inter-task heterogeneity. Given such heterogeneity, certain approaches to capture task relations are then required: when there exist useful network structures between tasks, [13] and [61] propose to utilize this knowledge to capture task relations; [6] assumes there is a *finite* number of task instances, considers the tasks arrive sequentially, and aims to transfer knowledge about this set of instances; [64] assumes there exist pairwise upper bounds on the difference in reward distributions; clustering of bandits [22, 40] and latent bandits [26, 27] assume there is a perfect clustered structure, that tasks in the same cluster share the same reward distribution. In contrast to all these setups, to our knowledge, this is the first work that can utilize metadata, an important information source that is commonly available. Practitioners are typically familiar with feature-based methods and our work adds one useful tool to the multi-task bandits toolbox. Besides, our modelling-based approach provides nice interpretability and is flexible to accommodate arbitrary order of interactions. Lastly, we learn task relations from data automatically, with arguably fewer structural assumptions.

In the presence of task-specific features, another approach to pool information is to adapt contextual MAB (CMAB) algorithms [43, 69, 51, 4, 58, 36, 38], by neglecting the task identities and regarding the metadata as "contexts". Specifically, one can assume a *deterministic* relationship $\boldsymbol{r}_i = f(\mathbf{x}_i; \boldsymbol{\theta})$ for some function $f$, and then apply CMAB algorithms to learn $\boldsymbol{\theta}$. However, such a solution heavily relies on the *realizability* assumption [21], that $\boldsymbol{r}_i = f(\mathbf{x}_i; \boldsymbol{\theta})$ holds exactly with no variation or error. When this assumption is violated, a linear regret is generally unavoidable. This limitation attracts increasing attention [21, 33, 32]. In our setup, the agent will interact with each task multiple times and the metadata is fixed, which naturally enables us to relax this restrictive assumption by regarding $\boldsymbol{r}_i$ as a random variable sampled from $\mathbb{P}(\boldsymbol{r}_i|\mathbf{x}_i)$. To our knowledge, this is the first work that can handle the realizability issue via utilizing the repeated-measurement structure.

Finally, there exists a rich literature on multi-task supervised learning (see [67] for a survey), where hierarchical models [44, 55, 63] are a popular choice, [49, 3, 46] studied aggregating historical data to construct informative priors, and [68] considered leveraging metadata to cluster tasks. Besides, this work is certainly related to single-task bandits. In particular, Thompson sampling is one of the most popular approaches with superior performance [1, 53, 60, 34, 2, 18]. Its popularity is partially due to its flexibility to incorporate prior knowledge [14]. We provide a data-driven approach to construct such a prior. Our work naturally builds on these two areas to provide an efficient solution to the metadata-based multi-task MAB problem.

## 3 Metadata-based Multi-task MAB

In this section, we introduce a general framework for the metadata-based multi-task MAB problem. For any positive integer $M$, we denote the set $\{1, 2, \ldots, M\}$ by $[M]$. Let $\mathbb{I}$ be the indicator function. Suppose we have a large number of $K$-armed bandit tasks with the same set of actions, denoted as $[K] = \{1, \ldots, K\}$. Each task instance $i$ is specified by a tuple $(\boldsymbol{r}_i, \mathbf{x}_i)$, where $\boldsymbol{r}_i = (r_{i,1}, \ldots, r_{i,K})^T$ is the vector of mean rewards for the $K$ arms and $\mathbf{x}_i$ is a $p$-dimensional vector of task-specific features (i.e., metadata). The metadata can include categorical and continuous variables, and even outputs from pre-trained language or computer vision models [16, 39]. We assume these task in-

stances are i.i.d. sampled from some unknown *task distribution* $\mathcal{P}_{\boldsymbol{r},\mathbf{x}}$, which induces a conditional distribution $\mathbb{P}(\boldsymbol{r}_i|\mathbf{x}_i)$. We will interact with each task repeatedly. For each task $i$, at its $t$-th decision point, we choose one arm $A_{i,t} \in [K]$, and then receive a random reward $R_{i,t}$ according to $R_{i,t} = \sum_{a=1}^{K} \mathbb{I}(A_{i,t} = a)r_{i,a} + \epsilon_{i,t}$, where $\epsilon_{i,t}$ is a mean-zero error term.

We call this setup *multi-task MAB* to focus on the fact that there is more than one MAB task. New tasks may appear and interactions with different tasks may happen in arbitrary orders. When the tasks arrive in a sequential manner (i.e., task $i + 1$ begins only when task $i$ is finished), our setup is an instance of *meta learning* [34, 28, 8, 59], which we refer to as the *sequential* setting in this paper.

Suppose until a time point, we have interacted with $N$ tasks, and for each task $i \in [N]$, we have made $T_i$ decisions. The performance of a bandit algorithm can be measured by its *Bayes regret*,

$$BR(N, \{T_i\}) = \mathbb{E}_{\mathbf{x},\boldsymbol{r},\epsilon} \sum\nolimits_{i\in[N]} \sum\nolimits_{t\in[T_i]} (max_{a\in[K]} r_{i,a} - r_{i,A_{i,t}}),$$

where the expectation is taken with respect to the task instances $\{(\boldsymbol{r}_i, \mathbf{x}_i)\}$, the random errors, the order of interactions, and the randomness due to the algorithm. Here, the metadata $\{\mathbf{x}_i\}$ and the order of interactions enter the definition because they might affect the decisions of a multi-task algorithm. When the algorithm does not share information across tasks, the definition reduces to the standard single-task Bayes regret [37] accumulated over $N$ tasks. The Bayes regret is particularly suitable for multi-task bandits, under the task distribution view [67, 34].

Given the existence of inter-task heterogeneity, the Bayes regret of any bandit algorithms will unavoidably scale linearly with $N$. We next introduce a new metric, refer to as the *multi-task regret*, which allows us to clearly see the benefits of multi-task learning and can serve as an appropriate metric for multi-task bandit algorithms.

To begin with, we first review Thompson sampling (TS), which is one of the most popular bandit algorithm framework [1, 53, 60]. For each task $i$, the standard single-task TS algorithm requires a prior $Q(\boldsymbol{r}_i)$ as the input, maintains a posterior of $\boldsymbol{r}_i$, and samples an action according to its posterior probability of being the optimal one. We denote $TS(Q(\boldsymbol{r}))$ as a TS algorithm with some distribution $Q(\boldsymbol{r})$ as the prior. Most Bayes regret guarantees for TS require the specified prior is equal to the true distribution of $\boldsymbol{r}$ [53, 37]. When this condition is satisfied, it is well-known that this TS algorithm is (nearly) optimal for a wide range of problems [53, 37]. In our setup, this assumption means that, for each task $i$, we will apply $TS\big(\mathbb{P}(\boldsymbol{r}_i|\mathbf{x}_i)\big)$. It is natural to set such an (nearly) optimal algorithm as our skyline, which we refer to as *oracle-TS*.

However, it is arguably a strong assumption that the agent can know $\mathbb{P}(\boldsymbol{r}_i|\mathbf{x}_i)$ *a priori*, and a growing literature finds that TS with a mis-specified prior can yield a poor regret due to over-exploration [14, 42, 25, 53, 23]. Therefore, a well-designed metadata-based multi-task MAB algorithm should be able to aggregate data to learn the task distribution pattern (i.e., $\mathbb{P}(\boldsymbol{r}_i|\mathbf{x}_i)$), so as to yield similar low regrets with oracle-TS. Motivated by these discussions, we define the *multi-task regret* of an algorithm as the difference between its Bayes regret and the Bayes regret of oracle-TS, which can be equivalently written as

$$MTR(N, \{T_i\}) = \mathbb{E}_{\mathbf{x},\boldsymbol{r},\epsilon} \sum\nolimits_{i\in[N]} \sum\nolimits_{t\in[T_i]} (r_{i,A_{i,t}^{\mathcal{O}}} - r_{i,A_{i,t}}),$$

where $A_{i,t}^{\mathcal{O}}$ is the action that oracle-TS takes, and the expectation is additionally taken over its randomness. This regret reflects the price of not knowing the prior *a priori*, and measures the performance of a multi-task algorithm in learning the task distribution pattern while maintaining a low regret. A similar concept is defined in [7] for meta learning in dynamic pricing.

## 4 Algorithm: MTTS

### 4.1 General framework

To address the metadata-based multi-task MAB problem, in this section, we introduce a general hierarchical Bayesian framework to characterize the information-sharing structure among tasks, and design a TS algorithm that can continuously learn $\mathbb{P}(\boldsymbol{r}|\mathbf{x})$ and yield a low regret. For ease of exposition, we assume discrete probability spaces. Our discussion is equally applicable to continuous settings. Two representative examples are given in the following subsections.

To learn $\mathbb{P}(\boldsymbol{r}|\mathbf{x})$, we make a model assumption that $\boldsymbol{r}_i$ is sampled from $f(\boldsymbol{r}_i|\mathbf{x}_i, \boldsymbol{\theta})$, for a model $f$ parameterized by the unknown parameter $\boldsymbol{\theta}$. For example, in Gaussian bandits, $f(\boldsymbol{r}_i|\mathbf{x}_i, \boldsymbol{\theta})$ can be

---
**Algorithm 1:** MTTS: Multi-task Thompson Sampling
---
**Input :** Prior $\mathbb{P}(\boldsymbol{\theta})$ and known parameters of the hierarchical model
**1** Set $\mathcal{H} = \{\}$
**2 for** *every decision point j* **do**
**3**     Retrieve the task index $i$
**4**     Compute the posterior for $\boldsymbol{r}_i$ as $\mathbb{P}(\boldsymbol{r}_i|\mathcal{H})$, according to the hierarchical model (1) (see
      Section 4.2 and 4.3 for examples)
**5**     Sample a reward vector $(\tilde{r}_{i,1}, \ldots, \tilde{r}_{i,K})^T \sim \mathbb{P}(\boldsymbol{r}_i|\mathcal{H})$
**6**     Take action $A_j = argmax_{a \in [K]} \tilde{r}_{i,a}$
**7**     Receive reward $R_j$
**8**     Update the dataset as $\mathcal{H} \leftarrow \mathcal{H} \cup \{(A_j, R_j, \mathbf{x}_i, i)\}$
**9 end**
---

the conditional density function of a linear model with Gaussian errors (see Section 4.2). We denote the prior of $\boldsymbol{\theta}$ as $\mathbb{P}(\boldsymbol{\theta})$. Therefore, our full model is the following Bayesian hierarchical model:

$$
\begin{aligned}
\text{(Prior)} \qquad & \boldsymbol{\theta} \quad \sim \mathbb{P}(\boldsymbol{\theta}), \\
\text{(Inter-task)} \quad & \boldsymbol{r}_i|\mathbf{x}_i, \boldsymbol{\theta} \sim f(\boldsymbol{r}_i|\mathbf{x}_i, \boldsymbol{\theta}), \forall i \in [N], \\
\text{(Intra-task)} \quad & R_{i,t} \quad = \sum\nolimits_{a \in [K]} \mathbb{I}(A_{i,t} = a) r_{i,a} + \epsilon_{i,t}, \forall i \in [N], t \in [T_i].
\end{aligned} \tag{1}
$$

We note that $\mathbb{P}(\boldsymbol{\theta})$ is specified by the users, and our regret analysis is a worst-case frequentist regret bound with respect to $\boldsymbol{\theta}$. Besides, the inter-task layer clearly subsumes the meta MAB model $\boldsymbol{r}_i|\boldsymbol{\theta} \sim f(\boldsymbol{r}_i|\boldsymbol{\theta})$ and the CMAB model $\boldsymbol{r}_i = f(\mathbf{x}_i; \boldsymbol{\theta})$ as two special cases.

Suppose until a decision point, we have accumulated a dataset $\mathcal{H} = \{(A_j, R_j, \mathbf{x}_j, i(j))\}_{j=1}^n$, where the data for all tasks have been combined and re-indexed. Here, $i(j)$ is the task index for the $j$th tuple in $\mathcal{H}$, $\mathbf{x}_j = \mathbf{x}_{i(j)}$ is its metadata, $A_j$ is the implemented action and $R_j$ is the observed reward. Suppose we need to make a decision for task $i$ at this time point. To adapt the TS algorithm, one needs to compute the posterior of $\boldsymbol{r}_i$ as $\mathbb{P}(\boldsymbol{r}_i|\mathcal{H})$ according to the specified hierarchical model, sample a reward vector, and then act greedily to it. Based on this framework, we designed the Multi-Task Thompson Sampling (MTTS) algorithm, which is summarized as Algorithm 1.

Algorithm 1 is natural and general. Once a hierarchical model is specified, the remaining step to adapt MTTS is to compute the posterior $\mathbb{P}(\boldsymbol{r}_i|\mathcal{H})$. We will discuss two concrete examples in the following sections. Before we proceed, we remark that the posterior can be written as

$$
\mathbb{P}(\boldsymbol{r}_i|\mathcal{H}) = \int_{\boldsymbol{\theta}} \mathbb{P}(\boldsymbol{\theta}|\mathcal{H}) f(\boldsymbol{r}_i|\boldsymbol{\theta}, \mathcal{H}) d\boldsymbol{\theta} \propto \mathbb{P}(\mathcal{H}_i|\boldsymbol{r}_i) \int_{\boldsymbol{\theta}} \mathbb{P}(\boldsymbol{\theta}|\mathcal{H}) f(\boldsymbol{r}_i|\mathbf{x}_i, \boldsymbol{\theta}) d\boldsymbol{\theta}, \tag{2}
$$

where $\mathcal{H}_i$ is the subset of $\mathcal{H}$ that contains the history of task $i$. Note that $\mathbb{P}(\mathcal{H}_i|\boldsymbol{r}_i)$ is the likelihood for task $i$ alone. Therefore, (2) provides a nice interpretation: up to a normalization constant, $\int_{\boldsymbol{\theta}} \mathbb{P}(\boldsymbol{\theta}|\mathcal{H}) f(\boldsymbol{r}_i|\mathbf{x}_i, \boldsymbol{\theta}) d\boldsymbol{\theta}$ can be regarded as a prior for task $i$, and this prior will be continuously updated by pooling data based on the hierarchical model. Specifically, data from all tasks are utilized to infer $\boldsymbol{\theta}$ via $\mathbb{P}(\boldsymbol{\theta}|\mathcal{H})$, which is then used to infer $\mathbb{P}(\boldsymbol{r}_i|\mathbf{x}_i)$ through $f(\boldsymbol{r}_i|\mathbf{x}_i, \boldsymbol{\theta})$. This is consistent with our motivations, that we would like to utilize the metadata $\{\mathbf{x}_i\}$ to guide exploration (via the prior), while also using the task-specific history $\mathcal{H}_i$ (via the likelihood $\mathbb{P}(\mathcal{H}_i|\boldsymbol{r}_i)$). As a multi-task algorithm, MTTS yields desired interpretability, as one can study how the information from other tasks affects one specific task, via investigating how its metadata determines its prior.

In addition, the relationship (2) suggests a computationally efficient variant of Algorithm 1. We note two facts: (i) it is typically more demanding to compute $\mathbb{P}(\boldsymbol{\theta}|\mathcal{H})$ than $\mathbb{P}(\mathcal{H}_i|\boldsymbol{r}_i) f(\boldsymbol{r}_i|\mathbf{x}_i, \boldsymbol{\theta})$, since the former involves a regression problem and usually has no explicit form, while the latter usually has analytic solutions by choosing conjugate priors; (ii) according to (2), to sample $\boldsymbol{r}_i$ from $\mathbb{P}(\boldsymbol{r}_i|\mathcal{H})$, it is equivalent to first sample one $\tilde{\boldsymbol{\theta}}$ from $\mathbb{P}(\boldsymbol{\theta}|\mathcal{H})$, and then sample $\boldsymbol{r}_i$ with probability proportional to $\mathbb{P}(\mathcal{H}_i|\boldsymbol{r}_i) f(\boldsymbol{r}_i|\mathbf{x}_i, \tilde{\boldsymbol{\theta}})$. Therefore, when it is computationally heavy to sample $\mathbb{P}(\boldsymbol{r}_i|\mathcal{H})$ every time, we can instead sample $\tilde{\boldsymbol{\theta}}$ from $\mathbb{P}(\boldsymbol{\theta}|\mathcal{H})$ at a lower frequency, and apply $TS\big(f(\boldsymbol{r}_i|\mathbf{x}_i, \tilde{\boldsymbol{\theta}})\big)$ to task $i$ before sampling the next value of $\boldsymbol{\theta}$. This variant can be regarded as updating the inter-task data pooling module in a batch mode, and it shows desired regret in our analysis and negligible costs in our experiments. Under the sequential setting, this variant is described in Algorithm 2, and the general form is given in Appendix C.2. See Section 4.3 for a concrete example.

**Algorithm 2:** Computationally Efficient Variant of MTTS under the Sequential Setting

---

  **Input :** Prior $\mathbb{P}(\boldsymbol{\theta})$ and known parameters of the hierarchical model
**1** Set $\mathcal{H} = \{\}$ and $\mathbb{P}(\boldsymbol{\theta}|\mathcal{H}) = \mathbb{P}(\boldsymbol{\theta})$
**2 for** *task* $i \in [N]$ **do**
**3**     Sample one $\tilde{\boldsymbol{\theta}}$ from the current posterior $\mathbb{P}(\boldsymbol{\theta}|\mathcal{H})$
**4**     Apply $TS\big(f(\boldsymbol{r}_i|\mathbf{x}_i, \tilde{\boldsymbol{\theta}})\big)$ to task $i$ for $T_i$ rounds, and collect observed data as $\mathcal{H}_i$
**5**     Update $\mathcal{H} = \mathcal{H} \cup \mathcal{H}_i$
**6**     Update the posterior of $\boldsymbol{\theta}$ according to the hierarchical model (1)
**7 end**

---

### 4.2 Gaussian bandits with Bayesian linear mixed models

In this section, we focus on Gaussian bandits, where the error term $\epsilon_{i,t}$ follows $\mathcal{N}(0, \sigma^2)$ for a known parameter $\sigma$. We will first introduce a linear mixed model to characterize the information-sharing structure, and then derive the corresponding posterior. We use $\boldsymbol{I}_m$ to denote an $m \times m$ identity matrix. Let $\boldsymbol{\phi}(\cdot, \cdot) : \mathbb{R}^p \times [K] \to \mathbb{R}^d$ be a known map from the metadata-action pair to a $d$-dimensional transformed feature vector, and let $\boldsymbol{\Phi}_i = (\boldsymbol{\phi}(\mathbf{x}_i, 1), \ldots, \boldsymbol{\phi}(\mathbf{x}_i, K))^T$. We focus on the case that $\mathbb{E}(\boldsymbol{r}_i|\mathbf{x}_i)$ is linear in $\boldsymbol{\Phi}_i$ and consider the following linear mixed model (LMM):

$$\boldsymbol{r}_i = \boldsymbol{\Phi}_i \boldsymbol{\theta} + \boldsymbol{\delta}_i, \;\; \forall i \in [N] \tag{3}$$

where $\boldsymbol{\theta}$ is an unknown vector, and $\boldsymbol{\delta}_i \overset{i.i.d.}{\sim} \mathcal{N}(\mathbf{0}, \boldsymbol{\Sigma})$ for some known covariance matrix $\boldsymbol{\Sigma}$. In this model, the *fixed effect* term $\boldsymbol{\Phi}_i \boldsymbol{\theta}$ captures the task relations through their metadata, and the *random effect* term $\boldsymbol{\delta}_i$ allows inter-task heterogeneity that can not be captured by the metadata. We adopt the prior distribution $\boldsymbol{\theta} \sim \mathcal{N}(\boldsymbol{\mu}_\theta, \boldsymbol{\Sigma}_\theta)$ with hyper-parameters $\boldsymbol{\mu}_\theta$ and $\boldsymbol{\Sigma}_\theta$.

The posterior of $\boldsymbol{r}_i$ under LMM has an analytic expression. We begin by introducing some notations. Recall that the history is $\mathcal{H} = \{(A_j, R_j, \mathbf{x}_j, i(j))\}_{j=1}^n$ of size $n$. Denote $\boldsymbol{R} = (R_1, \ldots, R_n)^T$ and $\boldsymbol{\Phi} = (\boldsymbol{\phi}(\mathbf{x}_1, A_1), \ldots, \boldsymbol{\phi}(\mathbf{x}_n, A_n))^T$. Let $\boldsymbol{\Sigma}_{a,a'}$ be the $(a, a')$-th entry of $\boldsymbol{\Sigma}$. The LMM induces an $n \times n$ kernel matrix $\boldsymbol{K}$, the $(l, m)$-th entry of which is $\boldsymbol{\phi}^T(\mathbf{x}_l, A_l)\boldsymbol{\Sigma}_\theta \boldsymbol{\phi}(\mathbf{x}_m, A_m) + \boldsymbol{\Sigma}_{A_l, A_m}\mathbb{I}(i(l) = i(m))$. Finally, define a $K \times n$ matrix $\boldsymbol{M}_i$, such that the $(a, j)$-th entry of $\boldsymbol{M}_i$ is $\mathbb{I}(i(j) = i)\boldsymbol{\Sigma}_{A_j, a}$. The posterior of $\boldsymbol{r}_i$ follows a multivariate normal distribution, with mean and covariance as

$$\mathbb{E}(\boldsymbol{r}_i|\mathcal{H}) = \boldsymbol{\Phi}_i \boldsymbol{\mu}_\theta + (\boldsymbol{\Phi}_i \boldsymbol{\Sigma}_\theta \boldsymbol{\Phi}^T + \boldsymbol{M}_i)(\boldsymbol{K} + \sigma^2 \boldsymbol{I}_n)^{-1}(\boldsymbol{R} - \boldsymbol{\Phi}\boldsymbol{\mu}_\theta),$$

$$cov(\boldsymbol{r}_i|\mathcal{H}) = (\boldsymbol{\Phi}_i \boldsymbol{\Sigma}_\theta \boldsymbol{\Phi}_i^T + \boldsymbol{\Sigma}) - (\boldsymbol{\Phi}_i \boldsymbol{\Sigma}_\theta \boldsymbol{\Phi}^T + \boldsymbol{M}_i)(\boldsymbol{K} + \sigma^2 \boldsymbol{I}_n)^{-1}(\boldsymbol{\Phi}_i \boldsymbol{\Sigma}_\theta \boldsymbol{\Phi}^T + \boldsymbol{M}_i)^T.$$

At each decision point, we can then follow Algorithm 1 to act. The computation is dominated by the matrix inverse of $(\boldsymbol{K} + \sigma^2 \boldsymbol{I}_n)$. In Appendix C.1, we introduce an efficient implementation via using the Woodbury matrix identity [50] and the block structure induced by the LMM.

**Remark 1.** *The idea of addressing metadata-based multi-task MAB problems via mixed effect models is generally applicable. One can replace the linear form of* (3) *with other functional forms and proceed similarly. As an example, a mixed-effect Gaussian process model and the corresponding MTTS algorithm are derived in Appendix D.2.*

### 4.3 Bernoulli bandits with Beta-Bernoulli logistic models

The Bernoulli bandits, where the reward is a binary random variable, is another popular MAB problem. As a concrete example, we consider the Beta-Bernoulli logistic model (BBLM) to characterize the information-sharing structure. Recall $r_{i,a}$ is the expected reward of the $a$th arm of task $i$. In BBLM, $r_{i,a}$ follows a Beta distribution with $\mathbb{E}(r_{i,a}|\mathbf{x}_i)$ being a logistic function of $\boldsymbol{\phi}^T(\mathbf{x}_i, a)\boldsymbol{\theta}$. We assume the prior $\boldsymbol{\theta} \sim \mathcal{N}(\boldsymbol{\mu}_\theta, \boldsymbol{\Sigma}_\theta)$ as well. The BBLM is then defined as

$$r_{i,a} \sim \text{Beta}\big(logistic(\boldsymbol{\phi}^T(\mathbf{x}_i, a)\boldsymbol{\theta}), \psi\big), \; \forall a \in [K], i \in [N]$$

$$R_{i,t} \sim \text{Bernoulli}\big(\sum\nolimits_{a \in [K]} \mathbb{I}(A_{i,t} = a)r_{i,a}\big), \forall t \in [T_i], i \in [N], \tag{4}$$

where $logistic(x) \equiv 1/(1 + exp^{-1}(x))$, $\psi$ is a known parameter, and $\text{Beta}(\mu, \psi)$ denotes a Beta distribution with mean $\mu$ and precision $\psi$. In regression analysis, the BBLM is popular for binary responses when there exist multiple levels in the data structure [20, 47].

One challenge to adapting Algorithm 1 to Bernoulli bandits is that the posterior does not have an explicit form. This is also a common challenge to generalized linear bandits [38, 36]. In model (4), the main challenge comes from the Beta logistic model part. We can hence follow the algorithm suggested by (2) to sample $\boldsymbol{\theta}$ at a lower frequency. The nice hierarchical structure of model (4) allows efficient computation of the posterior of $\boldsymbol{\theta}$ via approximate posterior inference algorithms, such as Gibbs sampler [29] or variational inference [9]. For example, with Gibbs sampler, the algorithm will alternate between the first layer of (4), which involves a Beta regression, and its second layer, which yields a Beta-Binomial conjugate form. Both parts are computationally tractable. We defer the details to Appendix C.3 to save space. Finally, we note that, similar models and computation techniques can be developed for other reward variable distributions with a conjugate prior.

## 5 Regret Analysis

In this section, we present the regret bound for MTTS under the linear mixed model introduced in Section 4.2. We will focus on a simplified version of Algorithm 1, due to the technical challenge to analyze the behaviour of TS with a misspecified prior. Indeed, to analyze MTTS, we need a tight bound on the difference in Bayes regret between TS with a misspecified prior and TS with the correct prior. To the best of our knowledge, it is still an open and challenging problem in the literature [7, 34].

We consider two modifications (see Appendix G for a formal description): (i) we introduce an "alignment period" for each task, by pulling each arm once in its first $K$ interactions, and (ii) instead of continuously updating the prior as in (2), we fix down a prior for each task after its alignment period, with the prior learned from data generated during all finished alignment periods. These two modifications are considered so that we can apply our "prior alignment" proof technique, which is inspired by [7] and allows us to bound the multi-task regret. We note similar modifications are imposed in the literature as well, due to similar technical challenges [7, 34]. The former modification causes an additional $O(NK)$ regret, and the latter essentially utilizes less information in constructing the priors than Algorithm 1 and is imposed to simplify the analysis. Even though, the modified version suffices to show the benefits of information sharing. Besides, the second modification is consistent with Algorithm 2 and hence we provide certain guarantees to this variant. Finally, the vanilla MTTS (Algorithm 1) shows desired sublinear regrets in experiments.

For a matrix $\boldsymbol{A}$, we denote its operator norm and minimum singular value by $||\boldsymbol{A}||$ and $\sigma_{min}(\boldsymbol{A})$, respectively. For a vector $\boldsymbol{v}$, we define $||\boldsymbol{v}||$ to be its $\ell_2$ norm. $\tilde{O}$ is the big-$O$ notation that hides logarithmic terms. We make the following regularity assumptions.

**Assumption 1.** $\sigma_{min}\big(\mathbb{E}(\boldsymbol{\Phi}_i^T \boldsymbol{\Phi}_i)\big) \geq Kc_1$ and $max_{i \in [N]}||\boldsymbol{\Phi}_i^T \boldsymbol{\Phi}_i|| \leq KC_1$ for some positive constants $c_1$ and $C_1$.

**Assumption 2.** $max_{a \in [K]}||\phi(\mathbf{x}_i, a)|| \leq C_2$ and $||\boldsymbol{\theta}|| \leq C_3$ for some positive constants $C_2$ and $C_3$.

**Assumption 3.** $||\boldsymbol{\mu_\theta}||$, $||\boldsymbol{\Sigma_\theta}||$ and $||\boldsymbol{\Sigma}||$ are bounded.

These regularity conditions are standard in the literature on Bayesian bandits [52, 37], linear bandits [2, 11] and feature-based structured bandits [48, 41]. Under these conditions, we can derive the following regret bound for the modified MTTS. For simplicity, we assume $T_1 = \cdots = T_N = T$.

**Theorem 1.** *Under Assumptions $1 - 3$, when $d = o(N)$ and $K < min(N, T)$, the multi-task regret of the modified MTTS under the LMM is bounded as*

$$MTR(N, \{T_i\}) = O\big(\sqrt{Nlog(NT)}(\sqrt{d} + \sqrt{log(NT)})\sqrt{TKlogT} + log^2(NT)\sqrt{(T-K)KlogT} + NK\big)$$
$$= \tilde{O}\big(\sqrt{Nd}\sqrt{TK} + NK\big). \tag{5}$$

We remark this is a worst-case frequentist regret bound with respect to $\boldsymbol{\theta}$. To save space, the proof of Theorem 1 is deferred to Appendix G. Note that the single-task Bayes regret of oracle-TS is $O(\sqrt{TKlogT})$ based on the literature (e.g., [37]). The Bayes regret of the modified MTTS then follows from the definition of multi-task regret.

**Corollary 1.** *Under the same conditions as Theorem 1, the Bayes regret of the modified MTTS can be bounded as $BR(N, \{T_i\}) = \tilde{O}\big(\sqrt{Nd}\sqrt{TK} + NK + N\sqrt{TK}\big)$.*

**Discussions.**     For comparison purposes, we informally summarize the regret bounds of several baselines in Table 5. To save space, the formal statements and their proofs are deferred to Appendix

**Table 1:** A summary of the multi-task regret bounds under the LMM. All logarithmic terms are hidden.

|  | MTTS (the proposed) | meta-TS | individual-TS | linear-TS | OSFA |
|---|---|---|---|---|---|
| Model | $\boldsymbol{r}_i \sim f(\boldsymbol{r}_i|\mathbf{x}_i; \boldsymbol{\theta})$ | $\boldsymbol{r}_i \sim f(\boldsymbol{r}|\boldsymbol{\theta})$ | $\boldsymbol{r}_i \sim \mathbb{P}(\boldsymbol{r})$ | $\boldsymbol{r}_i = f(\mathbf{x}_i; \boldsymbol{\theta})$ | $\boldsymbol{r}_i \equiv \boldsymbol{r} \sim \mathbb{P}(\boldsymbol{r})$ |
| MTR | $\sqrt{N}d\sqrt{TK} + NK$ | $N\sqrt{TK}$ | $N\sqrt{TK}$ | $NT$ | $NT$ |

H. We consider the following algorithms: (i) One TS policy for all tasks, referred to as *one-size-fit-all (OSFA)*; (ii) An individual TS policy for each task, referred to as *individual-TS*; (iii) TS for linear bandits [2], referred to as *linear-TS*; (iv) Meta-Thompson sampling (meta-TS) proposed in [34]. Specifically, meta-TS, under the Gaussian bandits setting, assumes $\boldsymbol{r}_i \sim \mathcal{N}(\boldsymbol{\mu}, \boldsymbol{\Sigma})$ for some unknown vector $\boldsymbol{\mu}$, and it pools data to learn $\boldsymbol{\mu}$. To recap, OSFA ignores the inter-task heterogeneity and linear-TS assumes $\boldsymbol{r}_i$ is fully determined by the metadata, while individual-TS does not share information and meta-TS fails to utilize the metadata to efficiently share information.

In the MTR bound for MTTS (5), the first term is sublinear in $N$, which implies MTTS is continuously learning about $\mathbb{P}(\boldsymbol{r}_i|\mathbf{x}_i)$ and will behave closer to oracle-TS as $N$ increases. The $\tilde{O}(NK)$ term is due to the imposed $K$ alignment rounds in our analysis. In contrast to MTTS, the multi-task regrets of these baseline algorithms unavoidably scale linearly in $N$, since they fail to utilize knowledge about $\mathbb{P}(\boldsymbol{r}_i|\mathbf{x}_i)$, which can efficiently guide the exploration. The regret rates suggest that MTTS is particularly valuable in task-rich settings ($N \gg d$). When $d$ is relatively large compared with $N$, certain additional structures (e.g., sparsity) can be considered. We note that, although their difference in multi-task regret is of the same order as the Bayes regret of oracle-TS, which implies the Bayes regret of MTTS, meta-TS, and individual-TS will be of the same order, the hidden multiplicative factors can differ a lot. We observe a substantial gap in the experiments. Finally, OSFA and linear-TS even suffer a linear regret in $T$, due to the bias caused by ignoring the heterogeneity. In contrast, MTTS yields a desired sublinear Bayes regret in $T$.

An $\tilde{O}(T^2\epsilon)$ bound on the additional regret for Bayesian bandit algorithms with a misspecified prior is provided in [56] under a simple setting, where $\epsilon$ is the total-variation between the true prior and the misspecified one. [34] adopts a recursive approach to obtain a $\tilde{O}(\sqrt{N}KT^2 + N\sqrt{KT})$ Bayes regret bound for meta-TS. Our bound demonstrates the usefulness of the prior alignment proof technique, which can be applied to meta-TS as well. Moreover, we note that the LMM can be reduced to a linear bandits with an extended context $\tilde{\phi}^T(\mathbf{x}_i, a) = (\mathbf{1}_{i,a}^T, \phi^T(\mathbf{x}_i, a))^T$, where $\mathbf{1}_{i,a}$ is a length-$NK$ vector taking value 1 at the $((i-1)N + a)$-th entry. Such a reduction is typically ill-conditioned and a direct application of results for linear bandits [52] gives us a Bayes regret bound $\tilde{O}((NK + d)\sqrt{NT})$, which reflects the necessity to carefully utilize the information sharing structure. Besides, we note that the reduction from a hierarchical model to a contextual bandit model is typically not possible. Finally, similar to some literature on meta bandits [65, 7], we adopt the *task distribution* viewpoint by assuming the tasks (and hence $\{\mathbf{x}_i\}_{i=1}^N$) are i.i.d. and considering Assumption 1. It should be feasible to relax these assumptions following the standard proof approach with adversarial contexts [2, 15, 5]. See Appendix G for more details.

## 6 Experiments

### 6.1 Synthetic Experiments

**Setting.** We conduct simulation experiments to support our theoretical results and study the empirical performance of MTTS. We use model (3) and (4) as our data generation model for Gaussian bandits and Bernoulli bandits, respectively. In the main text, we present results with $N = 200$, $T = 200$, $K = 8$ and $d = 15$. We set $\phi(\mathbf{x}_i, a) = (\mathbf{1}_a^T, \tilde{\phi}_{i,a}^T)^T$, where $\mathbf{1}_a$ is a length-$K$ indicator vector taking value 1 at the $a$-th entry, and $\tilde{\phi}_{i,a}$ is sampled from $\mathcal{N}(\mathbf{0}, \boldsymbol{I}_{d-K})$. The coefficient vector $\boldsymbol{\theta}$ is sampled from $\mathcal{N}(\mathbf{0}, d^{-1}\boldsymbol{I}_d)$. For Gaussian bandits, we set $\sigma = 1$ and $\boldsymbol{\Sigma} = \sigma_1^2 \boldsymbol{I}_K$, for different values of $\sigma_1$. For Bernoulli bandits, we vary the precision $\psi$. A higher value of $\sigma_1$ or $\psi$ implies a larger inter-task heterogeneity conditional on the metadata. We consider two representative types of task sequence: (i) the *sequential* setting, where the $(i+1)$-th task begins after we finish the $i$-th task, and (ii) the *concurrent* setting, where we interact with all tasks for the $t$-th time simultaneously.

For Gaussian bandits, we compare MTTS with the four algorithms discussed in Section 5: OSFA, individual-TS, linear-TS, and meta-TS. Besides, we also investigate the performance of the com-

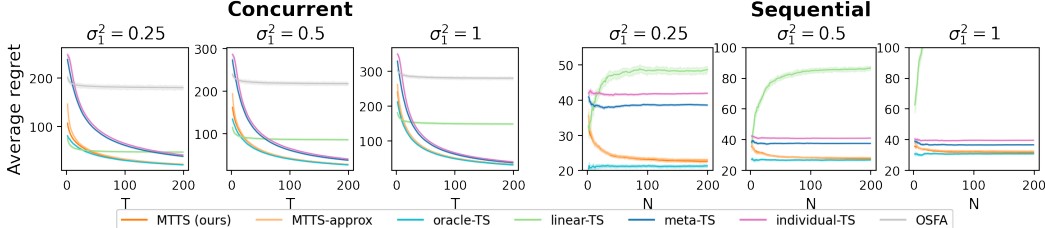

**(a)** Gaussian bandits. The regrets of OSFA are much higher in some subplots and hence hidden.

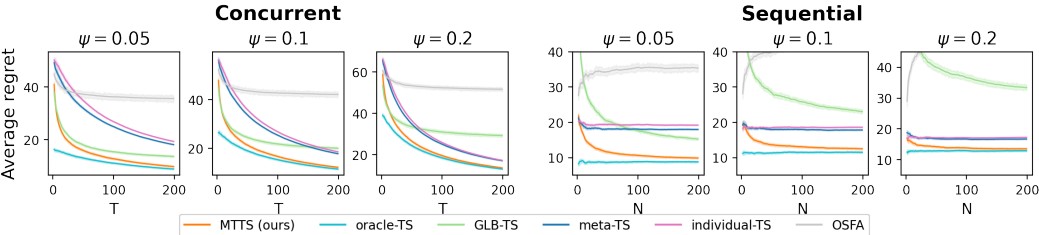

**(b)** Bernoulli bandits. A larger value of $\psi$ implies a larger variation of $r_i$ condition on $\mathbf{x}_i$.

**Figure 2:** Average Bayes regret for various methods. Shared areas indicate the standard errors of the averages.

putationally efficient variant of MTTS in Algorithm 2, referred to as MTTS-approx. For Bernoulli bandits, we replace linear-TS with a TS algorithm for generalized linear bandits [36], which is referred to as GLB-TS. The other baselines are also replaced with their Bernoulli variants. To set the priors for these TS algorithms, we apply the law of total expectation (variance) to integrate out $(\mathbf{x}, \boldsymbol{r})$ to specify the mean (variance) term. See Appendix F for more details about the implementations.

**Results.** Results aggregated over 100 random seeds are presented in Figure 2. Under the concurrent setting, the average Bayes regrets per interaction are displayed, which shows the trend with $T$; under the sequential setting, the average Bayes regrets per task are displayed, which shows the trend with $N$. Such a choice allows us to see the regret rates derived in Section 5 more clearly. Recall that the multi-task regret of an algorithm can be calculated as the difference between the Bayes regret of itself and oracle-TS. For ease of comparison, we also report the multi-task regrets in Appendix E.2.

Overall, MTTS performs favorably compared with the other baselines. Our findings can be summarized as follows. First, in the sequential setting, the Bayes regret of MTTS approaches that of oracle-TS, which shows that MTTS yields a sublinear multi-task regret in $N$. This implies MTTS is learning $\mathbb{P}(\boldsymbol{r}|\mathbf{x})$ and will eventually behave similarly to oracle-TS. In contrast, there exists a consistent and significant performance gap between oracle-TS and the other baselines. This demonstrates the benefit of metadata-based information sharing. Second, in the concurrent setting, MTTS shows a sublinear Bayes regret in $T$. This implies MTTS is learning the best arm for each task. Although individual-TS and meta-TS also yield sublinear trends, their cumulative regrets can be significantly higher. For example, in Gaussian bandits, the cumulative regret of MTTS is only $53.3\%$ of the regret for individual-TS when $\sigma_1^2 = 0.25$, and $68.2\%$ when $\sigma_1^2 = 0.5$. This demonstrates that, with information sharing and the guidance of metadata, MTTS can explore more efficiently. The benefit is particularly important when the horizon $T$ is not long, and MTTS enables a jump-start for each task. Third, the trend with $\sigma_1^2$ and $\psi$ is clear: when the metadata is less useful ($\sigma_1^2 = 1$ or $\psi = 0.2$), the performance of meta-TS and individual-TS becomes closer to MTTS, while linear-TS (GLB-TS) and OSFA suffer from the bias; when the metadata is highly informative, the advantage of MTTS over meta-TS and individual-TS becomes much more significant, and the performance of linear-TS (GLB-TS) becomes relatively better. Overall, MTTS shows robustness. Lastly, MTTS-approx yields similar performance with the vanilla MTTS, except when $N$ is still small.

**Additional results.** Results under other settings are reported in Appendix E.3, where MTTS consistently performs favorably, and shows reasonable trends with hyperparameters such as $N$, $K$, and $d$. Finally, recall that, to share information, we made a model assumption that $\boldsymbol{r}_i|\mathbf{x}_i, \boldsymbol{\theta} \sim f(\boldsymbol{r}_i|\mathbf{x}_i, \boldsymbol{\theta})$. When this model is correctly specified, MTTS has shown superior theoretical and numerical performance. Intuitively, this model is designed to pool information to provide an informative prior. As long as the learned prior provides reasonable information relative to a manually specified one, the performance of MTTS is expected to be better or at least comparable. The con-

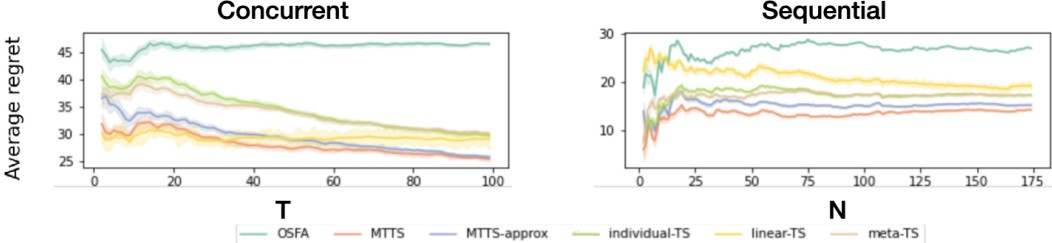

**Figure 3:** MovieLens: average regret for various methods. Shared areas indicate the standard errors of the averages.

jecture is partially supported by Lemma 5 in the appendix, which characterizes the impact of a misspecified prior on MTR. We empirically investigate the impact of model misspecication to MTTS in Appendix E.1, where MTTS demonstrates great robustness.

### 6.2 MovieLens Experiments

We evaluate the empirical performance on the MovieLens 1M dataset [24], which consists of the ratings (reward) by 6040 users (tasks) on 3900 movies. Each user has three features (metadata), including gender, age, and occupation. The movies belong to a few genres (arms). We preprocessed the dataset to focus on the top 5 genres, top 5 occupations and users with at least 500 data points, which gives us $N = 175$ users. We apply Gaussian bandit algorithms considered in Section 6.1 to recommend movie genres to each user and collect rewards by sampling corresponding records from the dataset. We repeat the experiment over 100 random seeds, with $T = 100$.

The regrets against the empirical oracle (which knows the user-specific optimal genre with the highest empirical mean reward *a priori*) are reported in Figure 3. Overall, MTTS performs preferably and demonstrate the benefits of metadata-based information sharing, compared with individual-TS and meta-TS. Linear-TS and OSFA seem to suffer a linear regret in $T$, which emphasizes the existence of heterogeneity. As expected, the demographical features of an user can partially predict her interests in movie genres, while there still exists strong heterogeneity even conditional on these features, which requires further personalization based on her own feedbacks.

## 7 Discussion

In this paper, we study the metadata-based multi-task MAB problem, where we can utilize the metadata to capture and learn task relations, so as to judiciously share knowledge. We formalize this problem under a general framework and define the notion of regret. We propose to address this problem with Bayesian hierarchical models and design a TS-type algorithm, MTTS, to share information and minimize regrets. Two concrete examples are analyzed and efficient implementations are discussed. The usefulness of MTTS are further supported by regret bounds as well as experiment results with both synthetic and real data.

The proposed framework can be extended in several aspects. First, the variance components are assumed known for the two examples in Section 4. In practice, we can apply empirical Bayes to update these hyperparameters adaptively. See Appendix D.3 for details. Second, our work can be extended to contextual bandits, based on the idea of clustering of bandits [22, 40] when the set of task instances is finite, or the idea of contextual Markov decision process [45] when the set is infinite. See Appendix D.4 for details. Third, while our analysis for MTTS can be directly extended to other functional forms, the prior alignment technique does utilize some properties of Gaussian distributions. A novel analysis of TS with mis-specified priors would be of independent interest. Besides, to our knowledge, the lower bound for the additional regret with a misspecified prior is still missing in the literature. Such a bound is the difference between two Bayes regrets and hence requires a quite precise description of the behaviors of Bayesian bandit algorithms. Lastly, we focus on TS-type algorithms in this paper since it is straightforward to incorporate side information via the prior. The Upper-Confidence Bound [37] (UCB) is another popular algorithm framework. Though directly adapting UCB to our setup is challenging, Bayesian UCB [31] naturally fits our problem and benefits from information sharing. We believe this is an interesting direction worthy of study.

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
