# Supplement to "Metadata-based Multi-Task Bandits with Bayesian Hierarchical Models"

## A  Review of Statistical Concepts

In this section, we first review several classic statistical concepts, including the hierarchical models, random effect model, fixed effect model, and mixed effect model. They are all general concepts in statistics and have been integrated with different models, and we may focus on their most related forms for ease of exposition. See [11, 42] for more detailed discussions. Next, we discuss their connection with our model and baselines mentioned in the main text.

### A.1  Statistical concepts

Consider a supervised learning problem, where we have $N$ subjects. For each subject $i$, we have $M_i$ measurements, denoted as $\{Y_{i,t}\}_{t=1}^{M_i}$. In a classic example, each subject corresponds to a hospital and each measurement corresponds to the outcome of a patient treated in one of these hospitals.

A random effect model assumes the data of subject $i$ is generated from some distributions $f_i(\cdot; \boldsymbol{\theta}_i)$ parameterized by $\boldsymbol{\theta}_i$, and these coefficients are sampled from an upper-level distribution as $\boldsymbol{\theta}_i \sim \mathcal{P}$. The coefficients and the corresponding distribution are hence treated as random variables, which enables us to characterize the heterogeneity between subjects. Therefore, this class of models are referred to as random effect models.

In contrast, in a fixed effect model, for each measurement $Y_{i,t}$, we additionally have some independent variables $X_{i,t}$, and they are related through a regression model $Y_{i,t} = f(X_{i,t}; \boldsymbol{\theta}) + \epsilon_{it}$, which is parameterized by an unknown parameter $\boldsymbol{\theta}$. Here, $\epsilon_{it}$ is the error term. The coefficient $\boldsymbol{\theta}$ is fixed across subjects, and hence this class of models are referred to as fixed effect models. It is mainly used to characterize the common structure across subjects.

A mixed effect model is a combination of both: it includes both fixed effect terms and random effect terms. The random intercept model $Y_{i,j} = f(X_{i,j}; \boldsymbol{\theta}) + \delta_i + \epsilon_{it}$ with $\delta_i \sim \mathcal{P}$ and the random coefficient model $Y_{i,j} = f(X_{i,j}; \boldsymbol{\theta} + \boldsymbol{\theta}_i) + \epsilon_{i,t}$ with $\boldsymbol{\theta}_i \sim \mathcal{P}$ are two examples.

Finally, these three models are all special case of the following hierarchical model (a.k.a. multilevel models):

$$\boldsymbol{\theta}_i \sim g(Z_i; \boldsymbol{\theta}),$$
$$Y_{i,t} \sim f(X_{i,t}, \boldsymbol{\theta}_i),$$

where $Z_i$ contains some (optional) static features for each task $i$, the first layer describes the relationship between subjects and the second layer describes the relationship between measurements for the same subject.

### A.2  Relationship with bandits

The aforementioned statistical concepts are typically introduced for supervised learning. We can naturally extend them to the decision-making setup, and connect them with the bandit problem.

We first recall the equivalent definition of the stochastic multi-armed bandit problem that, in the $t$th interaction with task $i$, the rewards for all arms are generated by $\mathbf{R}_{i,t} \sim \mathcal{N}(\boldsymbol{r}_i, \sigma^2 \boldsymbol{I})$, and after taking an action $A_{i,t}$, the $A_{i,t}$-th entry of $\mathbf{R}_{i,t}$ will be observed as $R_{i,t}$.

The meta bandit assumes that each task instance $\boldsymbol{r}_i$ is sampled from some task prior distribution $\mathcal{P}_{\boldsymbol{r}}$, and then the rewards are generated from $\boldsymbol{r}_i$. Therefore, at a high-level, the underlying model can be regarded as a random effect model. As a concrete example, we consider the Gaussian bandit case in Kveton et al. [22]. Let mean rewards of task $i$ be $\boldsymbol{r}_i = (r_{i,1}, \dots, r_{i,K})^T$. At each time point $t$, the rewards are generated by $\mathbf{R}_{i,t} \sim \mathcal{N}(\boldsymbol{r}_i, \sigma^2 \boldsymbol{I})$. We assume the task instances are sampled by $\boldsymbol{r}_i = \boldsymbol{\theta}_i \sim N(\boldsymbol{\mu}, \sigma_0^2 \boldsymbol{I})$. It is easy to see this model is a random effect model.

For contextual MAB, the reward $R_t$ are often related with the context $S_t$ and the action $A_t$ by a model $R_t \sim f(S_t, A_t; \boldsymbol{\theta}) + \boldsymbol{\epsilon}_{i,t}$ parameterized by $\boldsymbol{\theta}$. In our multi-task MAB setting, to adapt these algorithms, we can similarly assume $\mathbf{R}_{i,t} = f(S_{i,t}; \boldsymbol{\theta}) + \boldsymbol{\epsilon}_{i,t}$, where $S_{i,t} \equiv X_i$ for all $t$. It is straightforward to check that these two models are both special cases of the fixed effect model.

Finally, by setting $Z_i = \mathbf{x}_i$, $\boldsymbol{\theta}_i = \boldsymbol{r}_i$, $Y_{i,t} = R_{i,t}$ and $X_{i,t} = A_{i,t}$, it is easy to see our model (1) is a hierarchical model, with the other two models as special cases. In particular, the LMM in Section 4.2 is a mixed effect model.

# B  Additional Related Work

In this section, we compare this paper with additional related works. To begin with, we note that there are several terms used in the literature for problems concerned with multiple tasks, which is often a source of confusion. Based on [17], *meta learning* assumes tasks are drawn from a distribution and aims to maximize the average performance over this task distribution (*meta-objective*); *transfer learning* uses data from finished tasks to improve the performance on a new task, and typically focuses on the single-task objective, although the viewpoint of meta learning can be applied to achieve this goal (as in the sequential setting of our work); *multi-task learning* aims to jointly learn a few tasks, where typically the number of tasks is fixed and the meta-objective is not adopted, although the viewpoint of meta learning can be applied in these problems as well (as in the concurrent setting of our work). There are often some overlaps between these concepts. In this work, we adopt the task distribution viewpoint as well as the meta-objective [17], which are naturally related with our hierarchical Bayesian framework, allows us to model heterogeneity, and enables knowledge sharing via constructing informative priors. We refer to our problem as multi-task bandits to focus on the fact that there are multiple tasks, since our methodology is applicable to the sequential setting (transfer learning), the concurrent setting (multi-task learning), and even more complex settings.

According to the taxonomy developed in [46], multi-task algorithms can be classified into the feature learning approach, low-rank approach, task clustering approach, task relation learning approach, and decomposition approach. In the bandit literature, [44, 7, 30] consider that one interacts with multiple linear bandit tasks concurrently and the coefficient vectors of these tasks either share the same low-dimensional space or the same sparsity pattern, and hence belongs to the feature learning approach; [23] studies finding the maximum entry of a low-rank matrix in a bandit manner and [20] assumes the coefficient matrix of a bilinear bandit problem is low-rank, and hence they belong to the low-rank approach; clustering of bandits [12, 26] and latent bandits [15, 16] assume there is a perfect clustered structure, and hence they belong to the task clustering approach. In contrast to these papers, our work belongs to the relation learning approach, which aims to learn the task relations from data by assuming some probabilistic model or applying some penalty function. Our approach provides nice interpretability and is flexible with arguably less strict assumptions. In particular, to the best of our knowledge, this is the *first* work that can leverage the task-specific metadata in multi-task bandits, which is an important information source that none of the existing methods can utilize. Although we focus on MAB in the main text, the idea of utilizing metadata is generally applicable. We discuss its extensions to linear bandits and clustering of bandits in Appendix D.4.

Besides, similar with meta MAB, there are also several works on meta linear bandits. Meta linear bandits studies the problem where the coefficients of multiple linear bandit tasks are close [36] or are drawn from one prior distribution [3, 38, 6]. Therefore, these works focus on a different problem from ours and they also did not model the task relations with metadata. In addition, [31] proposes a sequential strategy for meta bandits, but as noted in [18], no efficient algorithm has been proposed.

85 Lastly, we note that there are two papers [9, 21] applying multi-task learning to share information
86 across multiple arms of a single task, and hence they have a totally different focus from ours.

87 Our bandit problem is certainly related to Reinforcement Learning (RL). In recent years, there is
88 a surge of interest in multi-task RL, transfer RL, and meta RL. See [47] and [40] for some recent
89 surveys. Among the existing works, [41, 25, 13] consider applying Bayesian hierarchical model
90 to multi-task/meta RL. However, none of these works can utilize the metadata, which is an impor-
91 tant and ubiquitous information source. In the concurrent work [37], the authors, for the first time,
92 consider using metadata in multi-task reinforcement learning to infer the appropriate state represen-
93 tation for each task. They consider a finite set of state encoders, design a special-purpose neural
94 network, and train the whole pipeline end-to-end. In contrast, we study MAB, allow infinite number
95 of task instances, aim to learn the average reward of each task instead of state representation, design
96 a unified Bayesian Hierarchical model framework for this problem, and establish theoretical guar-
97 antees. Therefore, although both work consider similar types of side information (i.e., metadata),
98 the problem and methodology are fundamentally different.

99 Finally, we would like to further compare our setup with CMAB, and discuss the limitations of
100 adopting CMAB in our problem. In the standard contextual MAB setup, we assume $\mathbb{E}(R_{i,t}|S_{i,t} =$
101 $s, A_{i,t} = a) = g(s, a; \gamma)$ for some function $g$, where $S_{i,t}$ is the context. Therefore, it already allows
102 the randomness of $R_{i,t}$. In our setup, by regarding the metadata as "contexts" and neglecting the
103 task identities, the relationship is equal to $\mathbb{E}(R_{i,t}|\mathbf{x}_i = \mathbf{x}, A_{i,t} = a) = g(\mathbf{x}, a; \gamma)$, where we note
104 that the expectation is taken over all tasks with metadata $\mathbf{x}$. This implies that for all tasks with the
105 same metadata, the decision rule will be always the same, no matter what their action-reward history
106 is. Instead, our setup adds another layer to allow variations of $\boldsymbol{r}_i$ conditional on $\mathbf{x}_i$. In another word,
107 we utilize the predictive power of $\mathbf{x}_i$ but does not assume $\boldsymbol{r}_i$ can be fully determined by it.

# C  Implementation

109 In this section, we discuss efficient ways to implement MTTS, and analyze its computational com-
110 plexity. We use $\boldsymbol{I}$ to denote identity matrix, the dimension of which can be inferred from the context.

## C.1  Efficient Implementation for Gaussian bandits with LMM

112 At each decision point, the computation will be dominated by the calculation of the matrix inverse
113 for $(\boldsymbol{K} + \sigma^2 \boldsymbol{I})$, which suffers from a cubic complexity $O(n^3)$ as well-known in the Gaussian process
114 literature [35]. To alleviate the computational burden, we utilize the block structure induced by the
115 mixed effect model via using the well-known Woodbury matrix identity [32], which is reviewed as
116 follows.

117 **Lemma 1** (Woodbury matrix identity). *For any matrix $W$ and any invertible matrices $Z$, $U$, and $V$*
118 *with appropriate dimensions, the following relationship holds:*

$$(\boldsymbol{Z} + \boldsymbol{U}\boldsymbol{W}\boldsymbol{V}^T)^{-1} = \boldsymbol{Z}^{-1} - \boldsymbol{Z}^{-1}\boldsymbol{U}(\boldsymbol{W}^{-1} + \boldsymbol{V}^T\boldsymbol{Z}^{-1}\boldsymbol{U})^{-1}\boldsymbol{V}^T\boldsymbol{Z}^{-1}$$

119 We start by writting $\boldsymbol{K} = \boldsymbol{\Phi}\boldsymbol{\Sigma_\theta}\boldsymbol{\Phi}^T + \boldsymbol{J}$. Without loss of generality, we can rearrange the tuples in
120 $\mathcal{D}$ so that $\boldsymbol{J}$ becomes a block diagonal matrix as $\boldsymbol{J} = diag(\boldsymbol{J}_1, \ldots, \boldsymbol{J}_N)$, where $\boldsymbol{J}_i$ is the submatrix
121 for task $i$. The dimension of $\boldsymbol{J}_i$ equals to $\#\{(A_j, R_j, \mathbf{x}_{i(j)}, i(j)) \in \mathcal{D} : i(j) = i\}$ and it is a pairwise
122 kernel matrix induced by the kernel $\tilde{\mathcal{K}}(O, O') = \boldsymbol{\Sigma}_{a,a'}$. Therefore, we have

$$(\boldsymbol{K} + \sigma^2 \boldsymbol{I})^{-1} = (\boldsymbol{\Phi}\boldsymbol{\Sigma_\theta}\boldsymbol{\Phi}^T + (\boldsymbol{J} + \sigma^2 \boldsymbol{I}))^{-1}$$
$$= (\boldsymbol{J} + \sigma^2 \boldsymbol{I})^{-1}\Big[\boldsymbol{I} - \boldsymbol{\Phi}(\boldsymbol{\Sigma_\theta}^{-1} + \boldsymbol{\Phi}^T(\boldsymbol{J} + \sigma^2 \boldsymbol{I})^{-1}\boldsymbol{\Phi})^{-1}\boldsymbol{\Phi}^T(\boldsymbol{J} + \sigma^2 \boldsymbol{I})^{-1}\Big],$$

123 the computation cost of which will be dominated by $(\boldsymbol{J} + \sigma^2 \boldsymbol{I})^{-1}$, which yields a block diagonal
124 structure and hence the matrix inverse can be more efficiently computed. Suppose we already have
125 $t_0$ interactions with each of $N$ tasks. The computational cost is reduced from $O(N^3 t_0^3)$ to $O(N t_0^3)$.
126 In addition, when $\boldsymbol{\Sigma}$ is diagonal, we can further apply the Woodbury matrix identity to compute
127 $(\boldsymbol{J}_i + \sigma^2 \boldsymbol{I})^{-1}$.

128 Furthermore, to make the computation more efficient, we define $\boldsymbol{\Sigma}_{in} = (\boldsymbol{\Sigma_\theta}^{-1} + \boldsymbol{\Phi}^T(\boldsymbol{J} +$
129 $\sigma^2 \boldsymbol{I})^{-1}\boldsymbol{\Phi})^{-1}, \boldsymbol{J}_{\boldsymbol{\Phi}} = (\boldsymbol{J} + \sigma^2 \boldsymbol{I})^{-1}\boldsymbol{\Phi}, \tilde{\boldsymbol{R}} = (\boldsymbol{R} - \boldsymbol{\Phi}\boldsymbol{\mu_\theta}), \boldsymbol{J}_{\boldsymbol{R}} = (\boldsymbol{J} + \sigma^2 \boldsymbol{I})^{-1}\tilde{\boldsymbol{R}}$. Next, we de-

130 fine

$$\boldsymbol{J}_{\boldsymbol{\Phi},\boldsymbol{R}} = \boldsymbol{\Phi}^T(\boldsymbol{J}+\sigma^2\boldsymbol{I})^{-1}\tilde{\boldsymbol{R}}$$
$$\boldsymbol{J}_{\boldsymbol{\Phi},\boldsymbol{\Phi}} = \boldsymbol{\Phi}^T(\boldsymbol{J}+\sigma^2\boldsymbol{I})^{-1}\boldsymbol{\Phi}$$
$$\boldsymbol{J}_{\boldsymbol{\Phi},M,i} = \boldsymbol{M}_i(\boldsymbol{J}+\sigma^2\boldsymbol{I})^{-1}\boldsymbol{\Phi}$$
$$\boldsymbol{J}_{\boldsymbol{R},M,i} = \boldsymbol{M}_i(\boldsymbol{J}+\sigma^2\boldsymbol{I})^{-1}\tilde{\boldsymbol{R}}$$
$$\boldsymbol{J}_{M,i} = \boldsymbol{M}_i(\boldsymbol{J}+\sigma^2\boldsymbol{I})^{-1}\boldsymbol{M}_i^T.$$

131 For these components, we can efficiently utilize the block structure to only update the corresponding
132 part. Finally, we have

$$\mathbb{E}(\boldsymbol{r}_i|\mathcal{D}) = \boldsymbol{\Phi}_i\boldsymbol{\mu}_{\boldsymbol{\theta}} + \boldsymbol{\Phi}_i\boldsymbol{\Sigma}_{\boldsymbol{\theta}}\boldsymbol{J}_{\boldsymbol{\Phi},\boldsymbol{R}} + \boldsymbol{J}_{\boldsymbol{R},M,i} - \boldsymbol{\Phi}_i\boldsymbol{\Sigma}_{\boldsymbol{\theta}}\boldsymbol{J}_{\boldsymbol{\Phi},\boldsymbol{\Phi}}\boldsymbol{\Sigma}_{in}\boldsymbol{J}_{\boldsymbol{\Phi},\boldsymbol{R}} - \boldsymbol{J}_{\boldsymbol{\Phi},M,i}\boldsymbol{\Sigma}_{in}\boldsymbol{J}_{\boldsymbol{\Phi},\boldsymbol{R}},$$
$$cov(\boldsymbol{r}_i|\mathcal{D}) = (\boldsymbol{\Phi}_i\boldsymbol{\Sigma}_{\boldsymbol{\theta}}\boldsymbol{\Phi}_i^T + \boldsymbol{\Sigma}) - \boldsymbol{J}_{M,i} - (\boldsymbol{\Phi}_i\boldsymbol{\Sigma}_{\boldsymbol{\theta}})\boldsymbol{J}_{\boldsymbol{\Phi},\boldsymbol{\Phi}}(\boldsymbol{\Phi}_i\boldsymbol{\Sigma}_{\boldsymbol{\theta}})^T - \boldsymbol{\Phi}_i\boldsymbol{\Sigma}_{\boldsymbol{\theta}}\boldsymbol{J}_{\boldsymbol{\Phi},M,i}^T - (\boldsymbol{\Phi}_i\boldsymbol{\Sigma}_{\boldsymbol{\theta}}\boldsymbol{J}_{\boldsymbol{\Phi},M,i}^T)^T$$
$$+ (\boldsymbol{\Phi}_i\boldsymbol{\Sigma}_{\boldsymbol{\theta}}\boldsymbol{J}_{\boldsymbol{\Phi},\boldsymbol{\Phi}} + \boldsymbol{J}_{\boldsymbol{\Phi},M,i})\boldsymbol{\Sigma}_{in}(\boldsymbol{\Phi}_i\boldsymbol{\Sigma}_{\boldsymbol{\theta}}\boldsymbol{J}_{\boldsymbol{\Phi},\boldsymbol{\Phi}} + \boldsymbol{J}_{\boldsymbol{\Phi},M,i})^T$$

133 **Alternative implementation.** For the $\boldsymbol{\theta}$-centered sampling scheme (e.g., Algorithm 3), note that

$$\boldsymbol{\theta}|\mathcal{H} \sim \mathcal{N}\Big(\tilde{\boldsymbol{\Sigma}}(\boldsymbol{\Phi}^T\boldsymbol{V}^{-1}\boldsymbol{R} + \boldsymbol{\Sigma}_{\boldsymbol{\theta}}^{-1}\boldsymbol{\mu}_{\boldsymbol{\theta}}), \tilde{\boldsymbol{\Sigma}}\Big),$$
$$\tilde{\boldsymbol{\Sigma}} = (\boldsymbol{\Phi}^T\boldsymbol{V}^{-1}\boldsymbol{\Phi} + \boldsymbol{\Sigma}_{\boldsymbol{\theta}}^{-1})^{-1}, \boldsymbol{V} = \sigma^2\boldsymbol{I}_n + \boldsymbol{J},$$

134 where the dominating step is still to compute $\boldsymbol{V}^{-1}$, and similar tricks can be applied. As a special
135 case, consider $\boldsymbol{\Sigma} = \sigma_1^2\boldsymbol{I}$. We rearrange $\boldsymbol{J}$ as $diag(\boldsymbol{J}_{1,1}, \ldots, \boldsymbol{J}_{1,K}, \ldots, \boldsymbol{J}_{N,1}, \ldots, \boldsymbol{J}_{N,K})$, where
136 $\boldsymbol{J}_{i,a} = \sigma_1^2\boldsymbol{11}^T$, and apply similar rearrangement to $\boldsymbol{\Phi}$ and $\boldsymbol{R}$. Hence we have $\boldsymbol{V}^{-1} = (\sigma^2\boldsymbol{I}_n +$
137 $\boldsymbol{J})^{-1} = diag((\sigma^2\boldsymbol{I} + \boldsymbol{J}_{1,1})^{-1}, \ldots, (\sigma^2\boldsymbol{I} + \boldsymbol{J}_{N,K})^{-1})$. According to the Woodbury matrix identity,
138 we have $(\sigma^2\boldsymbol{I} + \boldsymbol{J}_{i,a})^{-1} = \sigma^{-2}\boldsymbol{I} - \sigma^{-4}(\sigma_1^{-2} + n_{i,a}\sigma^{-2})^{-1}\boldsymbol{11}^T$, where $n_{i,a}$ is the count that action
139 $a$ is implemented for task $i$.

140 After we sample one $\tilde{\boldsymbol{\theta}}$, the prior of $\boldsymbol{r}_i$ can then be updated as $\mathcal{N}(\boldsymbol{\Phi}_i\tilde{\boldsymbol{\theta}}, \boldsymbol{\Sigma})$. Its posterior then follows
141 from the standard normal-normal conjugate relationship.

## C.2 Computationally Efficient Variant of MTTS under General Settings

143 In Section 4.1, we discuss how to ease the computation when directly sampling from $\mathbb{P}(\boldsymbol{r}_i|\mathbf{x}_i)$ is
144 computationally heavy, and present the variant under the sequential setting. We present the variant
145 under the general settings in Algorithm 3. Specifically, it requires an updating frequency $l$ as a
146 hyper-parameter, and will sample a new $\boldsymbol{\theta}$ once we have $l$ new data points. $\mathbb{P}(\boldsymbol{\theta}|\mathcal{H})$ can be computed
147 via various approximate posterior inference tools.

---

**Algorithm 3:** Computationally Efficient Variant of MTTS under General Settings

    **Input :** $\mathbb{P}(\boldsymbol{\theta})$, $\phi$, updating frequency $l$
1 Set $\mathcal{H} = \{\}$ and $\mathbb{P}(\boldsymbol{\theta}|\mathcal{H}) = \mathbb{P}(\boldsymbol{\theta})$
2 **for** *decision point $j = 0, \ldots,$* **do**
3     **if** $mod(j,l) = 0$ **then**
4         Update $\mathbb{P}(\boldsymbol{\theta}|\mathcal{H})$ (possibly via approximate posterior inference methods, such as
        Gibbs sampling or variational inference)
5         Sample one $\tilde{\boldsymbol{\theta}}$ from $\mathbb{P}(\boldsymbol{\theta}|\mathcal{H})$
6     **end**
7     Retrieve the task index $i$
8     Sample a reward vector $(\tilde{r}_{i,1}, \ldots, \tilde{r}_{i,K})^T$ from $c * f(\boldsymbol{r}_i|\mathbf{x}_i, \tilde{\boldsymbol{\theta}})\mathbb{P}(\mathcal{H}_i|\boldsymbol{r}_i)$, where $c$ is a
    normalization factor.
9     Take action $A_j = argmax_{a\in[K]} \tilde{r}_{i,a}$
10     Receive reward $R_j$
11     Update the dataset as $\mathcal{H} \leftarrow \mathcal{H} \cup \{(A_j, R_j, \mathbf{x}_i, i)\}$
12 **end**

---

### C.3 Implementation for Bernoulli bandits with BBLM

In this section, we discuss our implementation of MTTS for Bernoulli bandits, under the BBLM introduced in (4). It follows Algorithm 3.

Still, the dominating step is to sample from $\mathbb{P}(\boldsymbol{\theta}|\mathcal{H})$. Fortunately, the model yields a nice hierarchical structure, where the first term essentially requires us to fit a Beta-logistic regression, and the second term is the likelihood of a simple binomial distribution, with a beta conjugate in our case. Specifically, note that $f(\boldsymbol{r}_i|\mathbf{x}_i, \boldsymbol{\theta}) = \prod_{a=1}^{K} f'_{Beta}(r_{i,a}|\boldsymbol{\phi}(\mathbf{x}_i, a), \boldsymbol{\theta})$, where $f'_{Beta}(r_{i,a}|\boldsymbol{\phi}(\mathbf{x}_i, a), \boldsymbol{\theta})$ is the density function of $Beta\big(logistic(\boldsymbol{\phi}(\mathbf{x}_i, a)^T\boldsymbol{\theta}), \psi\big)$. Also note that, the likelihood for the Bernoulli distribution of task $i$ over $\mathcal{H}_i$, as specified by the second equation of (4), is $\mathbb{P}(\mathcal{H}_i|\boldsymbol{r}_i) = \prod_{a=1}^{K} r_{i,a}^{n_{i,a}}$, where $n_{i,a}$ is the number of times that action $a$ is selected for task $i$ in $\mathcal{H}_i$. Both parts are computationally tractable. In our experiments, we use the MCMC algorithm implemented by the Python package PYMC3 [34] to compute the approximate posterior.

### C.4 Computational complexity

In this section, we analyze the computational complexity for MTTS. The specific complexity depends on: (i) the reward distribution, (ii) the hierarchical model one chooses, (iii) whether the vanilla version or some variant is used, and (iv) the order of interactions. Therefore, we will first present a general complexity bound and then discuss examples. For simplicity, we assume $T_1 = \cdots = T_N = T$. Let $m(i, t)$ and $m'(i, t)$ be the number of data points for task $i$ alone and that for all tasks, until the $t$-th interaction with task $i$. The computation complexity is clearly dominated by one step: updating and sampling from the posterior. We denote its complexity at interaction $(i, t)$ by $c\big(m(i, t), m'(i, t)\big)$. Then, the total complexity is bounded by $O(\sum_{i=1}^{N} \sum_{t=1}^{T} c\big(m(i, t), m'(i, t)\big)$.

For Gaussian bandits with LMM, recall that we can update $\boldsymbol{\theta}$ incrementally, and according to its posterior form (11), each time $O(d^3)$ flops are required to sample $\boldsymbol{\theta}$. Besides, $O(dK)$ flops are required to compute the prior mean, and $O(K^2)$ flops are required to compute the posterior of $\boldsymbol{r}_i$. Therefore, with Algorithm 1, the total complexity is bounded by $O\big(NT(K^2 + d^3)\big)$. Under the sequential setting, with Algorithm 2, it can be reduced to $O\big(N(TK^2 + d^3)\big)$.

For Bernoulli bandits with BBLM, suppose $M$ samples are taken in each round of MCMC sampling, and $c'\big(m(i, t), m'(i, t)\big)$ flops are needed to sample each, then under Algorithm 3 with $l = T$, we have that the total complexity is bounded by $O\big(\sum_{t=1}^{T} Mc'\big(t, tN\big) + NTdK + NTK\big) = O\big(\sum_{t=1}^{T} Mc'\big(t, tN\big) + NTdK\big)$ under the concurrent setting, and $O\big(\sum_{i=1}^{N} Mc'\big(0, iT\big) + NdK + NTK\big) = O\big(\sum_{i=1}^{N} Mc'\big(0, iT\big) + NTK\big)$.

**Total amount of compute and the type of resources used.** Our experiments are run on an c5d.24xlarge instance on the AWS EC2 platform, with 96 cores and 192GB RAM. It takes roughly 20 minutes to complete a setting in Figure 2a and 3 hours for Figure 2b.

## D Details and Extensions of the Method

### D.1 Mean-precision parameterization of the Beta distribution

We note that a Beta distribution has different parameterizations. In the most common parameterization, a Beta random variable $u \sim Beta(\alpha_1, \alpha_2)$ is specified by two shape parameters $\alpha_1$ and $\alpha_2$. We have

$$\mu = \mathbb{E}(u) = \frac{\alpha_1}{\alpha_1 + \alpha_2}, \quad var(u) = \mu(1 - \mu)\psi/(1 + \psi),$$

where $\psi = 1/(\alpha_1 + \alpha_2)$ is the so-called precision parameter. Alternatively, this Beta random variable can be fully specified by $(\mu, \psi)$. Given $(\mu, \psi)$, we have $\alpha_1 = \mu/\psi$ and $\alpha_2 = (1 - \mu)/\psi$. Given $(\mu, var(u))$, we have $\psi = [\mu(1 - \mu)/var(u) - 1]^{-1}$.

In BBLM, we adopt this mean-precision parameterization, with $\mathbb{E}(r_{i,a}) = logistic(\boldsymbol{\phi}^T(\mathbf{x}_i, a)\boldsymbol{\theta})$ and the precision of $r_{i,a}$ equal to $\psi$, for any $i$ and $a$.

## D.2 Extension to Mixed-Effect Gaussian Process for Gaussian Bandits

In this section, we introduce an extension of the method developed in Section 4.2 to the Gaussian process setting. In such a setup, the function $f(\mathbf{x}) = \mathbb{E}(\boldsymbol{r}_i | \mathbf{x}_i = \mathbf{x})$ is a continuous function sampled from a Gaussian process. Specifically, for each action $a \in [K]$, we have a specified mean function $\mu_a : \mathcal{R}^p \to \mathcal{R}$ and a kernel function $\mathcal{K}_a : \mathcal{R}^p \times \mathcal{R}^p \to \mathcal{R}$. Let $\mathcal{GP}(\mu_a, \mathcal{K}_a)$ denote the corresponding Gaussian process. We consider the following mixed-effect Gaussian process model:

$$
\begin{aligned}
f_a &\sim \mathcal{GP}(\mu_a, \mathcal{K}_a), \forall a \in [K]; \\
\boldsymbol{\delta}_i &\sim \mathcal{N}(0, \boldsymbol{\Sigma}), \forall i \in [N]; \\
\boldsymbol{r}_i &= \mathbf{f}(\mathbf{x}_i) + \boldsymbol{\delta}_i, \forall i \in [N],
\end{aligned} \tag{1}
$$

where $\mathbf{f}(\mathbf{x}) = (f_1(\mathbf{x}), \ldots, f_K(\mathbf{x}))^T$.

To design the corresponding TS algorithm, we essentially need to derive the corresponding posterior for $\{\boldsymbol{r}_i\}$. We begin by introducing some notations. The model (1) induces a kernel function $\mathcal{K}$, such that for any two tuples $O = (a, r, \mathbf{x}, i)$ and $O' = (a', r', \mathbf{x}', i')$, we have

$$
\mathcal{K}(O, O') = \sum_a \mathcal{K}_a(\mathbf{x}, \mathbf{x}') \mathbb{I}(a = a') + \boldsymbol{\Sigma}_{a, a'} \mathbb{I}(i = i').
$$

Let $\boldsymbol{K}$ be an $n \times n$ kernel matrix of the pairwise kernel values for tuples in $\mathcal{H}$. Set $\boldsymbol{R} = (R_1, \ldots, R_n)^T$ be an $n$-dimensional vector of the observed rewards and $\boldsymbol{\mu} = (\mu_{A_1}(\mathbf{x}_{i(1)}), \ldots, \mu_{A_n}(\mathbf{x}_{i(n)}))^T$. Finally, we use $\boldsymbol{I}$ to denote an identity matrix, the dimension of which can be inferred from the context.

Define a $K \times n$ matrix $\boldsymbol{M}_i$, such that the $(a, j)$-th entry of $\boldsymbol{M}_i$ is $\mathcal{K}_a(\mathbf{x}_i, \mathbf{x}_{i(j)}) \mathbb{I}(a = A_j) + \boldsymbol{\Sigma}_{A_j, a} * \mathbb{I}(i(j) = i)$, and define a $K \times K$ diagonal matrix $\boldsymbol{K}_i$ with the $a$-th diagonal entry be $\mathcal{K}_a(\mathbf{x}_i, \mathbf{x}_i)$. Let $\boldsymbol{\mu}_i = (\mu_1(\mathbf{x}_i), \ldots, \mu_K(\mathbf{x}_i))^T$. The posterior of $\boldsymbol{r}_i$ conditional on the accumulated data follows a multivariate normal distribution, with mean and covariance given by:

$$
\begin{aligned}
\mathbb{E}(\boldsymbol{r}_i | \mathcal{H}) &= \boldsymbol{\mu}_i + \boldsymbol{M}_i (\boldsymbol{K} + \sigma_1^2 \boldsymbol{I})^{-1} (\boldsymbol{R} - \boldsymbol{\mu}), \\
cov(\boldsymbol{r}_i | \mathcal{H}) &= (\boldsymbol{K}_i + \boldsymbol{\Sigma}) - \boldsymbol{M}_i (\boldsymbol{K} + \sigma_1^2 \boldsymbol{I})^{-1} \boldsymbol{M}_i^T.
\end{aligned} \tag{2}
$$

The posterior will then be used to sample an action according to its probability of being the optimal one, in a Thompson sampling manner, as detailed in Algorithm 1.

## D.3 Adaptive hyper-parameter updating with empirical bayes

Following the literature [22, 1], we assume the variance components as known for the two examples in Section 4. In practice, we can apply empirical Bayes to update these hyperparameters adaptively. We take the LMM as an example. As discussed in Section 4.2, our model requires four hyperparametes: $\boldsymbol{\mu_\theta}$, $\boldsymbol{\Sigma_\theta}$, $\sigma$, and $\boldsymbol{\Sigma}$. Among them, $\boldsymbol{\mu_\theta}$ and $\boldsymbol{\Sigma_\theta}$, the prior parameters of $\boldsymbol{\theta}$, can be specified as any appropriate values according to domain knowledge, and will not affect our algorithm as well as its regret bound. For $\sigma$ and $\boldsymbol{\Sigma}$, similar with the approach commonly adopted in the literature on Gaussian process [38, 32], they can be learned from the data. Specifically, we can specify them as the maximizer of the marginal likelihood with respect to $(\sigma, \boldsymbol{\Sigma})$, marginalized over $\boldsymbol{\theta}$ and $\{\boldsymbol{\delta}_i\}_{i \in [N]}$. The log marginal likelihood can be derived as

$$
l(\sigma, \boldsymbol{\Sigma} | \mathcal{D}) = -\frac{1}{2} \Big[ (\boldsymbol{R} - \boldsymbol{\Phi}\boldsymbol{\mu_\theta})^T (\boldsymbol{K}(\boldsymbol{\Sigma}) + \sigma^2 \boldsymbol{I})^{-1} (\boldsymbol{R} - \boldsymbol{\Phi}\boldsymbol{\mu_\theta}) + log|\boldsymbol{K}(\boldsymbol{\Sigma}) + \sigma^2 \boldsymbol{I}| + Nlog(2\pi) \Big], \tag{3}
$$

where $\boldsymbol{K}(\boldsymbol{\Sigma})$ indicates the dependency of $\boldsymbol{K}$ on $\boldsymbol{\Sigma}$. Note that $\boldsymbol{\Sigma}$ controls the degree of heterogeneity conditional on the metadata. Such an Empirical Bayes [5] approach makes sure our method is adaptive to the prediction power of the metadata.

## D.4 Extension to contextual bandits

In this section, we discuss several possible directions to extend the metadata-based multi-task bandit framework to contextual bandits. We consider the following formulation of contextual bandits (the

other formulations can be similarly derived). For each task $i \in [N]$, at each decision point $t$, the agent choose an arm $A_{i,t} \in \mathcal{R}^p$, and then receive a stochastic reward $R_{i,t} \sim f(A_{i,t}; \boldsymbol{\beta}_i)$, where $\boldsymbol{\beta}_i$ is a length-$d$ vector of unknown parameters associated with task $i$. Here, $A_{i,t}$ is a feature vector which is a function of the context and the pulled arm. In addition, each task has a $p$-dimensional feature vector $\mathbf{x}_i$ (i.e., metadata).

A straightforward extension of the methodology presented in the main text is to consider that $\boldsymbol{\beta}_i$ is sampled from some conditional distribution on $\mathbf{x}_i$. Specifically, for some conditional distribution function $g$, we consider the following hierarchical model:

$$\begin{aligned} \boldsymbol{\theta} &\sim \mathbb{P}(\boldsymbol{\theta}), \\ \boldsymbol{\beta}_i &\sim g(\mathbf{x}_i; \boldsymbol{\theta}), \\ R_{i,t} &\sim f(A_{i,t}; \boldsymbol{\beta}_i). \end{aligned} \quad (4)$$

As a concrete example, we consider the linear Gaussian case. Let $\boldsymbol{\Theta} = (\boldsymbol{\theta}_1, \dots, \boldsymbol{\theta}_d)$ be a $p \times d$ coefficeint matrix and $\boldsymbol{\Sigma}_{\boldsymbol{\beta}}$ be some covariance matrix, we consider

$$\begin{aligned} \boldsymbol{\theta}_j &\sim \mathcal{N}(\boldsymbol{\mu}_{\boldsymbol{\theta}}, \boldsymbol{\Sigma}_{\boldsymbol{\theta}}), \forall j \in [d] \\ \boldsymbol{\beta}_i &\sim \boldsymbol{\Theta}^T \mathbf{x}_i + \mathcal{N}(\mathbf{0}, \boldsymbol{\Sigma}_{\boldsymbol{\beta}}), \forall i \in [N] \\ R_{i,t} &\sim A_{i,t}^T \boldsymbol{\beta}_i + \mathcal{N}(0, \sigma^2). \end{aligned}$$

The posterior of $\boldsymbol{\Theta}$ can be similarly developed as in the main text using the property of Gaussian bilinear model. With a sample $\tilde{\boldsymbol{\Theta}}$ from $\mathbb{P}(\boldsymbol{\Theta}|\mathcal{H})$, we can interact with each task $i$ using a single-task TS with $\mathcal{N}(\tilde{\boldsymbol{\Theta}}^T \mathbf{x}_i, \boldsymbol{\Sigma}_{\boldsymbol{\beta}})$ as the prior for $\boldsymbol{\beta}_i$.

One possible concern about model (4) is that, when the number of tasks is not large, a simpler model might be preferred. In that case, we can consider that $\boldsymbol{\beta}_i$ is a deterministic function of $\mathbf{x}_i$.

$$\begin{aligned} \boldsymbol{\theta} &\sim \mathbb{P}(\boldsymbol{\theta}) \\ \boldsymbol{\beta}_i &= g(\mathbf{x}_i; \boldsymbol{\theta}) \\ R_{i,t} &\sim f(A_{i,t}; \boldsymbol{\beta}_i). \end{aligned}$$

In reinforcement learning, this formulation shares similar forms with contextual Markov decision process [28, 29]. Now, the linear Gaussian case can be written as

$$\begin{aligned} \boldsymbol{\theta}_j &\sim \mathcal{N}(\boldsymbol{\mu}_{\boldsymbol{\theta}}, \boldsymbol{\Sigma}_{\boldsymbol{\theta}}), \forall j \in [d] \\ R_{i,t} &\sim A_{i,t}^T \boldsymbol{\Theta}^T \mathbf{x}_i + \mathcal{N}(0, \sigma^2). \end{aligned}$$

Such a model can utilize the data from all tasks, while avoiding a naive pooling which assumes the coefficient vector of all tasks are the same. Indeed, this is a kind of varying-coefficient models in statistics [10]. Low-rank assumption can also be considered in this bilinear problem.

Finally, as another direction to derive a simpler model, we may consider the idea of clustering of bandits [12, 26], by assuming there is a set of $M$ task instances, and each task are drawn independently from them. The metadata contain information about the probability that one task belong to each cluster, through a function $g$.

$$\begin{aligned} \boldsymbol{\theta} &\sim \mathbb{P}(\boldsymbol{\theta}), \\ \boldsymbol{\beta}_j &\sim \mathbb{P}(\boldsymbol{\beta}), \forall j \in [M] \\ k_i &\sim g(\mathbf{x}_i; \boldsymbol{\theta}), \forall i \in [N] \\ \boldsymbol{\beta}_i &= \sum_j \boldsymbol{I}(k_i = j) \boldsymbol{\beta}'_k, \forall i \in [N] \\ R_{i,t} &\sim f(A_{i,t}; \boldsymbol{\beta}_i), \forall i \in [N], \forall t \in [T_i]. \end{aligned}$$

Under appropriate model assumptions, the posterior can be obtained using approximate posterior inference tools.

These three models are all built under our metadata-based multi-task bandit framework, and the corresponding multi-task TS algorithms can be similarly developed. The key idea is to leverage the metadata to describe the relations between bandit tasks, while allowing heterogeneity. The choice between them reflects the bias-variance trade-off.

# E  Additional Experiment Results

## E.1  Robustness to model misspecifications

To allow efficient information sharing, we make a model assumption that $r_i|\mathbf{x}_i, \boldsymbol{\theta} \sim f(r_i|\mathbf{x}_i, \boldsymbol{\theta}), \forall i \in [N]$. When this model is correctly specified, we have shown superior theoretical and numerical performance of MTTS. However, we acknowledge that all models can be misspecifed. Intuitively, the model is used to pool information to provide an informative prior. As long as the learned prior is not significantly worse than a manually specified one, the performance would be comparable; and when the prior contains more information, we can attain a lower regret.

We empirically investigate the robustness of MTTS in this section. We focus on the Gaussian bandits case under the concurrent setting. Findings under the other settings are largely the same and hence omitted. Specifically, instead of generating data according to $r_i = \boldsymbol{\Phi}_i \boldsymbol{\theta} + \boldsymbol{\delta}_i$, we consider the data generation process $r_i = (1 - \lambda)cos(c\boldsymbol{\Phi}_i\boldsymbol{\theta})/c + \lambda\boldsymbol{\Phi}_i\boldsymbol{\theta} + \boldsymbol{\delta}_i$, where $cos$ applies the cosine function to each entry, c is a normalization constant such that the entries of $\boldsymbol{\Phi}_i\boldsymbol{\theta}$ are all in $[-\pi/2, \pi/2]$, and $\lambda \in [0, 1]$ controls the degree of misspecification. When $\lambda = 1$, we are considering the LMM; while when $\lambda = 0$, the metadata provides few information through such a linear form.

In results reported in Figure 3, we observe that MTTS is fairly robust to model misspecifications. When $\lambda = 1/2$ or $3/4$, that is, when there exists mild or moderate misspecification, MTTS still yields much lower regrets than individual-TS and meta-TS. When $\lambda = 1/4$, the performance of MTTS becomes comparable with individual-TS and meta-TS. Only when $\lambda = 0$, that is, the metadata are useless through a linear form, MTTS shows slightly higher regret in the initial period due to the additionally introduced variance. As expected, linear-TS and OSFA both severely suffer from the bias. Notably, in all cases, MTTS always yields the desired sublinear Bayes regret in $T$, as expected. Therefore, it shows that, even when the model is severely misspecified, the cost would be acceptable.

## E.2  Multi-task regrets

In the main text, we report the Bayes regret of different algorithms. Although the multi-task regrets for those figures can also be derived according to its definition, we choose to explicitly report them again in this section, in order to make the comparison more clearly.

Specifically, the multi-task regrets for Gaussian bandits and Bernoulli bandits are presented in Figure 4 and 5, respectively. In the sequential setting, we can see the multi-task regrets of MTTS converge to zero, while meta-TS and individual-TS have a constant regret, and linear-TS as well as OSFA suffer from the bias.

## E.3  Trends with experiment hyper-parameters

In this section, we report additional results under other combinations of $(K, d, T, N)$, to show that our conclusions in the main text are representative, and study the trend of the performance of MTTS. We focus on the Gaussian bandits case under the concurrent setting. To save computational cost, we set the base combination of hyper-parameters as $\sigma_1^2 = 0.5$, $K = 8$, $d = 15$, $N = 100$, and $T = 100$, and run 50 random seeds for each.

In Figure 6, we vary the value of $K$, $d$, $N$, $\sigma$ individually. Overall, MTTS consistently demonstrates lower regrets and shows robustness. Our findings can be summarized as follows.

- As $K$ increases, the learning problem for all algorithm becomes more difficult. MTTS still demonstrates better performance, and even when $T = 100$, its advantage is still fairly clear.
- As $d$ increases, the learning problem for MTTS becomes more difficult, while it still shows much better performance.
- As $\sigma$ increases, overall the learning problem for all algorithm becomes more difficult. MTTS still demonstrates better performance, and even when $T = 100$, its advantage is still fairly clear.
- As $N$ increases, MTTS can learn the task distribution more easily and its performance converges to that of oracle-TS more quickly.

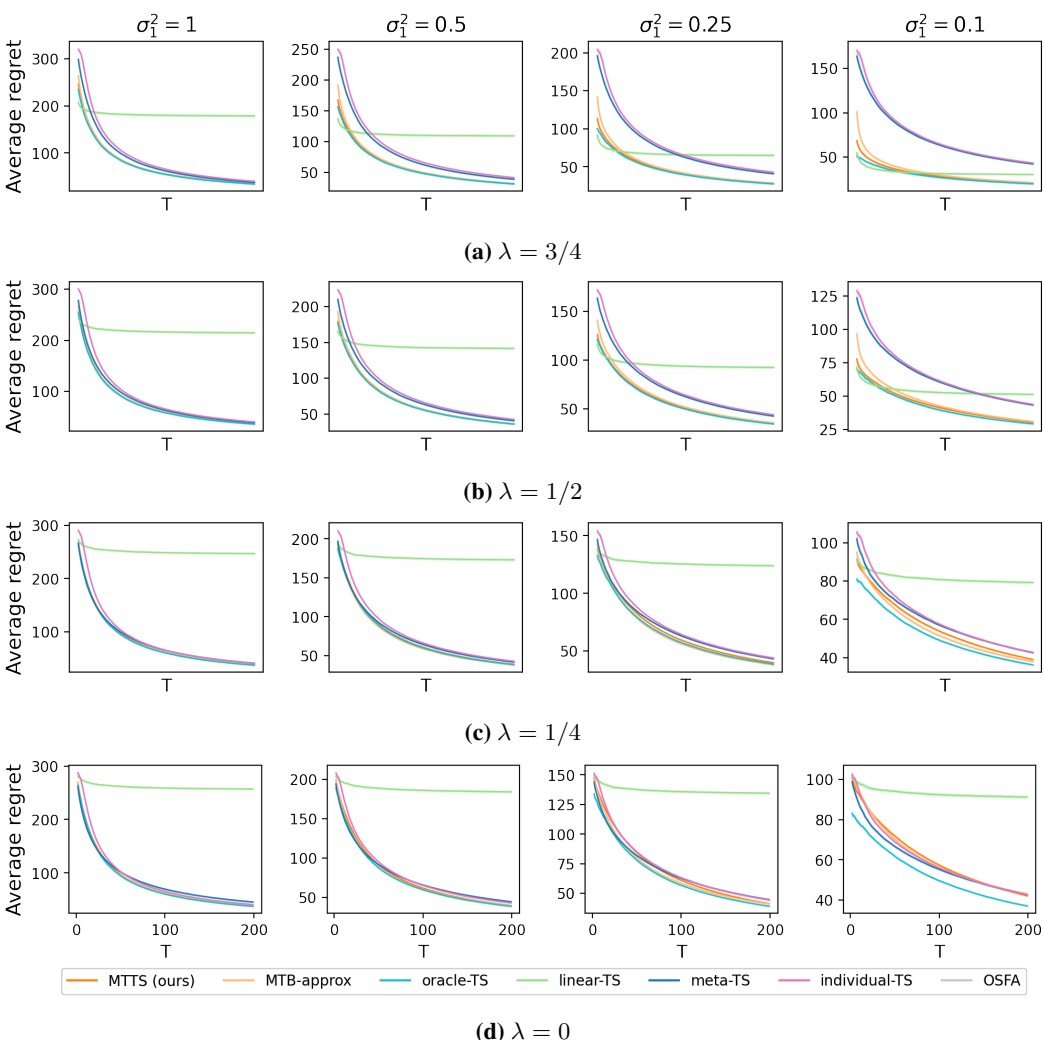

**Figure 3:** Average Bayes regret for Gaussian bandits with misspecified hierarchical models. A smaller value of $\lambda$ implies a more severe misspecification. The regrets of OSFA are an order of magnitude higher and hence hidden.

# F   More on the Experiment Details

In this section, we report additional details of our epperiments and implementations. Recall that we use $I$ to denote identity matrix, the dimension of which can be inferred from the context.

## F.1   Hyperparameters

Since all baselines that we consider are TS-type algorithms, we need to specify (i) the priors and (ii) the variance terms which are assumed to be known. For fair comparisons, we apply the law of total expectation and the law of total variance to derive these quantities. Roughly speaking, for meta-TS, we marginalize out $(\mathbf{x}, \boldsymbol{r})$ conditional on $\boldsymbol{\theta}$; for individual-TS and OSFA, we additionally marginalize out $\boldsymbol{\theta}$; For linear-TS or GLM-TS, we use the same prior of $\boldsymbol{\theta}$ as MTTS, and marginalize $\boldsymbol{r}_i$ conditional on $\mathbf{x}_i$ to set the variance of the stochastic reward variable. When the marginal distribution does not belong to a standard distribution family, we use the Gaussian distribution as an approximation.

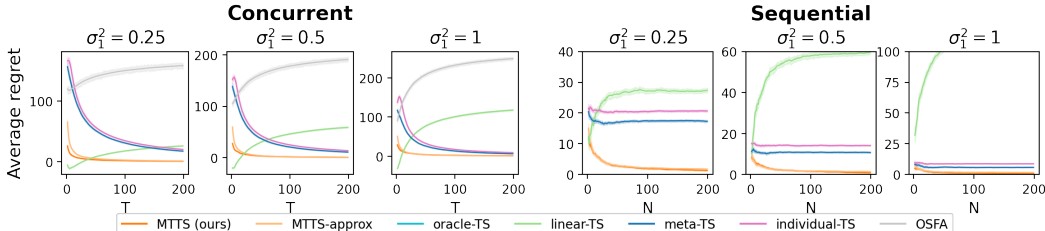

**Figure 4:** Gaussian bandits: the solid lines denote the average multi-task regret with the shared areas indicating the standard errors. The regrets of OSFA are much higher in some subplots and hence hidden.

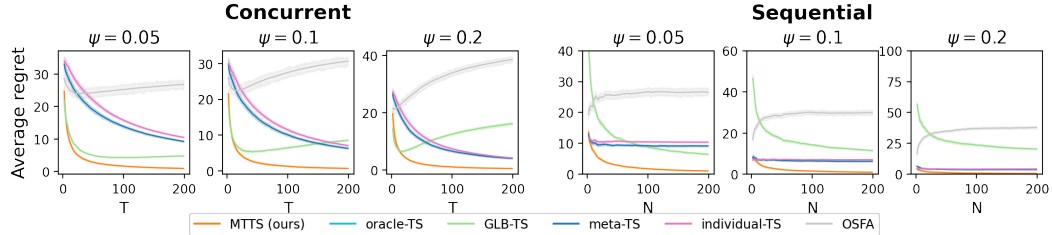

**Figure 5:** Bernoulli bandits: the solid lines denote the average multi-task regret with the shared areas indicating the standard errors. A larger value of $\psi$ implies a larger variation of $\boldsymbol{r}_i$ condition on $\mathbf{x}_i$.

**Gaussian bandits.** According to our data generation model, based on the law of total expectation and the law of total variance, it is easy to verify that

$$
\begin{aligned}
cov\big(\boldsymbol{r}_i\big|\boldsymbol{\theta}, \mathbf{x}_i\big) &= \boldsymbol{\Sigma} = \sigma_1^2 \boldsymbol{I} \\
\mathbb{E}\big(\boldsymbol{r}_i\big|\boldsymbol{\theta}\big) &= \boldsymbol{\theta}_{1:K}, \\
cov\big(\boldsymbol{r}_i\big|\boldsymbol{\theta}\big) &= \mathbb{E}\big(cov\big(\boldsymbol{r}_i|\mathbf{x}_i\big)\big|\boldsymbol{\theta}\big) + cov\big(\mathbb{E}\big(\boldsymbol{r}_i|\mathbf{x}_i\big)\big|\boldsymbol{\theta}\big) \\
&= \sigma_1^2 \boldsymbol{I} + ||\boldsymbol{\theta}_{(K+1):d}||^2 \boldsymbol{I} = (\sigma_1^2 + ||\boldsymbol{\theta}_{(K+1):d}||^2)\boldsymbol{I}, \\
\mathbb{E}\big(\boldsymbol{r}_i\big) &= \mathbf{0} \\
cov\big(\boldsymbol{r}_i\big) &= \mathbb{E}\big(cov\big(\boldsymbol{r}_i|\boldsymbol{\theta}\big)\big) + cov\big(\mathbb{E}\big(\boldsymbol{r}_i|\boldsymbol{\theta}\big)\big) \\
&= \big(\sigma_1^2 + \mathbb{E}(||\boldsymbol{\theta}_{(K+1):d}||^2)\big)\boldsymbol{I} + d^{-1}\boldsymbol{I} \\
&= \big(\sigma_1^2 + \mathbb{E}(||\boldsymbol{\theta}_{(K+1):d}||^2) + d^{-1}\big)\boldsymbol{I},
\end{aligned}
$$

where $\boldsymbol{\theta}_{1:K}$ is the first $K$ entries of $\boldsymbol{\theta}$, and $\boldsymbol{\theta}_{(K+1):d}$ is the remaining entries. Therefore, for OSFA and individual-TS, we use $\mathcal{N}\big(\mathbf{0}, \big(\sigma_1^2 + \mathbb{E}(||\boldsymbol{\theta}_{(K+1):d}||^2) + d^{-1}\big)\boldsymbol{I}\big)$ as the prior; for meta-TS, we use $\mathcal{N}\big(\boldsymbol{\mu}, (\sigma_1^2 + ||\boldsymbol{\theta}_{(K+1):d}||^2)\boldsymbol{I}\big)$ as the *unknown* prior, with $\boldsymbol{\mu}$ as an unknown parameter , and use $\mathbb{E}\big(\boldsymbol{r}_i\big|\boldsymbol{\theta}\big) = \boldsymbol{\theta}_{1:K} \sim \mathcal{N}(\mathbf{0}, d^{-1}\boldsymbol{I})$ as the hyper-prior for $\boldsymbol{\mu}$; for linear-TS, we use $\mathcal{N}(\mathbf{0}, d^{-1}\boldsymbol{I})$ as the prior for the regression coefficients $\boldsymbol{\theta}$ and $\sigma_1^2 + \sigma^2$ as the variance term of the stochastic reward variable.

**Bernoulli bandits.** According to our data generation model, based on the law of total expectation and the law of total variance, recall the discussion on the parameterization of Beta distributions in Appendix D.1 it is easy to verify that

$$
\begin{aligned}
\mathbb{E}\big(\boldsymbol{r}_i\big|\boldsymbol{\theta}\big) &= (\mathbb{E}(logistic(\boldsymbol{\phi}(\mathbf{x}_1, a)^T\boldsymbol{\theta})|\boldsymbol{\theta}), \ldots, \mathbb{E}(logistic(\boldsymbol{\phi}(\mathbf{x}_K, a)^T\boldsymbol{\theta})|\boldsymbol{\theta}))^T \\
cov\big(\boldsymbol{r}_i\big|\boldsymbol{\theta}\big) &= \mathbb{E}\big(cov\big(\boldsymbol{r}_i|\mathbf{x}_i\big)\big|\boldsymbol{\theta}\big) + cov\big(\mathbb{E}\big(\boldsymbol{r}_i|\mathbf{x}_i\big)\big|\boldsymbol{\theta}\big) \\
&= diag(c_{b,1}, \ldots, c_{b,K}) + diag(c'_{b,1}, \ldots, c'_{b,K}) \\
\mathbb{E}\big(\boldsymbol{r}_i\big) &= \frac{1}{2} * \mathbf{1} \\
cov\big(\boldsymbol{r}_i\big) &= \mathbb{E}\big(cov\big(\boldsymbol{r}_i|\boldsymbol{\theta}\big)\big) + cov\big(\mathbb{E}\big(\boldsymbol{r}_i|\boldsymbol{\theta}\big)\big),
\end{aligned}
$$

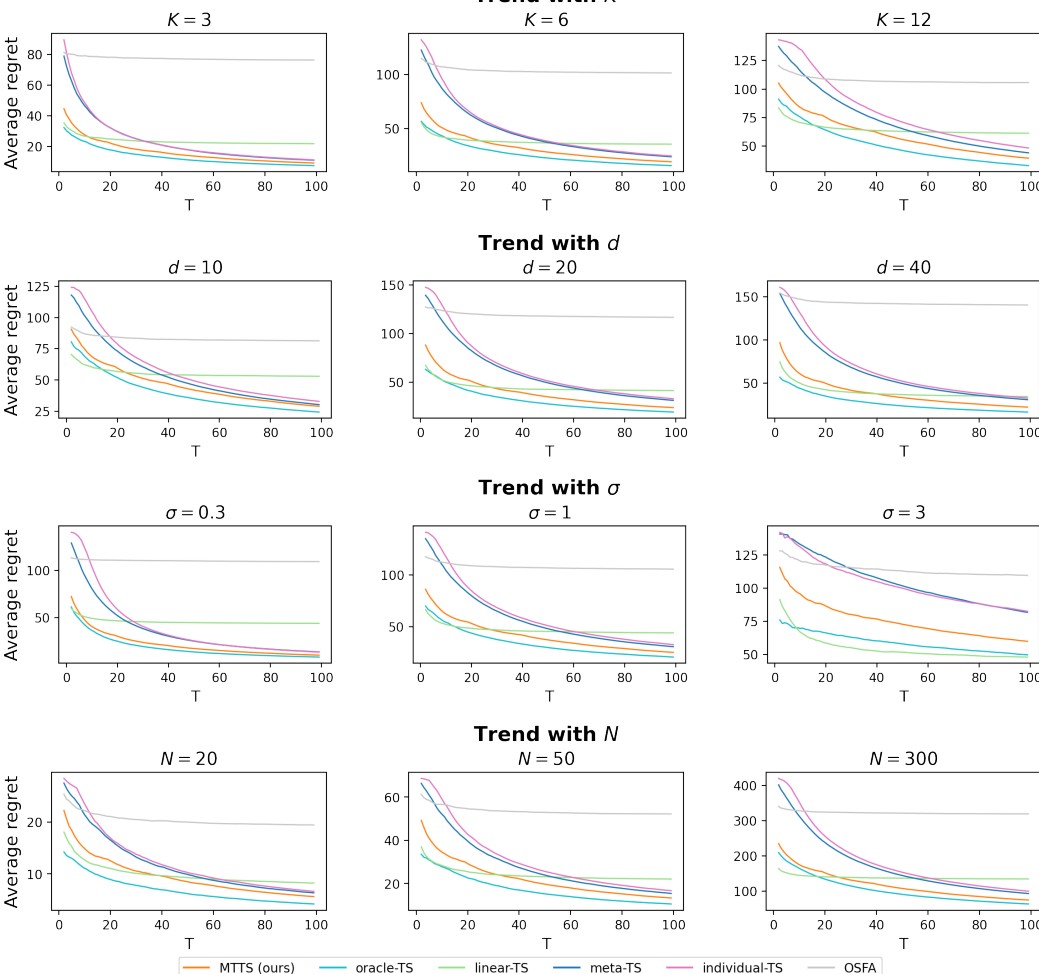

**Figure 6:** Trends of the average Bayes regret for Gaussian bandits with different parameters.

where $c_{b,i} = \frac{\psi}{1+\psi} \mathbb{E}_{\mathbf{x}}(logistic(\boldsymbol{\phi}(\mathbf{x}_i, a)^T \boldsymbol{\theta})(1 - logistic(\boldsymbol{\phi}(\mathbf{x}_i, a)^T \boldsymbol{\theta}))|\boldsymbol{\theta})$, and $c'_{b,i} = var_{\mathbf{x}}(logistic(\boldsymbol{\phi}(\mathbf{x}_i, a)^T \boldsymbol{\theta})|\boldsymbol{\theta})$. We obtain these two quantities as well as $\mathbb{E}(\boldsymbol{r}_i|\boldsymbol{\theta})$ and $cov(\boldsymbol{r}_i)$ via Monte carlo simulation, since there is no explicit form.

Define $\psi(\mu, \sigma_2^2) = [\mu(1 - \mu)/\sigma_2^2 - 1]^{-1}$. For OSFA and individual-TS, we use $Beta(1/2, \psi(1/2, cov(\boldsymbol{r}_i)))$ as the prior for each arm; for meta-TS, following the implementations in [22], we randomly pick a set of Beta distributions as candidates, with $Beta(\mathbb{E}(\boldsymbol{r}_i|\boldsymbol{\theta}), \psi(\mathbb{E}(\boldsymbol{r}_i|\boldsymbol{\theta}), cov(\boldsymbol{r}_i|\mathbf{x}_i)))$ as one of them, and maintain a Categorical distribution over them, with the uniform one as the initial prior; Finally, for GLB-TS, we choose the TS algorithm proposed in [24], and set the exploration parameter $\alpha = 1$ as in [24].

### F.2 Implementation Details

**Implementation of Meta-TS under the concurrent setting.** The original meta-TS proposed in [22] can only be applied to the episodic setting and does not fit in the concurrent setting. Based on their models, we derived the posterior and adapt meta-TS to the concurrent setting, as follows. For Gaussian bandits, meta-TS aims to learn the unknown parameter $\boldsymbol{\mu}_m = \mathbb{E}(\boldsymbol{r}_i)$, and assumes $\boldsymbol{r}_i \sim \mathcal{N}(\boldsymbol{\mu}_m, \sigma_m^2 \boldsymbol{I})$ for some known $\sigma_m^2$. See the last subsection for details. Therefore, given a dataset $\mathcal{H} = \{(A_j, R_j, \mathbf{x}_j, i(j))\}$, the posterior of $\boldsymbol{\mu}_m$ can be derived as follows. The entries of $\boldsymbol{\mu}_m$ are independent. Let $\bar{R}_{i,a}$ denote the mean observed reward for taking action $a$ in task $i$, and $n_{i,a}$ be the count. Then we have $\bar{R}_{i,a} \sim \mathcal{N}(\mu_{m,a}, \sigma_1^2 + \sigma^2/n_{i,a})$. Note the prior for $\mu_{m,a}$ is $\mathcal{N}(0, d^{-1})$.

Let $\sigma_{i,a} = \sqrt{\sigma_1^2 + \sigma^2/n_{i,a}}$. We have

$$\mathbb{P}\Big[\mu_{m,a}|\mathcal{H}\Big] \sim \mathcal{N}\Big(\frac{\sum_i \bar{R}_{i,a}/\sigma_{i,a}^2}{\sum_i \sigma_{i,a}^{-2} + d}, (\sum_i \sigma_{i,a}^{-2} + d)^{-1}\Big), \forall a \in [K]$$

Then, at each decision point, we sample one $\tilde{\boldsymbol{\mu}}_m$, and treat $\boldsymbol{r}_i \sim \mathcal{N}(\tilde{\boldsymbol{\mu}}_m, \sigma_m^2 \boldsymbol{I})$ as the prior to proceed with a standard single-task Gaussian TS algorithm. We similarly modified meta-TS for the Bernoulli bandits.

**Implementation of MTTS.**    For Bernoulli bandits, we use the Python package PyMC3 and adopt a popular MCMC algorithm , NUTS [14]. The acceptance rate is set as $0.85$, and the number of samples per time is set as $2000$. For the computationally efficient variant of MTTS, in both Gaussian bandits and Bernoulli bandits, we sample a new $\boldsymbol{\theta}$ at the end of interactions with a task under the sequential setting, and at the end of a round under the concurrent setting.

# G   Main Proof

For ease of notations, without loss of generality, we first consider the sequential setting, and we assume the tasks arrive according to their indexes, i.e., the $i$th task will arrive earlier than the $(i+1)$th. Otherwise, we can always re-index these tasks. Since in our proof, all analysis for the regret in task $i$ only uses the data generated by task $i$ itself and the data generated in the alignment periods of task $1, \ldots, i-1$, it is easy to verify that our proof continues to hold in other settings.

We first clarify some notations. We vertically stack the feature matrix for all tasks to define $\boldsymbol{\Phi}_{1:N} = (\boldsymbol{\Phi}_1; \ldots; \boldsymbol{\Phi}_N)$, and we can similarly define $\boldsymbol{\delta}_{1:N} = (\boldsymbol{\delta}_1^T, \ldots, \boldsymbol{\delta}_N^T)^T$. At the end of the alignment period of task $i$, we define the following notations: first, we define $\boldsymbol{\Phi}_{1:i}^e = (\boldsymbol{\Phi}_1; \ldots; \boldsymbol{\Phi}_i)$, and we can similarly define $\boldsymbol{\delta}_{1:i}^e$ and $\boldsymbol{R}_{1:i}^e$; moreover, we denote $\boldsymbol{V}_{1:i}^e = (\sigma^2 + 1)\boldsymbol{I}$ of dimension $iK \times iK$. finally, let $\mathcal{H}_{1:i}^e$ be the history of all alignment periods so far. To simplify the notations, when the context is clear, we may drop the subscript $1:i$ and superscript $e$. Recall that we use $\boldsymbol{I}$ to denote identity matrix, the dimension of which can be inferred from the context.

The modified MTTS algorithm is summarized in Algorithm 4, and formally described as follows. For any $i \in [N]$, during our first $K$ interactions with task $i$, we pull the $K$ arms in a round robin. After this period, we sample a value $\boldsymbol{r}_i^e$ from the posterior $\boldsymbol{r}|\mathcal{H}_{1:i}^e$, and then we use $\mathcal{N}(\boldsymbol{r}_i^e, \boldsymbol{\Sigma})$ as the prior of $\boldsymbol{r}_i$ to continue interact with task $i$ as a standard single-task TS algorithm. In another word, we run $TS(\mathcal{N}(\boldsymbol{r}_i^e, \boldsymbol{\Sigma}))$ with the following $T - K$ interactions. Note that this sampling is equivalent to first sample a value $\boldsymbol{\theta}_i^e \sim \boldsymbol{\theta}|\mathcal{H}_{1:i}^e$, and then define $\boldsymbol{r}_i^e = \boldsymbol{\Phi}_i \boldsymbol{\theta}_i^e$. Besides, as discussed in (2), this procedure is equivalent to only use the data generated in the alignment periods to estimate the prior. Finally, we define a modified oracle-TS which also has the alignment period. The main part of our proof will focus on derive the regret to the modified oracle-TS, since its regret to the vanilla oracle-TS can be easily bounded.

We begin by stating several lemmas, which will be used in our main proof. The proof of these lemmas are deferred to Appendix H. Without loss of generality, throughout the proof, we assume $\boldsymbol{\Sigma}_{\boldsymbol{\theta}} = \boldsymbol{I}$, $\boldsymbol{\mu}_{\boldsymbol{\theta}} = \boldsymbol{0}$, and $\boldsymbol{\Sigma} = \boldsymbol{I}$ to simplify the notations. It is easy to check that, under the boundedness assumption 3, for general values of these terms, the regret bound still holds.

We first establish the concentration of $\boldsymbol{\theta}_i^e$ around $\boldsymbol{\theta}$. It mainly utilizes the property of Bayesian LMM.

**Lemma 2** (Concentration of $\boldsymbol{\theta}_i^e$). *For any task $i \in [N]$, for any $\xi \in (0, 1)$, we have*

$$\mathbb{P}\Big(||\boldsymbol{\theta}_i^e - \boldsymbol{\theta}|| \geq [\frac{(\sigma^2 + 1)^{-1}}{2}ic_1 K + 1]^{-1/2}\big(2\sqrt{d} + 2\sqrt{2}\sqrt{-log\xi}\big) + \frac{||\boldsymbol{\theta}||}{\frac{(\sigma^2+1)^{-1}}{2}ic_1 K + 1}\Big) \leq 2\xi + d(\frac{e}{2})^{-\frac{c_1}{2C_1}i}.$$

Based on Lemma 2, we are now ready to prove that the estimated task-specific prior mean will be close to the true prior mean with high probability. For any task $i \in [N]$, let $\boldsymbol{r}_i^e = \boldsymbol{\Phi}_i \boldsymbol{\theta}_i^e$ be the sampled prior mean. Recall that $\mathbb{E}(\boldsymbol{r}_i|\mathbf{x}_i) = \boldsymbol{\Phi}_i \boldsymbol{\theta}$ is the true prior mean for a task with metadata $\mathbf{x}_i$. We have the following result.

**Lemma 3** (Concentration of $\boldsymbol{r}_i^e$). *For any task $i \in [N]$, we have*

$$\mathbb{P}\Big(||\boldsymbol{r}_i^e - \boldsymbol{\Phi}_i \boldsymbol{\theta}|| \geq 2\sqrt{2}C_2 \sqrt{K}\big(\sqrt{d} + \sqrt{log(NT)}\big)[ic_1' K + 1]^{-1/2} + C_3[ic_1' K + 1]^{-1}\Big) \leq \frac{2}{NT} + d(\frac{e}{2})^{-\frac{c_1}{2C_1}i}. \quad (5)$$

**Algorithm 4:** The Modified MTTS

**Input :** $\mathbb{P}(\boldsymbol{\theta})$ and known parameters for the hierarchical model

1  Set $\mathcal{H} = \{\}$ and $\mathbb{P}(\boldsymbol{\theta}|\mathcal{H}) = \mathbb{P}(\boldsymbol{\theta})$
2  Set $count(i) = 0, \forall i \in [N]$
3  **for** *each decision point $j$* **do**
4      Retrieve the task index $i$
5      **if** *count(i) ¡ K* **then**
6          count(i) = count(i) + 1
7          Take action $A_j = count(i)$ ;                        `// Alignment period`
8      **else**
9          Sample a reward vector $(\tilde{r}_{i,1}, \ldots, \tilde{r}_{i,K})^T$ from $c * \mathbb{P}(\mathcal{H}_i|\boldsymbol{r}_i)f(\boldsymbol{r}_i|\mathbf{x}_i, \tilde{\boldsymbol{\theta}}_i)$, where $c$ is
        a normalization factor. ;              `// TS with prior` $f(\boldsymbol{r}_i|\mathbf{x}_i, \tilde{\boldsymbol{\theta}}_i)$
10         Take action $A_j = argmax_{a \in [K]} \tilde{r}_{i,a}$
11     **end**
12     Receive reward $R_j$
13     Update the dataset as $\mathcal{H} \leftarrow \mathcal{H} \cup \{(A_j, R_j, \mathbf{x}_i, i)\}$
14     **if** *count(i) = K* **then**
15         Sample one $\tilde{\boldsymbol{\theta}}$ from $\mathbb{P}(\boldsymbol{\theta}|\mathcal{H})$, and denote it as $\tilde{\boldsymbol{\theta}}_i$ ;  `// Fixed down the prior for task` $i$
16 **end**

---

where $c_1' = c_1 \frac{(\sigma^2+1)^{-1}}{2}$.

To derive the multi-task regret of the modified MTTS, we aim to use a "prior alignment" proof technique inspired by [3]. Specifically, one central challenge to the analysis is that, no existing method can well characterize the behaviour of a TS algorithm when the input prior is different with the true task distribution, even when the difference is negligible [22, 3, 27]. Therefore, even though with Lemma 5, we can establish that the learned prior will be more and more close to the ground truth as the number of tasks increases, and it is intuitive to believe that the multi-task regret per task will decay, it is challenging to formally prove such a regret bound following standard regret analysis approaches.

The key idea of the "prior alignment" proof technique is that, suppose after some interactions with a task $i$, the posterior of $\boldsymbol{r}_i$ for the modified MTTS is equal to the posterior for the modified oracle-TS, then their behaviour in the following interactions with this task will be exactly the same (in the probabilistic sense), and hence the multi-task regret in the following interactions with this task will be exactly zero. Suppose such an event happens with high probability, and this probability approaches 1 as $i$ grows, we can then derive a bound for the multi-task regret. This idea forms the skeleton of our proof.

Towards this end, we begin by defining the realized random errors for the modified MTTS in the alignment period for the $i$th task as $\boldsymbol{\epsilon}_i^e = (\epsilon_{i,1}, \ldots, \epsilon_{i,K})^T$, where $\epsilon_{i,a}$ is the realized error when we pull arm $a$ in the $a$th interaction with task $i$. We can similarly define $\boldsymbol{\epsilon}_i^*$ for the modified oracle-TS algorithm. Note that both algorithms take the same actions in the alignment period, and we also have $\boldsymbol{\epsilon}_i^e \sim \boldsymbol{\epsilon}_i^* \sim \mathcal{N}(\mathbf{0}, \sigma^2 \boldsymbol{I})$. We first note the following lemma, which establishes the difference between the posterior with $\mathcal{N}(\boldsymbol{r}_i^e, \boldsymbol{\Sigma})$ as the prior and the posterior with $\mathcal{N}(\boldsymbol{\Phi}_i\boldsymbol{\theta}, \boldsymbol{\Sigma})$ as the prior, after observing $\boldsymbol{R}_i^e$.

**Lemma 4** (Posterior Difference). *For any task $i \in [N]$, let $\mathcal{N}(\tilde{\boldsymbol{r}}_i^e, \tilde{\boldsymbol{\Sigma}}_i^e)$ and $\mathcal{N}(\tilde{\boldsymbol{r}}_i^*, \tilde{\boldsymbol{\Sigma}}_i^*)$ be the posterior after observing $\boldsymbol{R}_i^e$, with $\mathcal{N}(\boldsymbol{r}_i^e, \boldsymbol{\Sigma})$ and $\mathcal{N}(\boldsymbol{\Phi}_i\boldsymbol{\theta}, \boldsymbol{\Sigma})$ as the prior, respectively. We have*

$$\tilde{\boldsymbol{r}}_i^e - \tilde{\boldsymbol{r}}_i^* = (\boldsymbol{\Sigma}^{-1} + \sigma^{-2}\boldsymbol{I})^{-1}\big[\boldsymbol{\Sigma}^{-1}(\boldsymbol{r}_i^e - \boldsymbol{\Phi}_i\boldsymbol{\theta}) + \sigma^{-2}(\boldsymbol{\epsilon}_i^e - \boldsymbol{\epsilon}_i^*)\big];$$
$$\tilde{\boldsymbol{\Sigma}}_i^e = \tilde{\boldsymbol{\Sigma}}_i^*.$$

A direct implication of this lemma is that, suppose when $\boldsymbol{r}_i^e$ is close to $\boldsymbol{\Phi}_i\boldsymbol{\theta}$, the right-hand side of the first equation is equal to 0 with high probability. Then, the multi-task regret of the modified MTTS in the following interactions with task $i$ will be exactly 0, since the posterior after the alignment period is the same.

Given Lemma 4, we are now ready to apply the "prior alignment" technique to derive a regret bound for each task. For every $i \in [N]$, we denote the stochastic cumulative rewards in the last $T - K$ interactions starting from a posterior $\tilde{r}$ as $R_i(T - K; \tilde{r})$. Denote the cumulative rewards following the optimal action by $R_i^*(T - K)$. We denote the Bayes regret of $TS(\mathcal{N}(\tilde{r}_i^e, \Sigma))$ in the last $T - K$ interactions with task $i$ by

$$BR_i(T - K; \tilde{r}_i^e) = \mathbb{E}\Big[R_i^*(T - K) - R_i(T - K; \tilde{r}_i^e)\Big].$$

We can similarly define $BR_i(T - K; \tilde{r}_i^*)$. Our proof relies on the following bound, which controls the Bayes regret of a TS algorithm in the last $T - K$ interactions starting from a different prior.

**Lemma 5.** *For any task $i \in [N]$ we have*

$$
\begin{aligned}
&\mathbb{E}_{\boldsymbol{\epsilon}_i^e}\Big[R_i^*(T - K) - R_i(T - K; \tilde{r}_i^e)\Big] \\
&\leq exp\big(\sigma\|\boldsymbol{r}_i^e - \boldsymbol{\Phi}_i\boldsymbol{\theta}\|\sqrt{2log(NT)} + \frac{\sigma^2}{2}\|(\boldsymbol{r}_i^e - \boldsymbol{\Phi}_i\boldsymbol{\theta})\|^2\big)BR_i(T - K; \tilde{r}_i^*) + \frac{2C_2C_3}{N}
\end{aligned}
\tag{6}
$$

We are now ready to combine these results and present our main proof.

*Proof of Theorem 1.* We first define the event that the estimated task-specific prior mean is close to the true one as

$$\mathcal{J}_{\boldsymbol{r}} \equiv \Big\{\|\boldsymbol{r}_i^e - \boldsymbol{\Phi}_i\boldsymbol{\theta}\| \leq 2\sqrt{2}C_2\sqrt{K}\big(\sqrt{d} + \sqrt{log(NT)}\big)[ic_1'K + 1]^{-1/2} + C_3[ic_1'K + 1]^{-1}, \forall i \in [N]\Big\}.$$

For simplicity, we denote $c_4(i) \equiv 2\sqrt{2}C_2\sqrt{K}\big(\sqrt{d} + \sqrt{log(NT)}\big)[ic_1'K + 1]^{-1/2} + C_3[ic_1'K + 1]^{-1}$. We note that $c_4(i)$ is a shorthand instead of a constant.

We first focus on bounding the regret when $\mathcal{J}_{\boldsymbol{r}}$ holds. Define

$$\mathcal{S} = \Big\{i \in [N] : 2c_4(i)\sqrt{log(NT)} \leq 1/2, \frac{\sigma}{2}c_4(i) \leq 2\sqrt{log(NT)}\Big\}.$$

We first focus on the case that $\mathcal{S}$ is not empty. Define $l = min(\mathcal{S})$. We have $l = O(dlog(NT) + log^2(NT))$ and $\mathcal{S} = \{i \in [N] : i \geq l\}$. It also implies $\frac{\sigma^2}{2}c_4(i)^2 \leq 2\sigma c_4(i)\sqrt{log(NT)}$ for $i \geq l$.

Therefore, for any $i \geq l$, by Lemma 5, we have

$$
\begin{aligned}
&\mathbb{E}\Big[R_i^*(T - K) - R_i(T - K; \tilde{r}_i^e)|\tilde{r}_i^e, \boldsymbol{\Phi}_i, \mathcal{J}_{\boldsymbol{r}}\Big] \\
&\leq exp\big(\sigma\|\boldsymbol{r}_i^e - \boldsymbol{\Phi}_i\boldsymbol{\theta}\|\sqrt{2log(NT)} + \frac{\sigma^2}{2}\|(\boldsymbol{r}_i^e - \boldsymbol{\Phi}_i\boldsymbol{\theta})\|^2\big)BR_i(T - K; \tilde{r}_i^*) + 2\frac{C_2C_3}{N} \\
&\leq exp\big(c_4(i)\sigma\sqrt{2log(NT)} + \frac{\sigma^2}{2}c_4(i)^2\big)BR_i(T - K; \tilde{r}_i^*) + 2\frac{C_2C_3}{N} \\
&\leq \big(1 + 8\sigma c_4(i)\sqrt{log(NT)}\big)BR_i(T - K; \tilde{r}_i^*) + 2\frac{C_2C_3}{N},
\end{aligned}
\tag{7}
$$

where the second inequality follows from the fact that $\|\boldsymbol{r}_i^e - \boldsymbol{\Phi}_i\boldsymbol{\theta}\| \leq c_4(i)$ conditional on $\mathcal{J}_{\boldsymbol{r}}$, and the last inequality is due to Lemma 6.

For $i < l$, similar with Lemma 11 in [3], we note that the Bayes regret for each task can be derived from the prior-independent regret bound for Gaussian bandits in the literature (e.g., [4]) as

$$\mathbb{E}\Big[R_i^*(T - K) - R_i(T - K; \tilde{r}_i^e)|\tilde{r}_i^e, \boldsymbol{\Phi}_i, \mathcal{J}_{\boldsymbol{r}}\Big] \leq C_5\sqrt{(T - K)KlogT}, \tag{8}$$

where $C_5$ is a positive constant.

When $\neg\mathcal{J}_{\boldsymbol{r}}$ holds, for $i \in [N]$, we define

$$\mathcal{J}_{\boldsymbol{r},i} \equiv \Big\{\|\boldsymbol{r}_i^e - \boldsymbol{\Phi}_i\boldsymbol{\theta}\| < 2\sqrt{2}C_2\sqrt{K}\big(\sqrt{d} + \sqrt{log(NT)}\big)[ic_1'K + 1]^{-1/2} + C_3[ic_1'K + 1]^{-1}\Big\}.$$

From Lemma 3, we know $\mathbb{P}[\neg\mathcal{J}_{\boldsymbol{r},i}] \leq 2\frac{1}{NT} + d(\frac{e}{2})^{-\frac{c_1}{2C_1}i}$. Then, by similar arguments with (8), we have

$$
\sum_{i=1}^{N} \mathbb{E}\Big[\big(R_i^*(T-K) - R_i(T-K;\tilde{\boldsymbol{r}}_i^e)\big)|\tilde{\boldsymbol{r}}_i^e, \boldsymbol{\Phi}_i, \neg\mathcal{J}_{\boldsymbol{r}}\Big] \times \mathbb{P}[\neg\mathcal{J}_{\boldsymbol{r}}]
$$
$$
\leq \sum_{i=1}^{N} \mathbb{E}\Big[\big(R_i^*(T-K) - R_i(T-K;\tilde{\boldsymbol{r}}_i^e)\big)|\tilde{\boldsymbol{r}}_i^e, \boldsymbol{\Phi}_i, \neg\mathcal{J}_{\boldsymbol{r}}\Big] \times \mathbb{P}[\neg\mathcal{J}_{\boldsymbol{r},i}] \quad (9)
$$
$$
\leq \sum_{i=1}^{N} C_5\sqrt{(T-K)K log T} \times \mathbb{P}[\neg\mathcal{J}_{\boldsymbol{r},i}]
$$
$$
\leq C_5' d\sqrt{(T-K)K log T},
$$

where $C_5'$ is an universal constant.

Finally, we note the following relationship

$$
\mathbb{E}_{\boldsymbol{\epsilon}_i^e}\Big(R_i(T-K;\tilde{\boldsymbol{r}}_i^*) - R_i(T-K;\tilde{\boldsymbol{r}}_i^e)\Big)
$$
$$
= \mathbb{E}_{\boldsymbol{\epsilon}_i^e}\Big(R_i^*(T-K) - R_i(T-K;\tilde{\boldsymbol{r}}_i^e)\Big) - \mathbb{E}_{\boldsymbol{\epsilon}_i^e}\Big(R_i^*(T-K) - R_i(T-K;\tilde{\boldsymbol{r}}_i^*)\Big)
$$
$$
= \mathbb{E}_{\boldsymbol{\epsilon}_i^e}\Big(R_i^*(T-K) - R_i(T-K;\tilde{\boldsymbol{r}}_i^e)|\mathcal{J}_{\boldsymbol{r}}\Big) \times \mathbb{P}[\mathcal{J}_{\boldsymbol{r}}] \quad (10)
$$
$$
+ \mathbb{E}_{\boldsymbol{\epsilon}_i^e}\Big(R_i^*(T-K) - R_i(T-K;\tilde{\boldsymbol{r}}_i^e)|\neg\mathcal{J}_{\boldsymbol{r}}\Big) \times \mathbb{P}[\neg\mathcal{J}_{\boldsymbol{r}}] - BR_i(T-K;\tilde{\boldsymbol{r}}_i^*)
$$

Denote the Bayes regret of the modified oracle-TS as $BR'(N, \{T_i\})$. Based on (7), (8), and (9), we sum (10) from $i = 1$ to $N$ to yield the regret of the modified MTTS to the modified oracle-TS as

$$
BR(N, \{T_i\}) - BR'(N, \{T_i\})
$$
$$
\leq \sum_{i=l}^{N} \Big[8\sigma c_4(i)\sqrt{log(NT)}BR_i(T-K;\tilde{\boldsymbol{r}}_i^*) + 2\frac{C_2 C_3}{N}\Big]
$$
$$
+ C_5 l\sqrt{(T-K)K log T} + C_5' d\sqrt{(T-K)K log T}
$$
$$
\leq \sum_{i=l}^{N} C_4'\Big[\sqrt{log(NT)}(\sqrt{K}(\sqrt{d} + \sqrt{log(NT)})[ic_1'K + 1]^{-1/2} + C_3[ic_1'K + 1]^{-1})BR_i(T-K;\tilde{\boldsymbol{r}}_i^*)
$$
$$
\Big] + O(1) + C_6(l+d)\sqrt{(T-K)K log T}
$$
$$
\leq \sum_{i=l}^{N} C_4'\Big[\sqrt{log(NT)}(\sqrt{K}(\sqrt{d} + \sqrt{log(NT)})[ic_1'K + 1]^{-1/2} + C_3[ic_1'K + 1]^{-1})BR_i(T-K;\tilde{\boldsymbol{r}}_i^*)\Big] + O(1)
$$
$$
+ C_6'(d log(NT) + log^2(NT))\sqrt{(T-K)K log T}
$$
$$
= O(\sqrt{log(NT)}(\sqrt{d} + \sqrt{log(NT)})\sqrt{N}\sqrt{(T-K)K log T} + (log(NT)d + log^2(NT))\sqrt{(T-K)K log T}),
$$

where $C_4'$, $C_6$, and $C_6'$ some universal constants, the last inequality is due to $l = O(d log(NT) + log^2(NT))$, and the last equality is due to $BR_i(T-K;\tilde{\boldsymbol{r}}_i^*) = O(\sqrt{(T-K)K log T})$ according to Proposition 2 in [33]. In the last equality, we also utilize the fact that $\sum_{i=l}^{N} 1/\sqrt{i} = O(\sqrt{N})$, and $\sum_{i=l}^{N} 1/i = O(log(N))$. Recall that, until now, we consider the case that $\mathcal{S}$ is not empty. When $\mathcal{S}$ is empty, we have $N = O(log^2(NT))$, and then we can follow the arguments of (8) to bound the regret when $\mathcal{J}_{\boldsymbol{r}}$ holds as $O(log^2(NT)\sqrt{(T-K)K log T})$, and we can similarly obtain the above bound.

Finally, we bound the regret of the modified oracle-TS to the vanilla oracle-TS. We denote the action that the modified oracle-TS takes at the $t$-th interaction with task $i$ as $A_{i,t}^{\tilde{\mathcal{O}}}$. This multi-task regret of

the modified oracle-TS is then equal to

$$\mathbb{E}_{\mathbf{x},\boldsymbol{r},\epsilon} \sum_{t=1}^{T} (r_{i,A_{i,t}^{\mathcal{O}}} - r_{i,A_{i,t}^{\tilde{\mathcal{O}}}}) = \mathbb{E}_{\mathbf{x},\boldsymbol{r},\epsilon} \Big[ \Big(\sum_{t=1}^{K} r_{i,A_{i,t}^{\mathcal{O}}} - \sum_{t=1}^{K} r_{i,A_{i,t}^{\tilde{\mathcal{O}}}}\Big) + \Big(\sum_{t=T-K+1}^{T} r_{i,A_{i,t}^{\mathcal{O}}} - \sum_{t=T-K+1}^{T} r_{i,A_{i,t}^{\tilde{\mathcal{O}}}}\Big) \Big]$$
$$= O(K).$$

Here, for the first term, We note the regret from $K$ interactions with task $i$ is always bounded by $K max(\boldsymbol{r}_i)$, the expectation of which over the task distribution is bounded by $K C_2 C_3$. The second part is bounded by $0$, since the two algorithms share the same prior and the modified oracle-TS essentially have $K$ more data points with no confounding variables [45].

Putting everything together, we conclude with

$$MTR(N, \{T_i\})$$
$$= BR(N, \{T_i\}) - BR'(N, \{T_i\}) + N \times O(K)$$
$$= O(\sqrt{log(NT)}(\sqrt{d} + \sqrt{log(NT)})\sqrt{N}\sqrt{(T-K)KlogT} + (log(NT)d + log^2(NT))\sqrt{(T-K)KlogT} + NK)$$
$$= O(\sqrt{log(NT)}(\sqrt{d} + \sqrt{log(NT)})\sqrt{N}\sqrt{TKlogT} + log^2(NT)\sqrt{(T-K)KlogT} + NK)$$
$$= \tilde{O}(\sqrt{N}\sqrt{dTK} + NK).$$

$\square$

Finally, we remark that, similar to some literature on meta bandits [43, 3], we adopt the *task distribution* viewpoint by assuming the tasks (and hence $\{\mathbf{x}_i\}_{i=1}^{N}$) are i.i.d. and considering Assumption 1. Following the standard proof approach with adversarial contexts [1, 8, 2] in contextual bandits, it would be feasible to relax these assumptions. Specifically, notice that the only place we need Assumption 1 is in Lemma 2 and 3, where we apply properties of linear mixed model with i.i.d. data to control the estimation error (and sampling error) of the task-specific prior mean $\boldsymbol{\Phi}_i\boldsymbol{\theta}$ at rate $\mathcal{O}(\sqrt{d/i})$ with high probability. Here $i$ is the index of the current task. This result mainly leads to an $\mathcal{O}(\sum_{i=1}^{N} \sqrt{d/i}) = \mathcal{O}(\sqrt{Nd})$ term (Appendix G), the product of which with the single-task Bayes regret $\tilde{\mathcal{O}}(\sqrt{TK})$ leads to the $\tilde{\mathcal{O}}(\sqrt{NdTK})$ term in our multi-task regret. Without Assumption 1, we can bound the estimation error by (approximately) $\mathcal{O}(\|\boldsymbol{\Phi}_i(\sum_{j=1}^{i-1} \boldsymbol{\Phi}_j^T \boldsymbol{\Phi}_j)^{-1}\boldsymbol{\Phi}_i^T\|)$ with high probability, the summation of which from $i = 1$ to $n$ can be bounded similarly as $\tilde{\mathcal{O}}(\sqrt{Nd})$, following similar arguments of Lemma 3 of [8]. The proof technique relies on careful relating the cumulative prediction error with the eigenspace of the growing design matrix, and is largely standard in the literature (starting from [2]).

# H   Proof of Lemmas and Propositions

## H.1   Proof of Lemma 2

*Proof.* Throughout the proof, recall the fact that $\boldsymbol{\theta}$ is a fixed vector instead of a random vector, and we only adopt the Bayesian approach to adapt the TS framework. We first note that, according to the results on Bayesian LMM (page 361, [42]), we have the following expression for the posterior of $\boldsymbol{\theta}$:

$$\boldsymbol{\theta}|\mathcal{H}_{1:i}^e \sim \mathcal{N}\Big((\boldsymbol{\Phi}^T\boldsymbol{V}^{-1}\boldsymbol{\Phi} + \boldsymbol{I})^{-1}\boldsymbol{\Phi}^T\boldsymbol{V}^{-1}\boldsymbol{R}, (\boldsymbol{\Phi}^T\boldsymbol{V}^{-1}\boldsymbol{\Phi} + \boldsymbol{I})^{-1}\Big). \tag{11}$$

Recall that, when the context is clear, we may drop the subscript $1 : i$ and superscript $e$. Let $\hat{\boldsymbol{\theta}}_i^e = (\boldsymbol{\Phi}^T\boldsymbol{V}^{-1}\boldsymbol{\Phi} + \boldsymbol{I})^{-1}\boldsymbol{\Phi}^T\boldsymbol{V}^{-1}\boldsymbol{R}$ be the maximum a posterior estimator of $\boldsymbol{\theta}$, which implies

$$\hat{\boldsymbol{\theta}}_i^e - \boldsymbol{\theta}|\boldsymbol{\Phi} \sim \mathcal{N}\Big(-[(\sigma^2+1)^{-1}\boldsymbol{\Phi}^T\boldsymbol{\Phi}+\boldsymbol{I}]^{-1}\boldsymbol{\theta}, [(\sigma^2+1)^{-1}\boldsymbol{\Phi}^T\boldsymbol{\Phi}+\boldsymbol{I}]^{-1} - [(\sigma^2+1)^{-1}\boldsymbol{\Phi}^T\boldsymbol{\Phi}+\boldsymbol{I}]^{-2}\Big). \tag{12}$$

**Concentration of $\hat{\boldsymbol{\theta}}_i^e$ around $\boldsymbol{\theta}$.**   To derive the concentration of $\hat{\boldsymbol{\theta}}_i^e - \boldsymbol{\theta}$ around $\mathbf{0}$, we analyze the concentration of its mean around $\mathbf{0}$ and the magnitude of its variance separately. We begin by defining the event

$$\mathcal{J}_{\boldsymbol{\Phi}} \equiv \big\{\sigma_{\min}[\boldsymbol{\Phi}^T\boldsymbol{\Phi}] \geq \frac{1}{2}ic_1K\big\},$$

According to Lemma 7, based on Assumption 1, we have

$$\mathbb{P}(\mathcal{J}_{\boldsymbol{\Phi}}) \geq 1 - d\left(\frac{e}{2}\right)^{-\frac{c_1}{2C_1}i}. \tag{13}$$

For the mean part of (12), based on the property of matrix operator norm, we can derive

$$\begin{aligned}
\mathcal{J}_{\boldsymbol{\Phi}} &= \left\{\sigma_{\min}[\boldsymbol{\Phi}^T\boldsymbol{\Phi}] \geq \frac{1}{2}ic_1K\right\} \\
&= \left\{\sigma_{\min}[(\sigma^2+1)^{-1}\boldsymbol{\Phi}^T\boldsymbol{\Phi} + \boldsymbol{I}] \geq \frac{(\sigma^2+1)^{-1}}{2}ic_1K + 1\right\} \\
&= \left\{\left\|[(\sigma^2+1)^{-1}\boldsymbol{\Phi}^T\boldsymbol{\Phi} + \boldsymbol{I}]^{-1}\right\| \leq \frac{1}{\frac{(\sigma^2+1)^{-1}}{2}ic_1K + 1}\right\} \\
&\subseteq \left\{\left\|[(\sigma^2+1)^{-1}\boldsymbol{\Phi}^T\boldsymbol{\Phi} + \boldsymbol{I}]^{-1}(-\boldsymbol{\theta})\right\| \leq \frac{\|\boldsymbol{\theta}\|}{\frac{(\sigma^2+1)^{-1}}{2}ic_1K + 1}\right\},
\end{aligned}$$

where the third equality is due to $\sigma_{min}(\mathbf{A}) = \|\mathbf{A}^{-1}\|$ for any invertible matrix $\mathbf{A}$. This relationship implies

$$\mathbb{E}[\hat{\boldsymbol{\theta}}_i^e - \boldsymbol{\theta}|\mathcal{J}_{\boldsymbol{\Phi}}] \leq \frac{\|\boldsymbol{\theta}\|}{\frac{(\sigma^2+1)^{-1}}{2}ic_1K + 1}.$$

For the variance part of (12), define a random vector $\mathbf{z} \sim \mathcal{N}(\mathbf{0}, [(\sigma^2+1)^{-1}\boldsymbol{\Phi}^T\boldsymbol{\Phi} + \boldsymbol{I}]^{-1} - [(\sigma^2+1)^{-1}\boldsymbol{\Phi}^T\boldsymbol{\Phi} + \boldsymbol{I}]^{-2})$. According to the tail inequality for the Euclidean norm of Gaussian random vectors (see Lemma 8), for any $\xi \in (0,1)$, we have

$$\mathbb{P}\left[\|\mathbf{z}\| \leq \sigma_{\mathbf{z}}\sqrt{d} + \sigma_{\mathbf{z}}\sqrt{2(-log\xi)}\Big|\boldsymbol{\Phi}\right] \geq 1 - \xi, \tag{14}$$

where $\sigma_{\mathbf{z}} = \|[(\sigma^2+1)^{-1}\boldsymbol{\Phi}^T\boldsymbol{\Phi} + \boldsymbol{I}]^{-1} - [(\sigma^2+1)^{-1}\boldsymbol{\Phi}^T\boldsymbol{\Phi} + \boldsymbol{I}]^{-2}\|^{1/2}$. We then focus on control $\sigma_{\mathbf{z}}$:

$$\begin{aligned}
\sigma_{\mathbf{z}} &= \|[(\sigma^2+1)^{-1}\boldsymbol{\Phi}^T\boldsymbol{\Phi} + \boldsymbol{I}]^{-1}\{\boldsymbol{I} - [(\sigma^2+1)^{-1}\boldsymbol{\Phi}^T\boldsymbol{\Phi} + \boldsymbol{I}]^{-1}\}\|^{1/2} \\
&\leq \left\{\|[(\sigma^2+1)^{-1}\boldsymbol{\Phi}^T\boldsymbol{\Phi} + \boldsymbol{I}]^{-1}\| \times \|\boldsymbol{I} - [(\sigma^2+1)^{-1}\boldsymbol{\Phi}^T\boldsymbol{\Phi} + \boldsymbol{I}]^{-1}\|\right\}^{1/2} \\
&\leq \|[(\sigma^2+1)^{-1}\boldsymbol{\Phi}^T\boldsymbol{\Phi} + \boldsymbol{I}]^{-1}\|^{1/2}
\end{aligned}$$

where the first inequality follows from the sub-multiplicative property of the matrix operator norm, and the second follows from the fact that $\|\boldsymbol{I} - (\boldsymbol{I} + \boldsymbol{A})^{-1}\| \leq 1$ for any symmetric matrix $\boldsymbol{A}$.

Therefore, conditional on $\mathcal{J}_{\boldsymbol{\Phi}}$, we have $\sigma_{\mathbf{z}} \leq [\frac{(\sigma^2+1)^{-1}}{2}ic_1K + 1]^{-1/2}$, which together with (14) implies

$$\mathbb{P}\left[\|\mathbf{z}\| \leq [\frac{(\sigma^2+1)^{-1}}{2}ic_1K + 1]^{-1/2}(\sqrt{d} + \sqrt{2}\sqrt{-log\xi})\Big|\mathcal{J}_{\boldsymbol{\Phi}}\right] \geq 1 - \xi.$$

Note that, based on the triangle inequality, we have $\|\hat{\boldsymbol{\theta}}_i^e - \boldsymbol{\theta}\| \leq \|\mathbb{E}(\hat{\boldsymbol{\theta}}_i^e|\mathcal{J}_{\boldsymbol{\Phi}}) - \boldsymbol{\theta}\| + \|\hat{\boldsymbol{\theta}}_i^e - \mathbb{E}(\hat{\boldsymbol{\theta}}_i^e|\mathcal{J}_{\boldsymbol{\Phi}})\|$. Combining the two parts, we conclude with

$$\mathbb{P}\left(\|\hat{\boldsymbol{\theta}}_i^e - \boldsymbol{\theta}\| \geq [\frac{(\sigma^2+1)^{-1}}{2}ic_1K + 1]^{-1/2}(\sqrt{d} + \sqrt{2}\sqrt{-log\xi}) + \frac{\|\boldsymbol{\theta}\|}{\frac{(\sigma^2+1)^{-1}}{2}ic_1K + 1}\Big|\mathcal{J}_{\boldsymbol{\Phi}}\right) \leq \xi. \tag{15}$$

**Concentration of $\theta_i^e$ around $\hat{\theta}_i^e$.** Note that

$$\boldsymbol{\theta}_i^e - \hat{\boldsymbol{\theta}}_i^e|\boldsymbol{\Phi} \sim \mathcal{N}\left(\mathbf{0}, (\boldsymbol{\Phi}^T\boldsymbol{V}^{-1}\boldsymbol{\Phi} + \boldsymbol{I})^{-1}\right). \tag{16}$$

We begin by defining a random vector $\mathbf{z} \sim \mathcal{N}(\mathbf{0}, ((\sigma^2+1)^{-1}\boldsymbol{\Phi}^T\boldsymbol{\Phi} + \boldsymbol{I})^{-1})$. By similar arguments with that for the variance part in (12), we get

$$\mathbb{P}\left[\|\mathbf{z}\| \leq [\frac{(\sigma^2+1)^{-1}}{2}ic_1K + 1]^{-1/2}(\sqrt{d} + \sqrt{2}\sqrt{-log\xi})\Big|\mathcal{J}_{\boldsymbol{\Phi}}\right] \geq 1 - \xi,$$

which implies

$$\mathbb{P}\Big[\|\boldsymbol{\theta}_i^e - \hat{\boldsymbol{\theta}}_i^e\| \le [\frac{(\sigma^2+1)^{-1}}{2}ic_1K + 1]^{-1/2}\big(\sqrt{d} + \sqrt{2}\sqrt{-log\xi}\big)|\mathcal{J}_{\boldsymbol{\Phi}}\Big] \ge 1 - \xi. \qquad (17)$$

Note that, based on the triangle inequality, we have $\|\boldsymbol{\theta}_i^e - \boldsymbol{\theta}\| \le \|\boldsymbol{\theta}_i^e - \hat{\boldsymbol{\theta}}_i^e\| + \|\hat{\boldsymbol{\theta}}_i^e - \boldsymbol{\theta}\|$. Therefore, applying an union bound to (15) and (17) yields that

$$\mathbb{P}\Big[\|\boldsymbol{\theta}_i^e - \boldsymbol{\theta}\| \le [\frac{(\sigma^2+1)^{-1}}{2}ic_1K + 1]^{-1/2}\big(2\sqrt{d} + 2\sqrt{2}\sqrt{-log\xi}\big) + \frac{\|\boldsymbol{\theta}\|}{\frac{(\sigma^2+1)^{-1}}{2}ic_1K + 1}|\mathcal{J}_{\boldsymbol{\Phi}}\Big] \ge 1 - 2\xi.$$

This result, together with (13), implies that

$$\mathbb{P}\big(\|\boldsymbol{\theta}_i^e - \boldsymbol{\theta}\| \ge [\frac{(\sigma^2+1)^{-1}}{2}ic_1K + 1]^{-1/2}\big(2\sqrt{d} + 2\sqrt{2}\sqrt{-log\xi}\big) + \frac{\|\boldsymbol{\theta}\|}{\frac{(\sigma^2+1)^{-1}}{2}ic_1K + 1}\big) \le 2\xi + d(\frac{e}{2})^{-\frac{c_1}{2C_1}i}.$$

$\square$

## H.2   Proof of Lemma 3

*Proof.* The first term needs assumptions

$$\begin{aligned}
\|\boldsymbol{r}_i^e - \boldsymbol{\Phi}_i\boldsymbol{\theta}\| &= \|\boldsymbol{\Phi}_i\boldsymbol{\theta}_i^e - \boldsymbol{\Phi}_i\boldsymbol{\theta}\| \\
&\le \|\boldsymbol{\Phi}_i\| \times \|\boldsymbol{\theta}_i^e - \boldsymbol{\theta}\| \\
&\le \sqrt{K}C_2\|\boldsymbol{\theta}_i^e - \boldsymbol{\theta}\|,
\end{aligned}$$

where the last inequality is based on Assumption 2. Recall the results in Lemma 2 that

$$\begin{aligned}
\mathbb{P}\Big(\|\boldsymbol{r}_i^e - \boldsymbol{\Phi}_i\boldsymbol{\theta}\| \ge \sqrt{K}C_2\big([\frac{(\sigma^2+1)^{-1}}{2}ic_1K + 1]^{-1/2}\big(2\sqrt{d} + 2\sqrt{2}\sqrt{-log\xi}\big) \\
+ \frac{\|\boldsymbol{\theta}\|}{\frac{(\sigma^2+1)^{-1}}{2}ic_1K + 1}\big)\Big) \le 2\xi + d(\frac{e}{2})^{-\frac{c_1}{2C_1}i}.
\end{aligned} \qquad (18)$$

Under Assumption 2, with $c_1' = c_1\frac{(\sigma^2+1)^{-1}}{2}$, by setting $\xi = \frac{1}{NT}$, we can derive

$$\mathbb{P}\Big(\|\boldsymbol{r}_i^e - \boldsymbol{\Phi}_i\boldsymbol{\theta}\| \ge 2\sqrt{2}C_2\sqrt{K}\big(\sqrt{d} + \sqrt{log(NT)}\big)[ic_1'K + 1]^{-\frac{1}{2}} + C_3[ic_1'K + 1]^{-1}\Big) \le \frac{2}{NT} + d(\frac{e}{2})^{-\frac{c_1}{2C_1}i}. \quad (19)$$

$\square$

## H.3   Proof of Lemma 4

*Proof.* The relationship follows from the posterior updating rule for multivariate Gaussian. Specifically, we have

$$\begin{aligned}
\tilde{\boldsymbol{r}}_i^e &= (\boldsymbol{\Sigma}^{-1} + \sigma^{-2}\boldsymbol{I})^{-1}(\boldsymbol{\Sigma}^{-1}\boldsymbol{r}_i^e + \sigma^{-2}\boldsymbol{R}_i^e); \\
\tilde{\boldsymbol{\Sigma}}_i^e &= (\boldsymbol{\Sigma}^{-1} + \sigma^{-2}\boldsymbol{I})^{-1}; \\
\tilde{\boldsymbol{r}}_i^* &= (\boldsymbol{\Sigma}^{-1} + \sigma^{-2}\boldsymbol{I})^{-1}(\boldsymbol{\Sigma}^{-1}\boldsymbol{\Phi}_i\boldsymbol{\theta} + \sigma^{-2}\boldsymbol{R}_i^*); \\
\tilde{\boldsymbol{\Sigma}}_i^* &= (\boldsymbol{\Sigma}^{-1} + \sigma^{-2}\boldsymbol{I})^{-1},
\end{aligned}$$

which implies

$$\begin{aligned}
\tilde{\boldsymbol{r}}_i^e - \tilde{\boldsymbol{r}}_i^* &= (\boldsymbol{\Sigma}^{-1} + \sigma^{-2}\boldsymbol{I})^{-1}\big[\boldsymbol{\Sigma}^{-1}(\boldsymbol{r}_i^e - \boldsymbol{\Phi}_i\boldsymbol{\theta}) + \sigma^{-2}(\boldsymbol{\epsilon}_i^e - \boldsymbol{\epsilon}_i^*)\big] \\
\tilde{\boldsymbol{\Sigma}}_i^e &= \tilde{\boldsymbol{\Sigma}}_i^*.
\end{aligned}$$

$\square$

## H.4 Proof of Lemma 5

*Proof.* Recall that

$$\tilde{\boldsymbol{r}}_i^e - \tilde{\boldsymbol{r}}_i^* = (\boldsymbol{\Sigma}^{-1} + \sigma^{-2}\boldsymbol{I})^{-1}\big[\boldsymbol{\Sigma}^{-1}(\boldsymbol{r}_i^e - \boldsymbol{\Phi}_i\boldsymbol{\theta}) + \sigma^{-2}(\boldsymbol{\epsilon}_i^e - \boldsymbol{\epsilon}_i^*)\big],$$

which implies $\tilde{\boldsymbol{r}}_i^e = \tilde{\boldsymbol{r}}_i^*$ when $\boldsymbol{\epsilon}_i^* = \boldsymbol{\epsilon}_i^e + \sigma^2\boldsymbol{\Sigma}^{-1}(\boldsymbol{r}_i^e - \boldsymbol{\Phi}_i\boldsymbol{\theta})$. We denote

$$h_i(\boldsymbol{\epsilon}_i^e) \equiv \boldsymbol{\epsilon}_i^e + \sigma^2\boldsymbol{\Sigma}^{-1}(\boldsymbol{r}_i^e - \boldsymbol{\Phi}_i\boldsymbol{\theta}), \tag{20}$$

which will then allow us to apply a change-of-variable trick. We note this is an one-to-one mapping. Besides, due to the round robin nature, we have $\boldsymbol{\epsilon}_i^* \sim \boldsymbol{\epsilon}_i^e \sim \mathcal{N}(\boldsymbol{0}, \sigma^2\boldsymbol{I})$.

Define $\mathcal{J}_{\boldsymbol{\epsilon},\boldsymbol{r},i}$ as the event $\{|(\boldsymbol{r}_i^e - \boldsymbol{\Phi}_i\boldsymbol{\theta})^T\boldsymbol{\epsilon}_i^e| \le \sigma\|\boldsymbol{r}_i^e - \boldsymbol{\Phi}_i\boldsymbol{\theta}\|\sqrt{2log(NT)}\}$. We begin by expressing $BR_i(T - K; \tilde{\boldsymbol{r}}_i^e)$ as a function of $BR_i(T - K; \tilde{\boldsymbol{r}}_i^*)$ via a change of measure.

$$\mathbb{E}_{\boldsymbol{\epsilon}_i^e}\Big[(R_i^*(T - K) - R_i(T - K; \tilde{\boldsymbol{r}}_i^e))\Big]$$

$$= \int_{\boldsymbol{\epsilon}_i^e} \frac{exp(-\|\boldsymbol{\epsilon}_i^e\|^2/2\sigma^2)}{(2\pi\sigma^2)^{K/2}}(R_i^*(T - K) - R_i(T - K; \tilde{\boldsymbol{r}}_i^e)d\boldsymbol{\epsilon}_i^e)$$

$$= \int_{\boldsymbol{\epsilon}_i^e} \frac{exp(-\|\boldsymbol{\epsilon}_i^e\|^2/2\sigma^2)}{exp(-\|h_i(\boldsymbol{\epsilon}_i^e)\|^2/2\sigma^2)}\frac{exp(-\|h_i(\boldsymbol{\epsilon}_i^e)\|^2/2\sigma^2)}{(2\pi\sigma^2)^{K/2}}(R_i^*(T - K) - R_i(T - K; \tilde{\boldsymbol{r}}_i^e)d\boldsymbol{\epsilon}_i^e)$$

$$= \int_{\boldsymbol{\epsilon}_i^e} \mathbb{I}[\mathcal{J}_{\boldsymbol{\epsilon},\boldsymbol{r},i}]\frac{exp(-\|\boldsymbol{\epsilon}_i^e\|^2/2\sigma^2)}{exp(-\|h_i(\boldsymbol{\epsilon}_i^e)\|^2/2\sigma^2)}\frac{exp(-\|h_i(\boldsymbol{\epsilon}_i^e)\|^2/2\sigma^2)}{(2\pi\sigma^2)^{K/2}}(R_i^*(T - K) - R_i(T - K; \tilde{\boldsymbol{r}}_i^e)d\boldsymbol{\epsilon}_i^e)$$

$$+ \int_{\boldsymbol{\epsilon}_i^e} \mathbb{I}[\neg\mathcal{J}_{\boldsymbol{\epsilon},\boldsymbol{r},i}]\frac{exp(-\|\boldsymbol{\epsilon}_i^e\|^2/2\sigma^2)}{exp(-\|h_i(\boldsymbol{\epsilon}_i^e)\|^2/2\sigma^2)}\frac{exp(-\|h_i(\boldsymbol{\epsilon}_i^e)\|^2/2\sigma^2)}{(2\pi\sigma^2)^{K/2}}(R_i^*(T - K) - R_i(T - K; \tilde{\boldsymbol{r}}_i^e)d\boldsymbol{\epsilon}_i^e). \tag{21}$$

In what follows, we will control the two parts of (21) separately.

**First part of** (21). For the first part of (21), conditional on $\mathcal{J}_{\boldsymbol{r}}$, we have

$$\int_{\boldsymbol{\epsilon}_i^e} \mathbb{I}[\mathcal{J}_{\boldsymbol{\epsilon},\boldsymbol{r},i}]\frac{exp(-\|\boldsymbol{\epsilon}_i^e\|^2/2\sigma^2)}{exp(-\|h_i(\boldsymbol{\epsilon}_i^e)\|^2/2\sigma^2)}\frac{exp(-\|h_i(\boldsymbol{\epsilon}_i^e)\|^2/2\sigma^2)}{(2\pi\sigma^2)^{K/2}}(R_i^*(T - K) - R_i(T - K; \tilde{\boldsymbol{r}}_i^e)d\boldsymbol{\epsilon}_i^e)$$

$$\le max\Big\{\mathbb{I}[\mathcal{J}_{\boldsymbol{\epsilon},\boldsymbol{r},i}]exp\Big(\frac{\|h_i(\boldsymbol{\epsilon}_i^e)\|^2 - \|\boldsymbol{\epsilon}_i^e\|^2}{2\sigma^2}\Big)\Big\}\int_{\boldsymbol{\epsilon}_i^e} \mathbb{I}[\mathcal{J}_{\boldsymbol{\epsilon},\boldsymbol{r},i}]\frac{exp(-\|h_i(\boldsymbol{\epsilon}_i^e)\|^2/2\sigma^2)}{(2\pi\sigma^2)^{K/2}}(R_i^*(T - K) - R_i(T - K; \tilde{\boldsymbol{r}}_i^e)d\boldsymbol{\epsilon}_i^e)$$

$$\le max\Big\{\mathbb{I}[\mathcal{J}_{\boldsymbol{\epsilon},\boldsymbol{r},i}]exp\Big(\frac{\|h_i(\boldsymbol{\epsilon}_i^e)\|^2 - \|\boldsymbol{\epsilon}_i^e\|^2}{2\sigma^2}\Big)\Big\}\int_{\boldsymbol{\epsilon}_i^e} \frac{exp(-\|h_i(\boldsymbol{\epsilon}_i^e)\|^2/2\sigma^2)}{(2\pi\sigma^2)^{K/2}}(R_i^*(T - K) - R_i(T - K; \tilde{\boldsymbol{r}}_i^e)d\boldsymbol{\epsilon}_i^e)$$

$$= max\Big\{\mathbb{I}[\mathcal{J}_{\boldsymbol{\epsilon},\boldsymbol{r},i}]exp\Big(\frac{\|h_i(\boldsymbol{\epsilon}_i^e)\|^2 - \|\boldsymbol{\epsilon}_i^e\|^2}{2\sigma^2}\Big)\Big\}BR_i(T - K; \tilde{\boldsymbol{r}}_i^*), \tag{22}$$

where the second inequality follows from the fact that the integrand is non-negative. Recall that, without loss of generality, we have assumed $\boldsymbol{\Sigma} = \boldsymbol{I}$. To control this term, we use the relationship (20) to yield

$$max\Big\{\mathbb{I}[\mathcal{J}_{\boldsymbol{\epsilon},\boldsymbol{r},i}]exp\Big(\frac{\|h_i(\boldsymbol{\epsilon}_i^e)\|^2 - \|\boldsymbol{\epsilon}_i^e\|^2}{2\sigma^2}\Big)\Big\}$$

$$= max\Big\{\mathbb{I}[\mathcal{J}_{\boldsymbol{\epsilon},\boldsymbol{r},i}]exp\Big(\frac{\|\boldsymbol{\epsilon}_i^e + \sigma^2(\boldsymbol{r}_i^e - \boldsymbol{\Phi}_i\boldsymbol{\theta})\|^2 - \|\boldsymbol{\epsilon}_i^e\|^2}{2\sigma^2}\Big)\Big\}$$

$$= max\Big\{\mathbb{I}[\mathcal{J}_{\boldsymbol{\epsilon},\boldsymbol{r},i}]exp\Big((\boldsymbol{\epsilon}_i^e)^T(\boldsymbol{r}_i^e - \boldsymbol{\Phi}_i\boldsymbol{\theta}) + \frac{\sigma^2}{2}\|(\boldsymbol{r}_i^e - \boldsymbol{\Phi}_i\boldsymbol{\theta})\|^2)\Big)\Big\}$$

$$\le exp\Big(\sigma\|\boldsymbol{r}_i^e - \boldsymbol{\Phi}_i\boldsymbol{\theta}\|\sqrt{2log(NT)} + \frac{\sigma^2}{2}\|(\boldsymbol{r}_i^e - \boldsymbol{\Phi}_i\boldsymbol{\theta})\|^2\Big),$$

where the inequality is due to the definition of $\mathcal{J}_{\boldsymbol{\epsilon},\boldsymbol{r},i}$.

**Second part of** (21). For the second part of (21), we can bound it by

$$\int_{\boldsymbol{\epsilon}_i^e} \mathbb{I}[\neg \mathcal{J}_{\boldsymbol{\epsilon},\boldsymbol{r},i}] \frac{exp(-\|\boldsymbol{\epsilon}_i^e\|^2/2\sigma^2)}{exp(-\|h_i(\boldsymbol{\epsilon}_i^e)\|^2/2\sigma^2)} \frac{exp(-\|h_i(\boldsymbol{\epsilon}_i^e)\|^2/2\sigma^2)}{(2\pi\sigma^2)^{K/2}} (R_i^*(T-K) - R_i(T-K;\tilde{\boldsymbol{r}}_i^e)d\boldsymbol{\epsilon}_i^e)$$

$$\leq \mathbb{E}\Big[R_i^*(T-K) - R_i(T-K;\tilde{\boldsymbol{r}}_i^e), \neg \mathcal{J}_{\boldsymbol{\epsilon},\boldsymbol{r},i}\Big] \times \mathbb{P}[\neg \mathcal{J}_{\boldsymbol{\epsilon},\boldsymbol{r},i}]. \tag{23}$$

For the first term of (23), we note the regret from $T-K$ interactions with task $i$ is always bounded by $(T-K)max(\boldsymbol{r}_i)$, the expectation of which over the task distribution is bounded by $(T-K)C_2C_3$.

For the second term of (23), we recall the tail inequality of Gaussian distributions: for any random variable $z \sim \mathcal{N}(0,\sigma^2)$, we have $\mathbb{P}[|z| \geq c\|\boldsymbol{r}_i^e - \boldsymbol{\Phi}_i\boldsymbol{\theta}\|\sqrt{2log(NT)}] \leq 2exp(-\|\boldsymbol{r}_i^e - \boldsymbol{\Phi}_i\boldsymbol{\theta}\|^2 c^2 log(NT)/\sigma^2)$, where $c$ is any constant. Notice that $(\boldsymbol{r}_i^e - \boldsymbol{\Phi}_i\boldsymbol{\theta})$ is independent with $\boldsymbol{\epsilon}_i^e$, which implies that, for any fixed value of $\boldsymbol{r}_i^e - \boldsymbol{\Phi}_i\boldsymbol{\theta}$, it holds that $(\boldsymbol{r}_i^e - \boldsymbol{\Phi}_i\boldsymbol{\theta})^T\boldsymbol{\epsilon}_i^e \sim \mathcal{N}(0,\sigma^2\|\boldsymbol{r}_i^e - \boldsymbol{\Phi}_i\boldsymbol{\theta}\|^2)$. Therefore, by setting $c = \sigma$, we have

$$\mathbb{P}[\neg \mathcal{J}_{\boldsymbol{\epsilon},\boldsymbol{r},i}] = \mathbb{P}[|(\boldsymbol{r}_i^e - \boldsymbol{\Phi}_i\boldsymbol{\theta})^T\boldsymbol{\epsilon}_i^e| \geq \sigma\|\boldsymbol{r}_i^e - \boldsymbol{\Phi}_i\boldsymbol{\theta}\|\sqrt{2log(NT)}]$$

$$= \mathbb{P}[|(\boldsymbol{r}_i^e - \boldsymbol{\Phi}_i\boldsymbol{\theta})^T\boldsymbol{\epsilon}_i^e| \geq \sigma\|\boldsymbol{r}_i^e - \boldsymbol{\Phi}_i\boldsymbol{\theta}\|\sqrt{2log(NT)}|\boldsymbol{r}_i^e - \boldsymbol{\Phi}_i\boldsymbol{\theta}]$$

$$\leq 2exp(-log(NT)) = \frac{2}{NT}$$

Putting these two parts together, we obtain a bound for the second term of (21) as

$$\int_{\boldsymbol{\epsilon}_i^e} \mathbb{I}[\neg \mathcal{J}_{\boldsymbol{\epsilon},\boldsymbol{r},i}] \frac{exp(-\|\boldsymbol{\epsilon}_i^e\|^2/2\sigma^2)}{exp(-\|h_i(\boldsymbol{\epsilon}_i^e)\|^2/2\sigma^2)} \frac{exp(-\|h_i(\boldsymbol{\epsilon}_i^e)\|^2/2\sigma^2)}{(2\pi\sigma^2)^{K/2}} (R_i^*(T-K) - R_i(T-K;\tilde{\boldsymbol{r}}_i^e)d\boldsymbol{\epsilon}_i^e)$$

$$\leq \frac{2C_2C_3}{N}. \tag{24}$$

Finally, combining (22) and (24), we can obtain

$$\mathbb{E}_{\boldsymbol{\epsilon}_i^e}\Big[R_i^*(T-K) - R_i(T-K;\tilde{\boldsymbol{r}}_i^e)\Big]$$

$$\leq exp\big(\sigma\|\boldsymbol{r}_i^e - \boldsymbol{\Phi}_i\boldsymbol{\theta}\|\sqrt{2log(NT)} + \frac{\sigma^2}{2}\|(\boldsymbol{r}_i^e - \boldsymbol{\Phi}_i\boldsymbol{\theta})\|^2\big)BR_i(T-K;\tilde{\boldsymbol{r}}_i^*) + \frac{2C_2C_3}{N} \tag{25}$$

$$\square$$

## H.5    Regrets of the baseline TS algorithms

In this section, we first recap the baseline TS algorithms discussed in Section 5, and then provide formal statements about their regret bounds. OSFA applies a single $TS(Q(\boldsymbol{r}_i))$ algorithm to all tasks, where $Q(\boldsymbol{r}_i)$ is the marginal distribution of $\boldsymbol{r}_i$, while individual-TS applies a separate $TS(Q(\boldsymbol{r}_i))$ algorithm to each tasks. For meta-TS, under the sequential setting, following [22], we apply $TS(\mathcal{N}(\boldsymbol{\mu}_i,\boldsymbol{\Sigma}))$ to the $i$-th task, where $\boldsymbol{\mu}_i$ is sampled from the posterior of $\mathbb{E}[\boldsymbol{r}_i]$ based on accumulated data from the finished $i-1$ tasks. Finally, linear-TS assumes $\boldsymbol{r}_i = \boldsymbol{\Phi}_i\boldsymbol{\theta}$, requires a prior over $\boldsymbol{\theta}$ and maintains a posterior over it. We have the following results.

**Proposition 1.** *Under Assumptions* $1-3$, *when* $K < min(N,T)$, *the multi-task regrets of OSFA and linear-TS under the LMM over $N$ tasks with $T$ interactions per task are both bounded by* $O(NT)$.

**Proposition 2.** *Under Assumptions* $1-3$, *when* $K < min(N,T)$, *the multi-task regret of individual-TS and meta-TS under the LMM over $N$ tasks with $T$ interactions per task are both bounded by* $O(N\sqrt{KT})$.

These two propositions can be proved as follows.

*Proof of Proposition 1.* According to Assumption 2, we note the regret from one interaction with task $i$ is always bounded by $Kmax(\boldsymbol{r}_i)$, the expectation of which over the task distribution is bounded by $C_2C_3$. Therefore, the total regrets over $N$ tasks with $T$ interactions per task are both bounded by $O(NT)$. $\square$

*Proof of Proposition 2.* The Bayes regret for each task can be derived from the prior-independent regret bound for Gaussian bandits in the literature (e.g., [4]) as $O(\sqrt{(T-K)KlogT})$. Therefore, the regret accumulated over $N$ tasks can be bounded by $O(N\sqrt{(T-K)KlogT})$. $\qquad\square$

We note that, similar with results in [3] and [22], it is non-trivial to obtain lower bounds on the multi-task regrets for these baseline TS algorithms, due to the lack of understanding of the behaviour of a TS algorithm with mis-specified priors. To our knowledge, the above bounds are the tightest in the existing literature. In contrast, with information-sharing and thanks to the prior alignment technique, MTTS can be shown to yield a lower regret in rate.

### H.6   Additional technical lemmas

In this section, we collect several additional technical lemmas. We first recap a mathematical result:

**Lemma 6** (Lemma 20 in [3]). *For any number $a \in [0,1]$, it holds that $exp(a) \leq 1 + 2a$.*

The following lemma will give a lower bound for the smallest eigenvalue of our design matrix.

**Lemma 7** (Theorem 3.1 in [39]). *For a series of independent, positive semidefinite matrices $\{\boldsymbol{A}_k\}$ with dimension d, suppose $||\boldsymbol{A}_k|| \leq R$ almost surely, then for any $\delta \in [0,1)$, we have*

$$\mathbb{P}\big[\sigma_{min}(\sum_k \boldsymbol{A}_k) \leq (1-\delta)\mu_{min}\big] \leq d[\frac{e^{-\delta}}{(1-\delta)^{1-\delta}}]^{\mu_{min}/R},$$

*where $\mu_{min} = \sigma_{min}(\sum_k \mathbb{E}\boldsymbol{A}_k)$.*

The following lemma states a tail inequality for the 2-norm of a Gaussian vector.

**Lemma 8** (Based on Lemma A.4 in [19]). *For a d-dimensional random vector $\mathbf{z} \sim \mathcal{N}(0, \boldsymbol{I})$ and any $\xi \in (0,1)$, we have*

$$\mathbb{P}[||\mathbf{z}||^2 \leq d + 2\sqrt{d(-log\xi)} + 2(-log\xi)] \geq 1 - \xi,$$

*which implies, for any matrix $\boldsymbol{A}$ with appropriate dimensions, we have*

$$\mathbb{P}\Big[||\boldsymbol{A}\mathbf{z}|| \leq ||\boldsymbol{A}||(\sqrt{d} + \sqrt{2(-log\xi)})\Big]$$
$$= \mathbb{P}\Big[||\boldsymbol{A}\mathbf{z}|| \leq ||\boldsymbol{A}||(d + 2\sqrt{2}\sqrt{d(-log\xi)} + 2(-log\xi))^{1/2}\Big]$$
$$\geq \mathbb{P}\Big[||\boldsymbol{A}\mathbf{z}|| \leq ||\boldsymbol{A}||(d + 2\sqrt{d(-log\xi)} + 2(-log\xi))^{1/2}\Big] \geq 1 - \xi.$$

*Notice that $\boldsymbol{A}\mathbf{z} \sim \mathcal{N}(0, \boldsymbol{A}^T\boldsymbol{A})$.*