# OpenReview forum: "Metadata-based Multi-Task Bandits with Bayesian Hierarchical Models"
_NeurIPS.cc/2021/Conference — NeurIPS 2021 Poster_

### Official Review · Reviewer_JzUf · 2021-07-12

**Rating:** 6
**Confidence:** 3

**Summary:**

Authors provide a new solution of the problem of transfer learning across different bandit tasks. Specifically they assume policy to leverage external task-features describing task similarities. They focus on Bernoulli and Gaussian bandits as learning instances.
The theoretical findings have been tested on synthetic data.

**Limitations And Societal Impact:**

See above.

**Main Review:**

Authors claimed to introduce two novel ingredients with respect to traditional works on meta-learning with bandit tasks. First they say to consider a more general setting thanks to the presence of metadata. Secondly, here task interactions are allowed to arrive in arbitrary order. Yet, they are not forced to arrive sequentially.
To the best of my knowledge I am not aware of the usage of metadata to explain bandit task similarites. I found the idea interesting and novel. That beind said, I would not say that this is a more general framework as it does not incorporate all the task-similarities assumptions that have been investigated in the literature. Indeed, I would consider it an alternative framework.

In the state of the art about meta-learning with bandit tasks I would incorporate the following recent works:
- Simchowitz et al.. Bayesian decision-making under misspecified priors with applications to meta-learning. https://arxiv.org/abs/2107.01509
- L Cella, M Pontil. Multi-Task and Meta-Learning with Sparse Linear Bandits. UAI 2021.
- Moradipari et al.. Parameter and Feature Selection in Stochastic Linear Bandits. https://arxiv.org/abs/2106.05378

As far as the MTTS regret bound is concerned, I wonder if there is a lower bound result in the literature to better understand the quality of the proposed solution. For instance, it seems to me that setting $d=N$ the MTTS regret bound would be of order N\sqrt{TK} which is the same of individual-TS. Am I wrong?

Finally, I wonder if there are clear limitations in running experiments over real data.
I found the paper clear and well written.

**Time Spent Reviewing:**

8

---

> ### Author Response · Authors · 2021-08-10
> **Response to Reviewer JzUf**
>
> We sincerely appreciate your careful reading, constructive questions and suggestions.  In the following, we present our point-by-point responses to your comments.
>
> * **Real data.**
> We greatly appreciate this valuable comment! We agree that real data experiments can further support the proposed method.
> One challenge comes from the absence of
> open-source dataset of logged feedbacks from meta bandits, hence many papers on meta/multi-task bandits (e.g., [1], [2], [3]) focus on simulated datasets.
> That said, we strongly agree on the value of open-source datasets and real experiments.
> To address your concern, we have evaluated the algorithms on a popular recommender system dataset, Movielens 1M [4].
> The dataset consists of the ratings (reward) by  6000 users (tasks) on 4000 movies.
> Each user has three features (metadata), including gender, age, and occupation.
> The movies belong to 18 genres (arms).
> We preprocessed the dataset to leave users with at least $500$ data points, which gives us $N = 175$ users.
> We apply algorithms considered in our experiments to recommend movie genres to users and collect rewards by sampling corresponding records from the dataset (e.g., [5], [6], [7], [8]).
> We repeat the experiment over 100 random seeds, with $T = 100$.
> The average cumulative regrets against the empirical oracle (which knows the user-specific optimal genre with the highest empirical mean reward) are reported in the following table.
> The lower the better (standard errors in the parentheses). The results demonstrate the benefits of metadata-based information sharing and emphasize the existence of heterogeneity. Plots, experiment details, and reproducible code have been uploaded to the anonymous Github link "https://anonymous.4open.science/r/MTTS-Real-Analysis-FFE9 ".
> We will add the real data experiment to the final version. Thanks again!
>
>
> Algorithm  |MTTS | MTTS-approx | linear-TS | meta-TS | individual-TS | OSFA
> ------------- | ------------- | ------------- | ------------- | ------------- | ------------- | -------------
> Regret (episodic)  | **2498.0** (13.1) | 2672.3 (13.7) | 3360.9 (65.4) | 3028.4 (11.5) | 3031.4 (10.7) | 4721.9 (3.9)
> Regret (concurrent)  |  **2534.2** (16.3) | 2569.2 (14.0) | 2898.2 (77.7) | 3004.8 (15.2) | 2967.1 (15.5)  | 4656.7 (9.9)
>
>
>
> * **Lower bound.**
> Many thanks for this suggestion.
> We agree some discussion about the lower bound is quite meaningful.
> Up to the best of our knowledge, similar bounds are absent in the literature, and remain a challenging open question as noted in [9, 10].
> We acknowledge that it is challenging to derive such a lower bound, as the multi-task regret is the difference between two Bayes regrets, which requires a quite precise description of the behaviors of Bayesian bandit algorithms under our setup.
> We conjecture that the dependency on $\sqrt{NTKd}$ is unavoidable, as the problem complexity naturally scales with these parameters, and most possibly at least at a root-n parametric rate.
> We will add related discussion in the final version and emphasize this as a future direction.
>
>
> * **Upper bound when N = d.**
> Your derivation is absolutely correct.
> This "N = d" setting actually corresponds to the high-dimensional problem.
> Indeed, without additional assumptions (e.g., sparsity), the number of tasks is not sufficient for us to learn shared structures for so many unknown parameters. Therefore, we stated this point in Theorem 1 and also in line 311 that "MTTS is particularly valuable in task-rich settings ($N \gg d$)".
> We will further emphasize the appropriate applications of MTTS in our final version, and consider the high-dimensional problem as our next step.
>
>
> * **Alternative frameworks.**
> We greatly appreciate this valuable comment!
> We agree that there are other multi-task bandit approaches, depending on the information available.
> As mentioned in line 28, lines 96-103 and Appendix B, there are several existing methods using information such as social networks or bounded pairwise difference, and our approach is the first one to leverage metadata.
> As you pointed out, our method adds one useful tool to the toolbox.
> We believe our approach is widely applicable, as features are commonly available and most practitioners are familiar with feature-based methods.
> Finally, we would like to clarify that,
> the framework is claimed as general in the sense that,
> assuming the existence of features,
> there is no other specific requirements on the generative model (linear model, Gaussian process, Bayesian network, etc.) or on the reward distribution.
> We will further discuss the other frameworks in the final version.
>
>
> * **Recommended references.**
> Thank you very much for referring us to these latest papers!
> Since the three papers were all released this June or July,  we couldn't include them in our current version, but they are definitely related to meta bandits and we will add them in our final version.
> Reference 1 provides an $O(T^2 \epsilon)$ bound on the additional regret for Bayesian bandit algorithms with misspecified prior, where $\epsilon$ is the total-variation between the true and the misspecified priors.
> Although a direct application of this bound to MTTS is looser in $T$ than ours, the dependency on $\epsilon$ is quite interesting and can provide readers with more insights.
> Reference 2 and 3 study utilizing the low-rank assumption or joint sparsity assumption in meta linear bandits, and can provide readers a wider picture of meta bandits (we also included some references in Appendix B, due to page limit).
> We will include all the three papers in our final version.
>
>
> *  [1] Kveton et al.. Meta-Thompson Sampling (http://proceedings.mlr.press/v139/kveton21a.html)
> *  [2] Boutilier et al.. Differentiable Meta-Learning of Bandit Policies (https://proceedings.neurips.cc/paper/2020/file/171ae1bbb81475eb96287dd78565b38b-Paper.pdf)
> *  [3] Hsu et al.. Empirical Bayes Regret Minimization (https://arxiv.org/abs/1904.02664)
> *  [4] https://grouplens.org/datasets/movielens/
> *  [5] Rao, D. (2020). Contextual Bandits for adapting to changing User preferences over time. (https://arxiv.org/abs/2009.10073)
> *  [6] Chen et al.. Fair contextual multi-armed bandits (http://proceedings.mlr.press/v124/chen20a.html)
> *  [7] Gupta et al.. A unified approach to translate classical bandit algorithms to the structured bandit setting. IEEE Journal on Selected Areas in Information Theory, 1(3), 840-853.
> *  [8] Gupta et al.. Best-Arm Identification in Correlated Multi-Armed Bandits. IEEE Journal on Selected Areas in Information Theory, 2(2), 549-563.
> *  [9] Kveton et al.. Meta-Thompson Sampling (http://proceedings.mlr.press/v139/kveton21a.html)
> *  [10] Bastani, H.; Simchi-Levi, D.; and Zhu, R. 2019. Meta Dynamic Pricing: Transfer Learning Across Experiments. Available at SSRN 3334629 .

---

### Official Review · Reviewer_ALPC · 2021-07-12

**Rating:** 6
**Confidence:** 4

**Summary:**

This paper proposes a Bayesian hierarchical modelling framework to learn shared information across multiple multi-armed bandits. It extends both meta MAB that doesn't uses task specific metadata and linear bandit models that does not assume inter-task heterogeneity. It defines a new regret, called multi-task regret and provides theoretical analysis on the regret bound compared to other methods. Simulated experiments confirm the superior performance against other baselines when the model is well specified. It also shows robustness when under some model specification. Unfortunately, there are no real data experiments to show the performance in practice.

**Limitations And Societal Impact:**

See my comments above about the lack of real data evaluation.

**Main Review:**

This paper is well written. It has a good motivation and a general introduction to the problem. It has sufficient discussions on related work. The general hierarchical Bayesian model framework is natural under the problem setting, and the inference algorithm is standard in the Bayesian literature. It discusses two concrete and simple problem instantiations under linear Gaussian and Beta-Bernoulli assumptions with more extensions in the appendix.

One of the main contribution is the regret analysis using the MTR. It shows the advantage of the proposed method when the model is well specified. It would actually be useful to comment in the scenario with model misspecification, in which case I guess it will reduces to the same bound as individual TS in Table 1 (or worse depending on the type of misspecification).

I am a bit disappointed about the lack of empirical evaluation on real problems.
Experiments in the main text shows the advantage of the proposed method when the data generation process is the same as the model prior. In my opinion, it merely serves as a sanity check that the Bayesian inference algorithm works as expected. The additional experiments in E.1 to show the robustness under misspecified reward structure is actually more interest and I would suggest move it to the main text. Nonetheless, I think it would make the paper much stronger if the authors could show it outperforms contextual bandit methods by modelling the additional inter-task heterogeneity in common problems studied in that literature.

As a minor comment, as many concepts and proving techniques are inspired by [6], it maybe be helpful to the reader by mentioning it in the related work section and explaining the difference from this work.

**Time Spent Reviewing:**

3

---

> ### Author Response · Authors · 2021-08-10
> **Response to Reviewer ALPC**
>
> We sincerely appreciate your careful reading, constructive questions and suggestions.  In the following, we present our point-by-point responses to your comments.
>
>
> * **Real data.**
> We greatly appreciate this valuable comment! We agree that real data experiments can further support the proposed method.
> One challenge comes from the absence of
> open-source dataset of logged feedbacks from meta bandits, hence many papers on meta/multi-task bandits (e.g., [1], [2], [3]) focus on simulated datasets.
> That said, we strongly agree on the value of open-source datasets and real experiments.
> To address your concern, we have evaluated the algorithms on a popular recommender system dataset, Movielens 1M [4].
> The dataset consists of the ratings (reward) by  6000 users (tasks) on 4000 movies.
> Each user has three features (metadata), including gender, age, and occupation.
> The movies belong to 18 genres (arms).
> We preprocessed the dataset to leave users with at least $500$ data points, which gives us $N = 175$ users.
> We apply algorithms considered in our experiments to recommend movie genres to users and collect rewards by sampling corresponding records from the dataset (e.g., [5], [6], [7], [8]).
> We repeat the experiment over 100 random seeds, with $T = 100$.
> The average cumulative regrets against the empirical oracle (which knows the user-specific optimal genre with the highest empirical mean reward) are reported in the following table.
> The lower the better (standard errors in the parentheses). The results demonstrate the benefits of metadata-based information sharing and emphasize the existence of heterogeneity. Plots, experiment details, and reproducible code have been uploaded to the anonymous Github link "https://anonymous.4open.science/r/MTTS-Real-Analysis-FFE9 ". We will add the real data experiment to the final version. Thanks again!
>
>
> Algorithm  |MTTS | MTTS-approx | linear-TS | meta-TS | individual-TS | OSFA
> ------------- | ------------- | ------------- | ------------- | ------------- | ------------- | -------------
> Regret (episodic)  | **2498.0** (13.1) | 2672.3 (13.7) | 3360.9 (65.4) | 3028.4 (11.5) | 3031.4 (10.7) | 4721.9 (3.9)
> Regret (concurrent)  |  **2534.2** (16.3) | 2569.2 (14.0) | 2898.2 (77.7) | 3004.8 (15.2) | 2967.1 (15.5)  | 4656.7 (9.9)
>
>
>
>
>
>
>
> * **Multi-task regret (MTR) under model misspecifications.** This is a meaningful and important question.
> The conjecture is correct that our MTR will be of the same order with individual-TS.
> As demonstrated in the robustness analysis, which one is better depends on the degree of misspecifications, and MTTS typically outperforms.
> This question is essentially the same as the prediction task:
> with some carefully chosen features and functional forms (i.e., the inter-task layer), can we trust the predicted prior is better than a manually chosen one?
> If so, we should go with the data-driven approach.
> We agree it is pretty useful to comment on this point and will add it to the final version.
>
>
> * **Move experiments under model misspecifications to the main text.**
> Many thanks for another valuable comment.
> We agree that model misspecifications always exist, and the robustness results in E.1 are of interest and can further support MTTS.
> The performance with no model misspecification can demonstrate (i) the finite-sample performance (as the theory is a big-O bound) and (ii) the magnitude of improvement.
> We defer the robustness results to the appendix given the page limit.
> Following your suggestion, we plan to better organize the main text and move part of the robustness results to the experiment section, and also further emphasize this point in the paper.
>
>
>
> * **Reference [6].** Thank you for the suggestion.
> [6] studies the dynamic pricing problem with a linear demand model, under the metadata-agnostic setting.
> Therefore, [6] considers a different problem.
> It inspires us on the regret notion and the prior alignment technique, so we cited [6] several times.
> It is a good idea to move [6] to the related work section with more discussions.
> We will update in the final version.
>
>
>
> *  [1] Kveton et al.. Meta-Thompson Sampling (http://proceedings.mlr.press/v139/kveton21a.html)
> *  [2] Boutilier et al.. Differentiable Meta-Learning of Bandit Policies (https://proceedings.neurips.cc/paper/2020/file/171ae1bbb81475eb96287dd78565b38b-Paper.pdf)
> *  [3] Hsu et al.. Empirical Bayes Regret Minimization (https://arxiv.org/abs/1904.02664)
> *  [4] https://grouplens.org/datasets/movielens/
> *  [5] Rao, D. (2020). Contextual Bandits for adapting to changing User preferences over time. (https://arxiv.org/abs/2009.10073)
> *  [6] Chen et al.. Fair contextual multi-armed bandits (http://proceedings.mlr.press/v124/chen20a.html)
> *  [7] Gupta et al.. A unified approach to translate classical bandit algorithms to the structured bandit setting. IEEE Journal on Selected Areas in Information Theory, 1(3), 840-853.
> *  [8] Gupta et al.. Best-Arm Identification in Correlated Multi-Armed Bandits. IEEE Journal on Selected Areas in Information Theory, 2(2), 549-563.

---

### Official Review · Reviewer_sdSm · 2021-07-16

**Rating:** 6
**Confidence:** 3

**Summary:**


The paper studies the multi-task multi-armed bandits, where tasks are played sequentially and each task is governed by some unknown hidden parameter (metadata) which follows certain prior distribution.

1. The paper comes up with a Bayesian framework for the metadata-based multi-task multi-armed bandits.
2. The paper designs Thompson sampling-type algorithm, called MTTS, and demonstrates its supremacy over other algorithms, by proving regret bounds for the common Bernoulli / Gaussian prior cases.
3. The paper shows detailed experimental results to corroborate the effectiveness of the algorithm MTTS.

**Limitations And Societal Impact:**


### Limitations

Please refer to Main Review.

### Societal Impact

The paper focuses on general framework of a wide class of problems, so I do not see any potential negative societal impact.

**Main Review:**


The paper proposes a new setting of multi-task bandits, and provides a rather complete answer to it by showing both theoretical and empirical results. The proofs are easy to read and easy to follow. The writing is very well. The paper also provides the codes for the experiments, which is appreciated.


Assumption 1 is a little bit annoying to me. It is possible to remove it under other multi-task bandit setting [arXiv:2102.04132]. Since the setting in this paper is not like transfer learning (i.e. all tasks follow the same distribution), I wonder if it is possible to remove Assumption 1? The authors claim it is standard, but there are indeed some papers that does not require a counterpart of Assumption 1 (like [arXiv:2102.04132]). I think the authors should further justify it.

The paper seldom mentions the upper confidence bound (UCB) algorithms, which can definitely be used to solve this setting  (though it might get worse regret bound). I suggest the authors add a discussion about them. Also, the paper lacks a lower bound. Since the authors propose a novel notion of regret, the lower bound cannot be immediately seen, and thus I suggest the authors discuss about it.



**Time Spent Reviewing:**

6

---

> ### Author Response · Authors · 2021-08-10
> **Response to Reviewer sdSm**
>
> We sincerely appreciate your careful reading, constructive questions and suggestions.  In the following, we present our point-by-point responses to your comments.
>
> * **Assumption 1.**
>     Many thanks for this great suggestion.
>     It should be feasible to relax the assumption, which is essentially similar to the "stochastic contexts v.s. adversarial contexts" choice in linear bandits [5], i.e., whether we assume the features are generated i.i.d. or chosen by an adversary (so the sample covariance matrix might be ill-conditioned).
>     In the current version, similar to some literature (e.g., [4] and [9]), we choose the first option to focus on the main idea and also to be consistent with the task distribution view.
>     However, following the standard proof approach with adversarial contexts (e.g., in [6]), we should get the same bound in $\tilde{\mathcal{O}}$.
>     Due to the space limit,
>     we first provide a proof stretch here.
>     Notice that the only place we need Assumption 1 is in Lemma 2 and 3, where we apply properties of the linear mixed model with i.i.d. data to control the estimation error (and sampling error) of the task-specific prior mean $\Phi_i\boldsymbol{\theta}$ at rate $\mathcal{O}(\sqrt{d / i})$ with high probability.
>     Here $i$ is the index of the current task.
>     This result mainly leads to an $\mathcal{O}(\sum_{i=1}^N \sqrt{d/i}) = \mathcal{O}(\sqrt{Nd})$ term (page 15, appendix G), the product of which with the single-task Bayes regret $\tilde{\mathcal{O}}(\sqrt{TK})$ leads to the $\tilde{\mathcal{O}}(\sqrt{NdTK})$ term in our multi-task regret.
>     Without Assumption 1, we can bound the estimation error by (approximately)
>     $\mathcal{O}(||\Phi_i (\sum_{j=1}^{i-1}\Phi_j^T \Phi_j )^{-1}
>     \Phi_i^T||)$ with high probability, the summation of which from $i = 1$ to $n$ can be bounded similarly as $\tilde{\mathcal{O}}(\sqrt{Nd})$, following similar arguments of Lemma 3 of [7].
>     The proof technique relies on careful relating the cumulative prediction error with the eigenspace of the growing design matrix, and is largely standard in the literature (starting from [8]).
>     We sincerely appreciate this comment, which makes our theory more general.
>     We will update this generalization in the final version with full details.
>
>
>
> * **UCB.**
> We greatly appreciate this valuable suggestion!
> UCB is definitely another popular algorithm framework.
> The main challenge to adapt UCB to our problem is that, unlike Bayesian bandit algorithms which have a prior that can incorporate additional information,
> it is less flexible for UCB to share information, under our setup. Directly constructing confidence bounds for $\boldsymbol{r}_i$ under the hierarchical model is typically computationally challenging.
> Your suggestion inspires us to consider Bayesian UCB [1], a framework that selects arms according to the upper quantiles of the posteriors, instead of sampling reward vectors from the posteriors as TS does.
> Bayesian UCB also naturally fits our problem and benefits from information sharing.
> We believe this is an interesting idea worthy of study.
> We plan to add the discussions to our final version,
> emphasize the UCB nature of several meta algorithms cited in our current version,
> and also discuss UCB and Bayesian UCB as future directions.
>
>
> * **Lower bound.**
> Many thanks for another valuable comment.
> We agree some discussions about the lower bound are quite meaningful.
> We acknowledge the challenge to derive such a lower bound, as the multi-task regret is the difference between two Bayes regrets, which requires a quite precise description of the behaviors of Bayesian bandit algorithms under our setup.
> Up to the best of our knowledge, similar bounds are absent in the literature, and remain a challenging open question as noted in [2, 3].
> We conjecture that the dependency on $\sqrt{NTKd}$ is unavoidable, as the problem complexity naturally scales with these parameters, and most possibly at least at a root-n parametric rate.
> We will add related discussion in the final version and emphasize this as a future direction.
>
>
> * [1] Kaufmann et al.. On Bayesian Upper Confidence Bounds for Bandit Problems. http://proceedings.mlr.press/v22/kaufmann12.html
> * [2] Kveton et al.. Meta-Thompson Sampling (http://proceedings.mlr.press/v139/kveton21a.html)
> * [3] Bastani, H.; Simchi-Levi, D.; and Zhu, R. 2019. Meta Dynamic Pricing: Transfer Learning Across Experiments. Available at SSRN 3334629 .
> * [4] Yang et al.. Provable Benefits of Representation Learning in Linear Bandits
> * [5] Han et al.. Sequential Batch Learning in Finite-Action Linear Contextual Bandits
> * [6] Agrawal, Shipra, and Navin Goyal. "Thompson sampling for contextual bandits with linear payoffs."
> * [7] Chu, Wei, et al. "Contextual bandits with linear payoff functions."
> * [8] Auer, Peter. Using Confidence Bounds for Exploitation-Exploration Trade-offs. Journal of Machine Learning Research, 3:397–422, 2002.
> * [9] Bastani, Hamsa, David Simchi-Levi, and Ruihao Zhu. "Meta Dynamic Pricing: Transfer Learning Across Experiments." Available at SSRN 3334629 (2019).

---

### Official Review · Reviewer_DATJ · 2021-07-19

**Rating:** 7
**Confidence:** 4

**Summary:**

The paper considers the multi-task bandit setting with a contextual information related to the task. The setting is defined in a fully Bayesian way, pathing the way to a Thompson Sampling (TS) bandit algorithm to handle this setting. A worst-case regret bound with respect to the parameters of the task-generator is proven and experiments on synthetic data show that the proposed model has smaller regret than baselines independently of the entropy of the task with respect to the contextual information.


**Limitations And Societal Impact:**

Recommender systems are typical application of bandit algorithms. However, by recommending the preferred items to users, they tend to restrain the user to these preferences which may raise concerns regarding the corresponding societal impact.



**Main Review:**

Having access to a contextual information regarding tasks, the proposed Bayesian model and the corresponding TS algorithm are strait-forward. However, the theoretical analysis and the experiments demonstrate the interest of the proposed approach with respect both to meta-learning approaches overlooking the contextual information, and to non-meta-learning approaches.

At first read, the presence of both $d$ and $K$ in the regret bound is surprising as a task is defined by its context and the unknown vector $\theta$ of dimension $d$. Obviously, this reasoning is wrong, as a task also depends on the randomness in its generation which may be expressed by a vector $\delta_i$ of size $K$ (see by example Equation (3)). This point could be stressed out in the paper.

Note that the contextual multi-task setting may be reduced to standard contextual setting using (i) the extended context $\Phi'(x_i, a) := (1_{i,a}^T, \phi(x_i,k)^T)^T$, where $1_{i,a}^T$ is a length $NK$ indicator vector taking value 1 at $((i-1)*N+a)$-th entry, and (ii) the extended parameter $\theta' = (\delta_1^T, \dots, \delta_N^T, \theta)$. Such reduction inflates the size of the contextual information, leads to badly conditioned context-vectors, and is somehow equivalent to considering the tasks as independent. However, I'm wondering how it would compare to the proposed approach in terms of regret.

## typos
* L230: I guess the parameter of the indicatrice function should be $i(l) = i(m)$



**Time Spent Reviewing:**

3

---

> ### Author Response · Authors · 2021-08-10
> **Response to Reviewer DATJ**
>
> We sincerely appreciate your careful reading, constructive questions and suggestions.  In the following, we present our point-by-point responses to your comments.
>
> * **Rate dependency on K**.
> Thank you for the thoughtful comment.
> MTTS needs to learn the heterogeneity for task instances of $K$ arms using $d$ features, and hence the complexity naturally grows with $K$ and $d$. We will emphasize this rate dependency in the final version to provide more insights into our regret bounds.
> *  **Reduction to contextual bandits.** This is a very interesting idea. The regret for linear TS is known as $\tilde{\mathcal{O}}(p\sqrt{H})$ [1], where $p$ is the dimension and $H$ is the horizon. Take the Gaussian bandits with linear mixed model (refer to section 4.2, page 6) as an example: in your formulation we have $p = (NK + d)$ and $H = NT$, therefore the Bayes regret is bounded by $\tilde{\mathcal{O}}((NK + d)\sqrt{NT})$, which is even larger than the one for individual-TS as $\tilde{\mathcal{O}}(N\sqrt{TK})$. This is somewhat surprising as your formulation essentially merges the inter-task layer and intra-task layer of MTTS.
> The bound reflects that the theory for linear-TS is not tailored to our problem (e.g., the feature vector is a high-dimensional sparse indicator vector).
> Besides, we note that the reduction from a hierarchical model to a contextual bandit model is not always possible, e.g., for our Beta-Bernoulli logistic model or when the inter-task layer is not a linear function.
> We will include this discussion in the final version.
> * We will fix the typo and add the discussion on societal impact of bandit-based recommender systems in our final version. Thank you for these valuable suggestions!
>
> * [1] Daniel Russo and Benjamin Van Roy. Learning to Optimize Via Posterior Sampling. arXiv:1301.2609 [cs], January 2013. arXiv: 1301.2609.

---

### Author Response · Authors · 2021-08-29
**Thank You for the Comments and We are Glad to Address Any More Questions**

We would like to appreciate all reviewers for your time spent and valuable comments, and also our AC for handling this paper! We hope our point-by-point responses clarified/addressed your questions about our paper, and please feel free to let us know any other feedbacks/questions/comments before the end of the rebuttal period. We would be also happy to do any follow-up discussion.

In general, we are so honored and glad to learn about that you all find the paper *interesting* and *well-written*:

* Reviewer DATJ: "the theoretical analysis and the experiments demonstrate the interest of the proposed approach";
* Reviewer sdSm: "The paper proposes a new setting of multi-task bandits, and provides a rather complete answer to it by showing both theoretical and empirical results.", "The proofs are easy to read and easy to follow. The writing is very well";
* Reviewer ALPC: "This paper is well written. It has a good motivation and a general introduction to the problem. It has sufficient discussions on related work";
* Reviewer JzUf: "To the best of my knowledge I am not aware of the usage of metadata to explain bandit task similarities. I found the idea interesting and novel.", "I found the paper clear and well written."

Most suggestions are around *adding some discussions* (on 'rate dependency on K', 'insights from a simple reduction to contextual bandits', 'UCB', 'regrets under misspecification', 'alternative frameworks') or *paper layout* ('​move robustness analysis to the main text', 'move [6] to the related work section'). We found all these comments quite valuable. We are grateful and will incorporate your suggestions/our discussions in the final version. Reviewer sdSm's question regarding Assumption 1 also helps further extend our theory. We are working on a detailed proof following the stretch we presented, and will add to the final version.

Reviewer sdSm and JzUf suggested to discuss about the *lower regret bound*. We acknowledge the technical challenge and noted this is an open question (lower bound on the price to pay with a misspecified prior in TS, under general setups) in the literature and hence missing in most related papers. Simchowitz et al. achieved some progress but the setup is still restricted. We agree this is a very interesting question and will emphasize this as a future direction.

Reviewer ALPC and JzUf asked about *real experiments*. We acknowledge the challenge due to the absence of open-sourced meta bandit dataset, and applied our algorithm to MovieLens to demonstrate its usefulness and attempted to address your concerns. The results show our method performs preferably and there are interesting findings that are consistent with the motivations of this paper (e.g., features are useful but heterogeneity should not be ignored). All code/data/details/insights have been uploaded for you review. We plan to add this part to the final version and please feel free to let us know your comments.

Our framework is quite general and can be applied to solve transfer/cold-start problems, address low-traffic issues, allow personalized decision making for bandit tasks in the long tail, etc. In practice, one can consider adopting offline training (for the inter-task layer) and online deployment (for the intra-task layer) (see Algorithm 2 as an example), and complex generative models can be used for the inter-task layer.

Finally, many thanks again and feel free to let us know any other questions. We will try to respond as soon as possible before the end of the rebuttal.

---

### Decision · Program_Chairs · 2021-09-27

**Decision:**

Accept (Poster)

**Comment:**

Thank you to the authors and the reviewers for their contributions to the conference! This paper proposes a new bandit formulation with metadata, which they solve using a Bayesian hierarchical framework. They then propose a meta Thompson sampling algorithm and show regret guarantees using prior alignment. The reviewers uniformly appreciated the paper, so my recommendation is to accept the paper. A few concerns were raised regarding the lack of a lower bound and experiments on real data. For the former, I would suggest that the authors add some discussion on why this is challenging to do. For the latter, please add the new MovieLens experiments. Additional clarifications based on the rebuttal response are of course welcome as well.